# CFD Investigations of Water Supply and Distribution Systems of Ancient Old and New World Archaeological Sites to Recover Ancient Water Engineering Technologies

**Charles R. Ortloff** [1,2]

1   CFD Consultants International, Ltd., 18310 Southview Avenue, Los Gatos, CA 95033, USA; ortloff5@aol.com
2   Research Associate in Anthropology, University of Chicago, Chicago, IL 60637, USA

**Abstract:** New to archaeological studies is the field of paleohydrology characterized by the use of modern hydraulic engineering and computational fluid dynamics (CFD) analysis of ancient urban and agricultural water supply and distribution systems. Examples are presented in nine chapters of CFD investigations of old and new world archaeological sites (several of which are World Heritage sites) using FLOW-3D CFD software to bring forward new discoveries revealing the depth of ancient water engineers' knowledge and creativity not previously noted in the archaeological literature. As modern analysis methods reveal technical details of ancient water systems, equivalent ancient technologies exist that were used in the design and operation of ancient water systems, albeit in formats, texts, and origins yet to be discovered. The nine chapters to follow present brief summaries of ancient sites' water systems and the use of CFD and modern hydraulic engineering methods to discover the water engineering knowledge base used by ancient water engineers. Results are new revelations unknown in the current literature of ancient sites. Paleohydrology studies presented serve to add a further dimension to the history of ancient new and old world archaeological sites by bringing forward added details of water engineering projects accomplished by ancient engineers.

**Keywords:** archaeology; paleohydrology; CFD; ancient hydraulic engineering; ancient world heritage sites; new and old world archaeological sites; recovery of lost technologies





## 1. Introduction

Water supply and distribution systems in the ancient world reflect hydraulic engineering technologies that in many ways are comparable to modern practice. One way to uncover the hydraulic engineering technologies used by ancients is to create computational fluid dynamics (CFD) models of their urban and agricultural water transfer and distribution systems and, together with use of modern hydraulic engineering analysis methods, uncover hydraulic engineering technologies used by the ancients in their water system designs. Use of CFD analysis in the new field of paleohydrology provides details of the water engineering used to address the many problems ancient water engineers faced to supply their agricultural and urban water requirements. CFD models together with analysis of ancient fluid mechanics structures provide visualization and understanding of water flow patterns within these structures that reveal the design intent and function created by ancient water engineers. By observation of the computed water flow patterns and the hydraulic engineering designs that produce these patterns, insight into the technology base available to ancient engineers is recovered by CFD analysis to gain insight into ancient engineering practice. To illustrate the use of paleohydrology to understand ancient water engineering, examples of ancient water transport and control features used in canals, channels, aqueducts, reservoirs, and pipelines serve as examples from the archaeological record for sites in ancient South America and the Middle East. For the nine sites examined in the present paper, technical details of FLOW-3D CFD usage (3D mesh size, number of computational cells, turbulence model, solution convergence criteria, fluid properties,

CFD model relation to Google Earth and onsite field photographs, CFD model description details) are noted in the specific references listed each chapter. It is noted that hundreds, if not thousands, of years of destruction by nature's forces as well as man's destructive interventions render many ancient water system remains a fraction of their original form and obscure original design intent. This constraint is typical of archaeological remains where limited data combined with unlimited imagination provide probable answers to the problems and solutions of ancient water engineers. As few documents on water system engineering have survived the centuries and those that do have technical terms that cannot be associated with modern hydraulic engineering technology usage, CFD analysis provides a start to uncover what ancient water engineers had available in their knowledge base to solve water supply problems. In chapters to follow, CFD usage applied to several sites illustrates the use of paleohydrology to uncover the hydraulic knowledge base of ancient societies, and serves as an introduction to this new field.

## 2. The Water Supply and Distribution System of (300 BCE–CE 106) Petra in Jordan

Aspects of the water supply and distribution system of Petra in Jordan are detailed in the literature ([1], pp. 244–264; [2], pp. 137–169; [3], pp. 137–169; [4–8]). Of interest as a paleohydraulic CFD application is a hilltop water feature located in Figure 1, B-2, lower left side corner (Al Habis), associated with part of the lower Farasa housing district and a contributor to the Great Temple's (Figure 2) water supply system. From Figure 1, branches of the springs at Ain Braq and Ain Ammon serve the outlying areas near the Treasury (El Kazneh located in Figure 1, 1-C) and provide a partial water source for the Roman Soldier's Tomb complex (16, Figure 1), the Renaissance Tomb area sites (17), and a channel to serve the southern canal branch of the Siq water system. Water supply to the Wadi Farasa hilltop area (Figure 1, lower lefthand corner of B-2) may originate from an Ain Braq canal branch and/or the northern branch Siq pipeline to serve the downhill residence quarter with further continuance on to the Great Temple area and other sites on Colonnade Street (25, Figure 1) that include the marketplace areas and the recently excavated temple-pool complex [9–11] on the northern part of Colonnade Street.

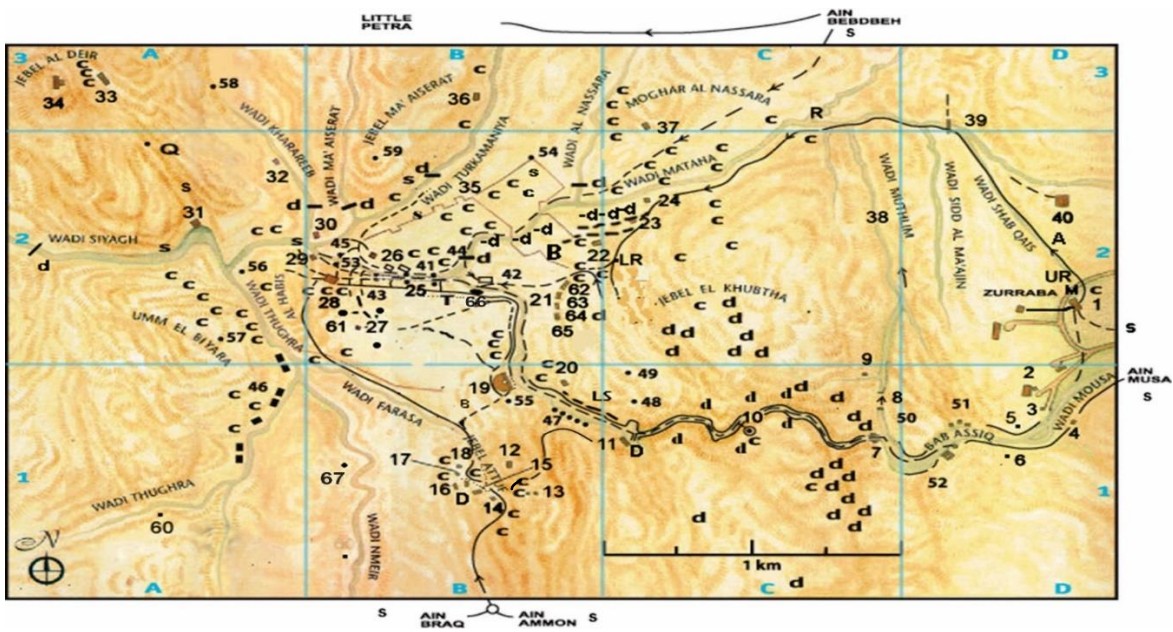

**Figure 1.** Petra with numbered site names given in Appendix A.1. Figure from [2], p. 266.

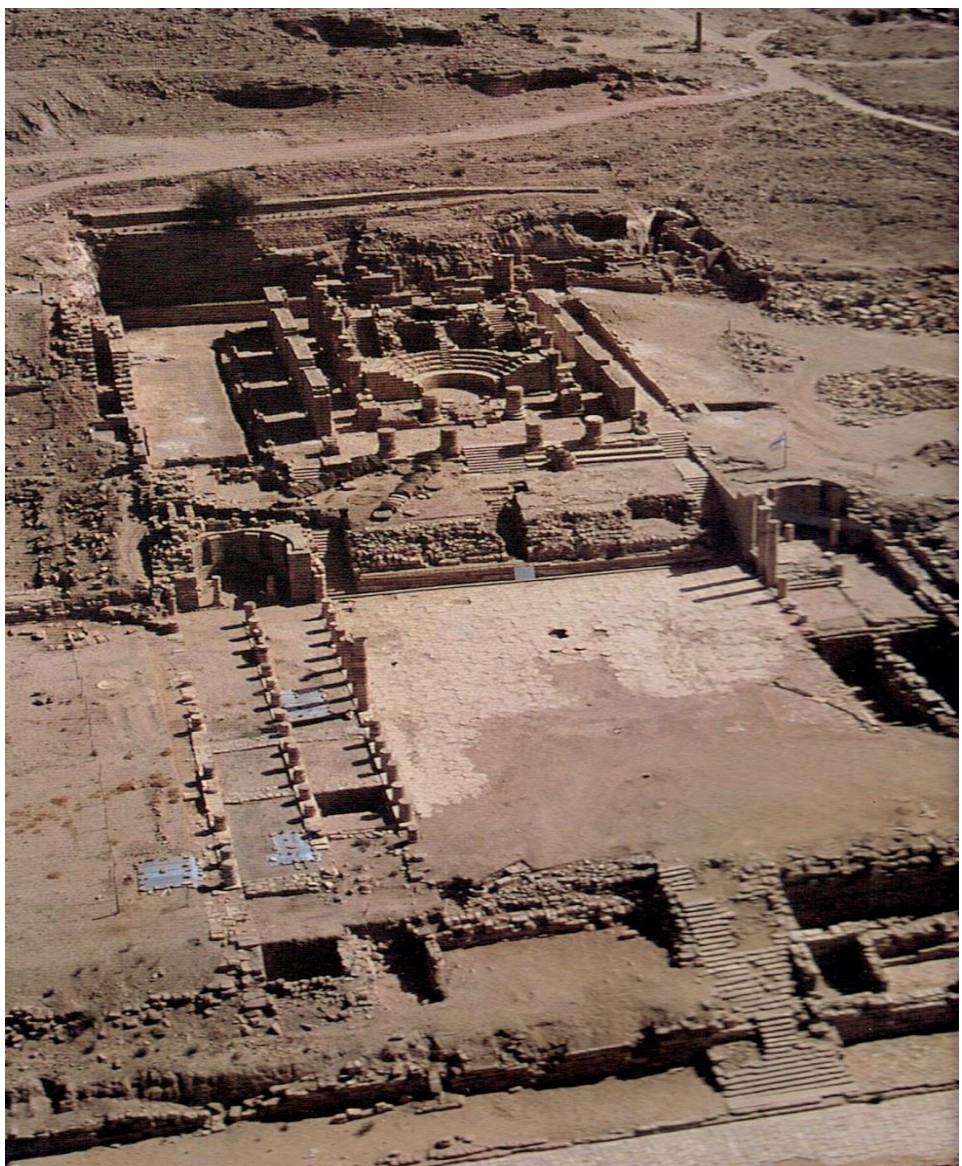

**Figure 2.** The Great Temple (28, Figure 1). Figure from [2], p. 293.

The Great Temple's water supply originates partially from extension of the Siq pipeline past the Theater (B-2, 19), the Marketplace (B-2, 25), the adjacent Paradeisos Water Basin, and the Temenos Gate Al Habis hilltop to the south of the Great Temple. From notes available from the early (1812) Petra explorer Johann Burckhardt, bridges crossing the Wadi Mataha from a castellum (Figure 1, C-2) likely transferred additional pipeline water to the Great Temple and Paradeisos basin areas from springs in that area. Most pipelines are of Nabataean origin, dating from ~300 BCE to CE 106 with later Roman pipelines primarily in the marketplace area, the Great Temple area, and the Roman–Byzantine bath complex [12,13] located in Figure 2 (the top left structure) dating from late Roman occupation of the city. This bath complex had an earlier Greek precedent structure with a novel water supply system later modified or rebuilt in later Roman–Byzantine times, as Section 8 details.

From the Al Habis hilltop junction (lower left corner of B-2, Figure 1) higher in height and to the south of the Great Temple, a pipeline component assemblage exists (Figure 3), likely sourced from the Ain Braq pipeline (B-1) and/or a Siq pipeline extension. A further pipeline branch on the hillside western slope above the Wadi Farasa streambed (B-1 to the southwest corner of B-2) originating from Ain Braq spring source exists but is yet to be

confirmed by excavation. Dual downslope pipelines emanate from the Figure 3 assemblage, with one pipeline leading to a downslope housing district and the other to the Great Temple. Remnants of this pipeline's presence were noted by conversations with excavators from the Brown University Joukowsky excavation team active in Great Temple excavations in the 1990s.

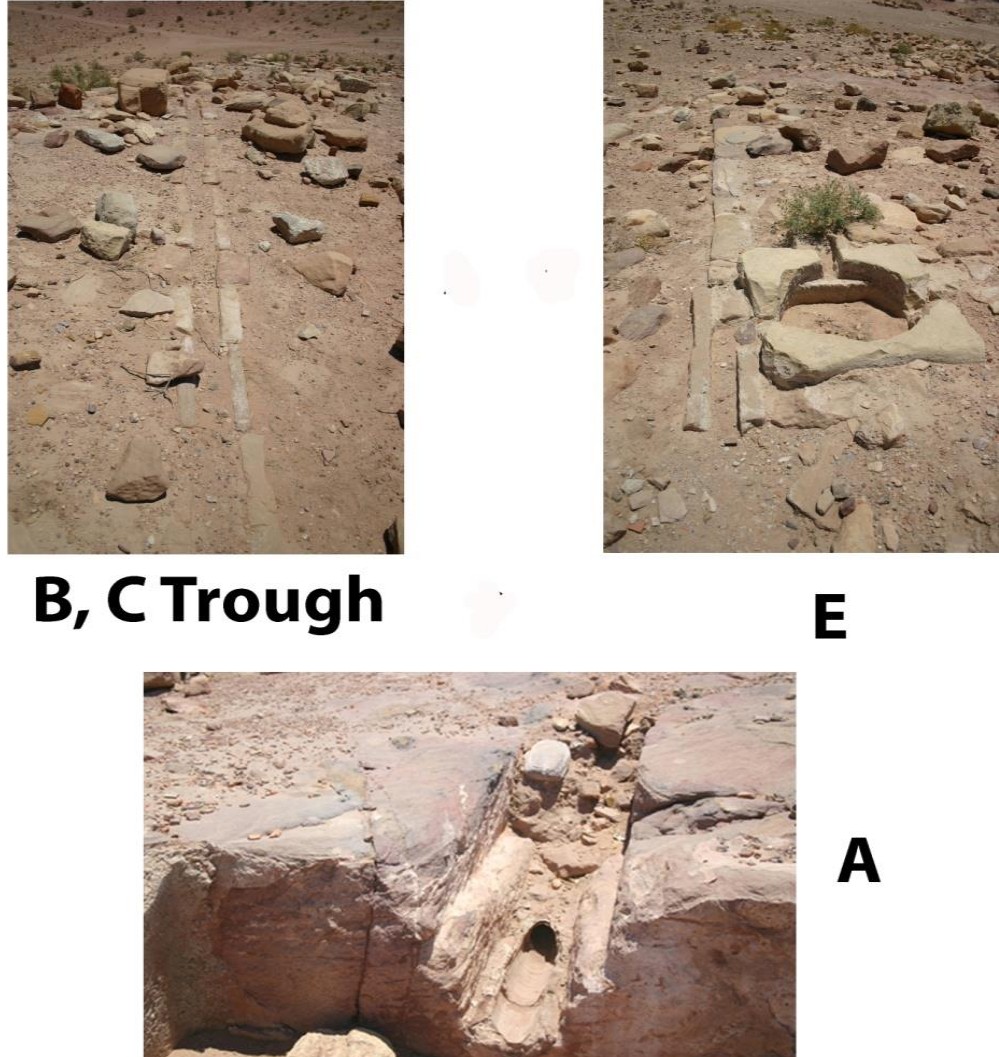

**Figure 3.** Assemblage components A–C and E of Figure 1, lower left hand B-2 corner of the El Habis hilltop pipeline water supply system. Photo by author on site in Jordan.

It is noted that a Crusader outpost exists on the Al Habis hilltop, indicating a water source from a pipeline emanating from the Ain Braq spring source. Excavated pipeline traces outside of the southernmost back wall of the Great Temple exist ([1], p. 266), further indicating a pipeline connection (Figure 4D,C) derived from an Ain Braq pipeline source to the Figure 3 assemblage. Also located in the Great Temple area above its inner semicircular *Theatron* (Figure 2) is a rear wall, outside of which was a trough (no longer present) that captured pipeline water from the Figure 4D,C pipelines and channeled it to an underground reservoir in the paved area west of the *Theatron*. This reservoir had a top channel to lower reaches of the Great Temple, as excavations from the Joukowsky excavation team indicated. As Great Temple excavation soil was deposited on Al Habis slopes over this deeply buried pipeline, positive determination of its existence awaits excavation. In summary, the Great Temple had multiple water sources. One pipeline from the Figure 3 assemblage was sourced by the Ain Braq spring and Siq pipeline extensions (Figure 1, B-1). A further water source

likely exists from a further branch of the Ain Braq pipeline into the Great Temple area, yet to be confirmed by excavation. The Ain Braq pipeline shows multiple branching to the near El Kaznah Treasury area, the Roman Warrior's Tomb complex, and now to the Wadi Farasa area to provide supplemental water supplies to the elite housing area and further on to the Great Temple area and the Paradeisos pool complex located on the main thoroughfare of Petra [9–11]. Only further excavations of the totality of the Ain Braq water supply system can reveal its true extent.

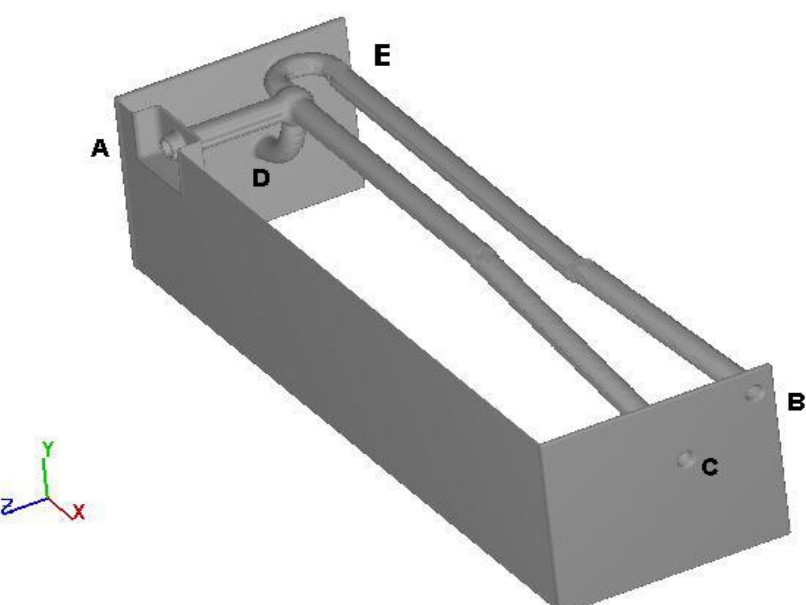

**Figure 4.** FLOW-3D CFD model of the pipeline connections constructed from Figure 3. Pipeline inner diameter ~15 cm. Figure from [2], p. 291.

Combining excavated connection details of the components shown in Figure 3, the CFD model shown in Figure 4 originated from in-place field observations of A–C and E components.

Assuming an Ain Braq and/or Siq extension supply branch to deliver water to pipeline D, the pipeline component assemblage (Figure 4) consists of basin A with a pipeline segment to E shown in Figure 3. From junction E, pipeline E–B supplied water to the downhill Farasa housing district ([9–14], p. 136) [15], while pipeline branch E–C provided water to the Great Temple and to the nearby Paradeisos water basin area through additional piping from the Great Temple area. Given water supply to the Great Temple, water was transferred to internal Great Temple reservoirs and directly to lower portions of the Great Temple that had drainage to the Wadi Mataha stream bed (Figure 1, path C-3, C-2, B-2, A-2) with drainage to the Wadi Siyagh, A-2. The FLOW-3D computer model (Figure 4) utilizes an input water velocity for full flow input conditions in a level portion of the pipeline at D before the dual pipeline branches E–B, E–C continue to steeper terrain to supply downhill destinations.

The A basin served to manage transient flow excursions, as Ain Braq spring output can vary depending on spring recharge history from seasonal rainfall duration and intensity as well as water diversions into other pipeline branches upstream of the main distribution center with its dual pipeline branches.

The next step is to determine the hydraulic function of the basin A connection to the distribution junction E. For one of several hydraulic functions, the basin pipeline A served to divert flow away from the dual downhill branches if these were blocked during repair or if the distribution center was blocked to divert water to other pipeline branches of the Ain Braq system. The blockage then diverts water to basin A shown in Figure 3. The flow into the basin continues until a hydrostatic pressure balance of water accumulated in basin A balances with the hydrostatic pressure in water supply pipeline D. Thus, an automatic means for system shutoff exists once the lower exit ports of B and C (Figure 3) are blocked.

A further, more sophisticated hydraulic engineering use of the Figure 3 pipeline assemblage configuration originates from the observation that subcritical full flow conditions exist in the supply pipeline D due to the low slope lead-in pipelines to D and pipeline flow resistance from inner-wall pipeline roughness. Due to the hydraulically steep declination angles for dual pipelines E–B and E–C, input full flow past the E junction area is transforms to supercritical partial flow in steep pipeline branches E–B and E–C. This flow transition from full to partial flow induces partial vacuum regions above the partial flow water surface in E–B and E–C pipelines. As the subterranean branch pipelines exits are submerged into reservoirs at their destination locations, air is drawn into pipeline exit openings toward the partial vacuum regions. Transient water/air bubble streams then proceed upstream in dual steep declination pipelines branches to counter partial vacuum regions, leading to flow intermittency and an oscillatory hydraulic jump within the pipeline branches converting pipeline supercritical flow to subcritical flow downstream of the hydraulic jump. This unsteady hydraulic phenomenon affects flow stability of the entire pipeline system, leading to pulsating discharges into the B, C downstream reservoirs as well as affecting the stability of input flow at D.

Additionally, pulsating flows cause internal pressure level changes that can flex pipeline joint connections, causing leakage. A cure to transient unstable conditions in the steep declination dual E–B and E–C pipelines was anticipated by Nabataean water engineers by air supplied through the basin pipeline A (Figure 3) into the dual pipelines at junction E to counter and eliminate partial vacuum regions to promote stable flow conditions.

Air led into partial vacuum regions cancels the vacuum regions and subsequent flow instabilities in the dual subterranean pipeline branches. For situations for which hydraulic instabilities cause a backflow to propagate upstream in the D supply in line, and/or water surges in the E–D supply line, Figure 5 indicates a water deposit in the A basin to help damp flow instabilities.

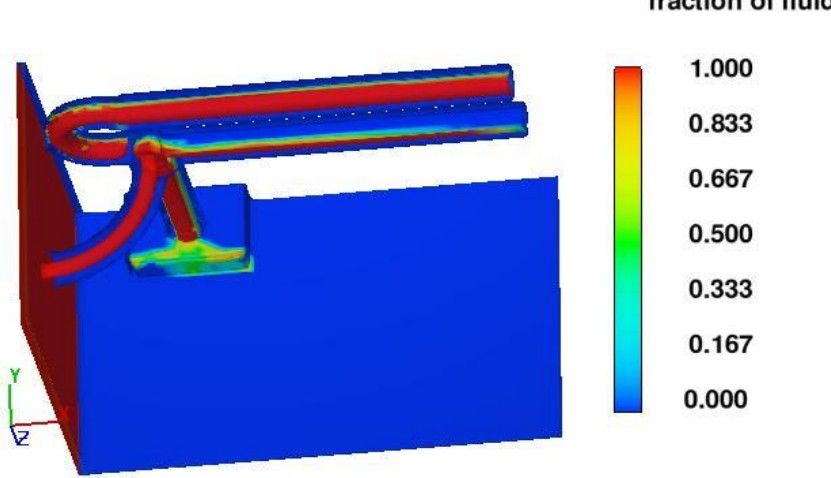

**Figure 5.** Backflow into the A basin caused by surges in the E–D and E–C water supply line. Figure from [2], pp. 291–292.

In the figures to follow, fluid fraction ff = 1 denotes water; ff = 0 denotes air; and intermediate ff values denote water droplet/air mixtures. Figure 5 indicates full flow at E with the development of partial, supercritical flow in pipeline branches E–B and E–C due to the large declination slopes (~15–20°) of the E–B and E–C pipelines.

Figure 5 indicates the transfer of air into the E–B and E–C pipelines through the A air pipeline to prevent flow transient instabilities from developing as the partial vacuum regions leading to flow instabilities are cancelled. Figure 6 represents the steady state establishment of flow to downstream housing areas and to the Great Temple. Water that has been deposited into the A basin during a transient event together with full subcritical

flow from D supplies the E location where partial flow originates in the E–B and E–C pipelines due to their large declination slopes. Figure 6 represents steady-state conditions.

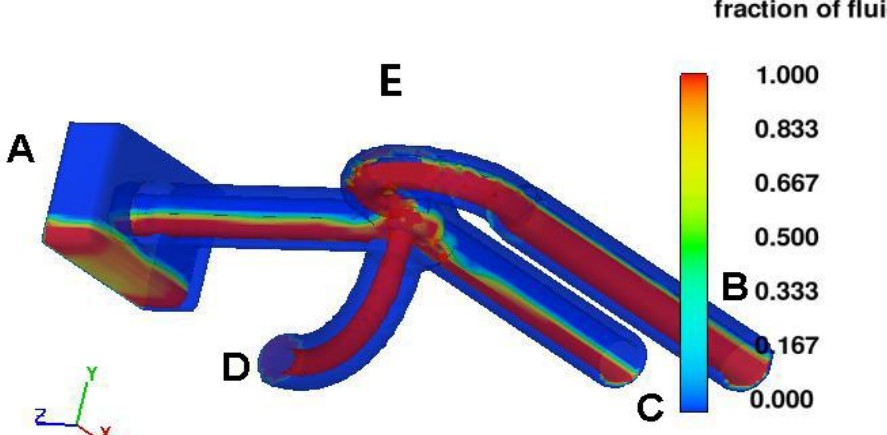

**Figure 6.** Establishment of steady-state flow conditions in pipelines. Figure from [2], pp. 291–292.

In Figures 5–7, blue areas denote the presence of air in pipelines, red areas denote water, and intermediate colors denote water–air mixtures caused by splashing and/or local turns in the piping system that cause transient water instabilities leading to a localized surface air–water mixing event. For Figure 6, the steady-state design flow rate is achieved; for Figure 5, backflow into basin A has occurred due to an overdesign flow rate occurrence.

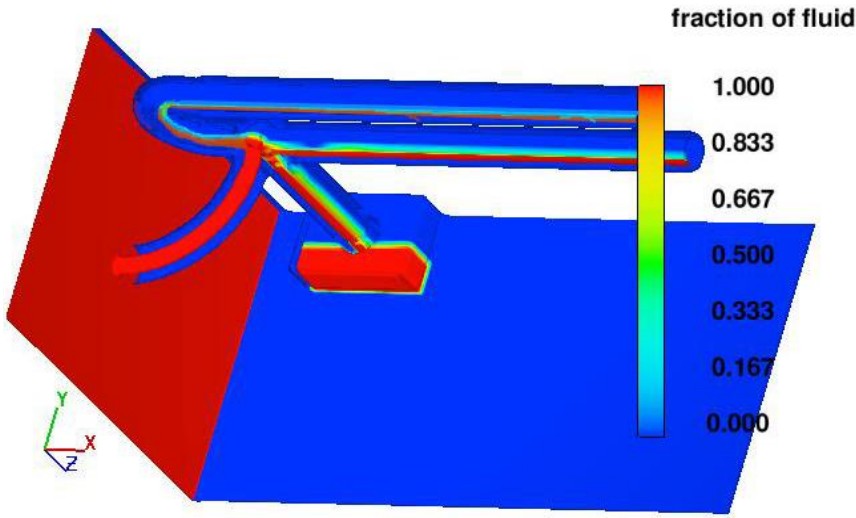

**Figure 7.** Automatic termination of flow by balancing input D hydrostatic pressure with back pressure from water accumulation in basin A. Figure from [2], pp. 291–292.

Figure 7 represents a situation where an excessive flow transient occurs in supply pipeline D. Due to flow resistance of downstream piping, water accumulates in basin A as the design system flow rate is exceeded. When hydrostatic pressures match from basin A and input pipeline D, then flow terminates as now pipelines E–C and E–B contain mostly air. Before this situation, the water transfer to basin A helps to damp transient water, and pressure surges as may occur upon system start up to promotes rapid development of steady-state flow conditions. The basin A air pipeline then serves to (1) minimize transient water surges originating from air pulses entering partial vacuum regions by providing an air passageway to E; (2) facilitate damping of transient water instabilities in the downstream high declination slope dual pipelines E–B and E–C by transferring backup water caused by transient hydraulic jump motion to the basin and flow rate surges from supply pipeline D,

thus promoting steady-state flow stability. One further hydraulic function of basin pipeline A is to divert flow away from the dual downhill branches if these were blocked during repair or if the main water distribution center was blocked to divert water to other pipeline branches of the Ain Braq water supply system. Thus, it is an automatic way to maintain B and C system shut-off through a pressure balance using A and D features.

In summary, the hydraulic engineering use of the pipeline configuration shown in Figure 3 originates from the observation that subcritical full flow conditions exist in the lengthy supply pipeline to D due to its low slope and the inner pipeline surface roughness pipeline conditions resulting from the many pipeline joint connections. Given half-meter individual pipeline sections, and the long length of the supply pipeline, many thousands of joint rough connections ensure full flow conditions. Due to the hydraulically steep declination angle in dual pipelines E, input full flow past the junction area is converted to supercritical partial flow in steep pipeline branches E–B, E–C. This flow transition induces partial vacuum regions above the partial flow water surface. As the subterranean branch pipelines exits are submerged into reservoirs at their destination locations, air is drawn into pipeline exit openings toward the partial vacuum region. The intermittent air bubble stream proceeds upstream in dual steep declination pipelines branches to counter the vacuum regions, leading to flow intermittency and an oscillatory hydraulic jump in the dual pipeline branches. This hydraulic phenomenon affects the flow stability of the entire system, leading to pulsating discharges into the E–B, E–C downstream reservoirs as well as affecting the stability of the input flow at D. Additionally, pulsating flows cause internal pressure changes that can affect pipeline joint connections causing leakage. The cure to this unstable condition was anticipated by Nabataean water engineers when air is supplied by the basin pipeline A. Air led into the partial vacuum regions cancels the vacuum regions and subsequent flow instabilities in the dual subterranean pipeline branches. As pipeline air access is shown to have substantial benefits, use of this technology was evident in other contemporary sites. For example, top holes in stone block piping are evident to alleviate partial vacuum creation as flow transitions from full to partial flow on a steep slope ([1], p. 312) for a Laodicean water supply system and are evident in much of the surface pipelines at Ephesus where partial, open channel flow exists in pipelines [15]. Given access to a wide range of water technologies as a trade center, Petra could incorporate many learned water control technologies from trade route partners from different continents.

### 3. The Water Control System for Moche-Chimú (CE 400–1400) Pacatnamu in the Jequetepeque Valley of North Coast Peru

The ceremonial site of Pacatnamu in the Jequetepeque Valley of north coast Peru originates from Mochica origins in the Early Intermediate Period (CE 250–750) with later occupation by Chimú conquest in the Late Intermediate Period (CE 1000–1400). The Late Intermediate Period and earlier Middle Horizon Sicán influence plays a significant role in valley aqueduct design, as evidenced by coastal far north water systems in their homeland areas. Later occupation of the Jequetepeque Valley by Inka conquest in Late Horizon times (CE 1400–1532) indicates Pacatnamu site abandonment and Inka efforts to collaborate with marginal Chimú occupation of the valley area centered at the site of Farfan to provide water to Inka administrative buildings vital to control valley resources [16]. Vital to Pacatnamu existence during Moche and Chimú occupation was a reliable water supply for urban and agricultural use provided by the nearby Jequetepeque River during multiple occupations by different societies. Details of the site's long history and water system development are provided in ([1], pp. 111–122), with one area of special interest that demonstrates an application of advanced water engineering knowledge.

Figure 8 indicates a plan view of a main canal originating from the Jequetepeque River source that has multiple earth-fill aqueducts crossing multiple F6 shallow erosion gullies formed from drainage of the eastern valley hills from rainfall and El Niño events. As the canal approached the deep Hoya Hondada Quebrada, valley water engineers decided on a different aqueduct design that lies at a low level (about four meters) above the Hoya

Hondada quebrada bed supplied by a steep declination canal that runs from the elevated embankment northern edge to the low-level aqueduct (Figure 9). The canal proceeds across this low aqueduct to supply water to Moche and Chimú farming settlements as well as continuing to agricultural field systems south of Pacatnamu. The main canal has multiple off-take canals that supply irrigation water to field systems west of the main aqueduct, as indicated in Figure 8. Of interest is that field system agriculture on irrigated flat surfaces is supplemented by fields located on sloped embankments to the F6 quebradas; this is carried out for crops that need only partial sunlight for their growth and is a testament to the need to use any available land suitable for agriculture.

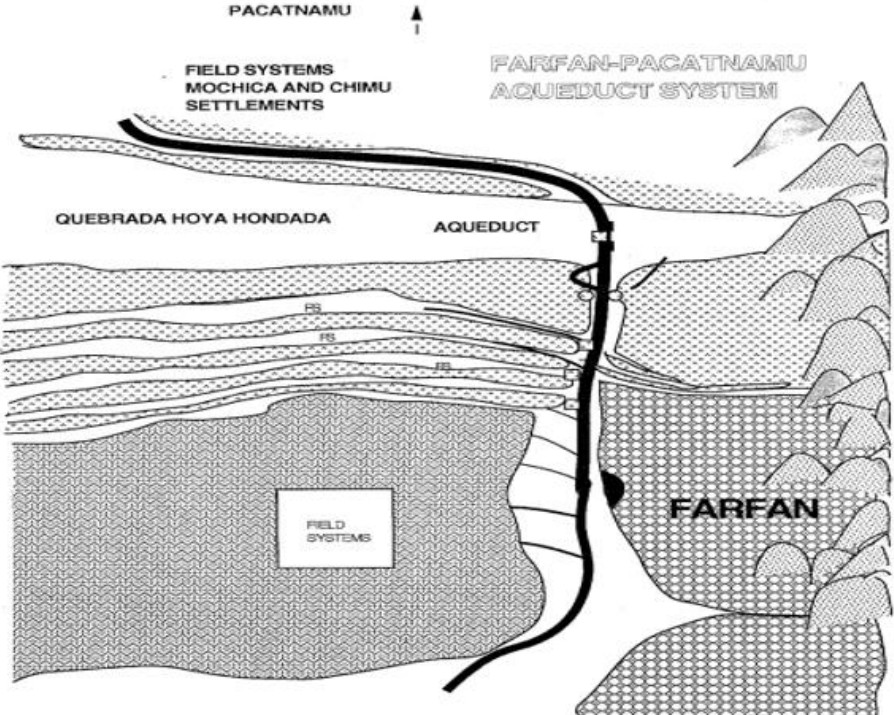

**Figure 8.** Details of the Farfan–Pacatnamu aqueduct system. Figure from [1], p. 114.

Figure 9 shows the low-level aqueduct crossing the bottom of the wide Hoya Hondada aqueduct. As indicated from Figure 8, the canal continues westward on to the shallow slope of the northern Hoya Hondada embankment to provide water to Moche and Chimú farming settlements serving the main southern field systems along its route toward Pacatnamu. The low aqueduct is an earth-fill structure without canal bed lining other than layers of silt deposit from extensive use. No notable culverts built through the bottom of the low aqueduct to drain accumulated rainfall runoff behind the aqueduct are apparent from field observation, nor are any other culverts found at multiple aqueducts crossing F6 quebradas, as indicated in Figure 8. As five steep short aqueducts supply the lower Figure 8 field systems, high-speed water flowing down these steep channels encounters dual stone "chokes" that create a hydraulic jump before the choke to lower the water velocity entering the flat field systems. The choke opening widths at the bottom of the steep aqueducts are carefully spaced to make the passage flow at the maximum critical flow with excess flow as the source of the hydraulic jump. This design was made to limit the entry water speed to level field systems to limit soil transfer from field system internal channels.

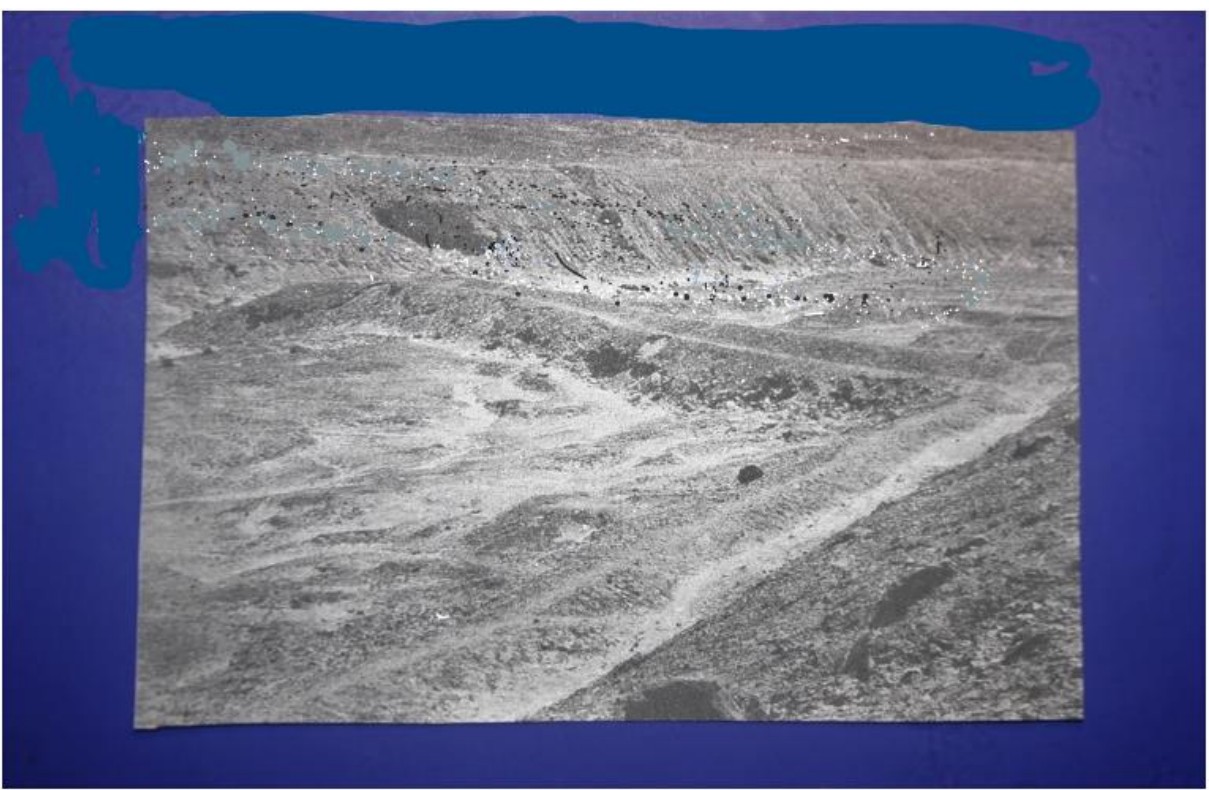

**Figure 9.** The low-height Hoya Hondada Aqueduct viewed from the northern embankment. The aqueduct channel proceeds downhill along the southern bank of the quebrada to a plateau with Moche and Chimú settlements. Figure from [1], p. 115. Photo by author at site in Peru.

If the low aqueduct height were constructed at the same level as the ~15 m high embankment, it would require construction of a massive ~5 m high earth-fill dam structure to support the canal at the same slope as the approach canal length. During a massive El Niño flood event, water accumulating behind the ~15 m high dam would have hydrostatic pressure forces sufficient to wash the dam away, as it is composed of unconsolidated soils; this failure disaster was anticipated by ancient water engineers by using the low earth-fill dam of ~4 m height that would survive low hydrostatic pressures derived from water accumulated behind the low height dam. However, this design choice introduced a new problem. A steep canal descent was required from the high embankment to the low aqueduct height. As the low aqueduct channel was unlined, water descending the steep canal from the high embankment to the low aqueduct accelerates to high velocity due to gravity and would erode the unlined canal at the impact junction location. The solution to this erosion impact failure problem by water engineers is considered by a FLOW-3D CFD canal model of the canal geometry on the southern bank (Figure 10) just before the steep canal descent to the low Figure 9 aqueduct.

The Figure 10 canal segment has several features: (1) the canal width contracts (location A) as it approaches the descent canal continuance to the lower aqueduct; (2) an elevated left-side canal wall opening before the dual opposing obstacles exists, directing water flow into a side drainage channel (location C); (3) two opposing stone obstacles contract the flow in the contracted canal section (B); (4) further downstream, a right wall opening (location D) exists connected to the main canal. The canal segment shown in Figure 9 is on the order of 20 m length; canal bottom widths range from ~2.5 m in the canal wide (A) area to ~1.3 m in the downstream contracted (D) area. Canal bottom depth from the adjacent ground surface is on the order of ~1 m. While over 1000 years separates the construction of the canal by Chimú water engineers to the form observed from current field observations, excavation clearing of this canal segment provided a best estimate of its key features for the CFD model used to explore its design function to regulate water flow. Upstream of this

canal segment, the canal maintains the same near-rectangular cross-section shape with a bottom dimension of ~2 m.

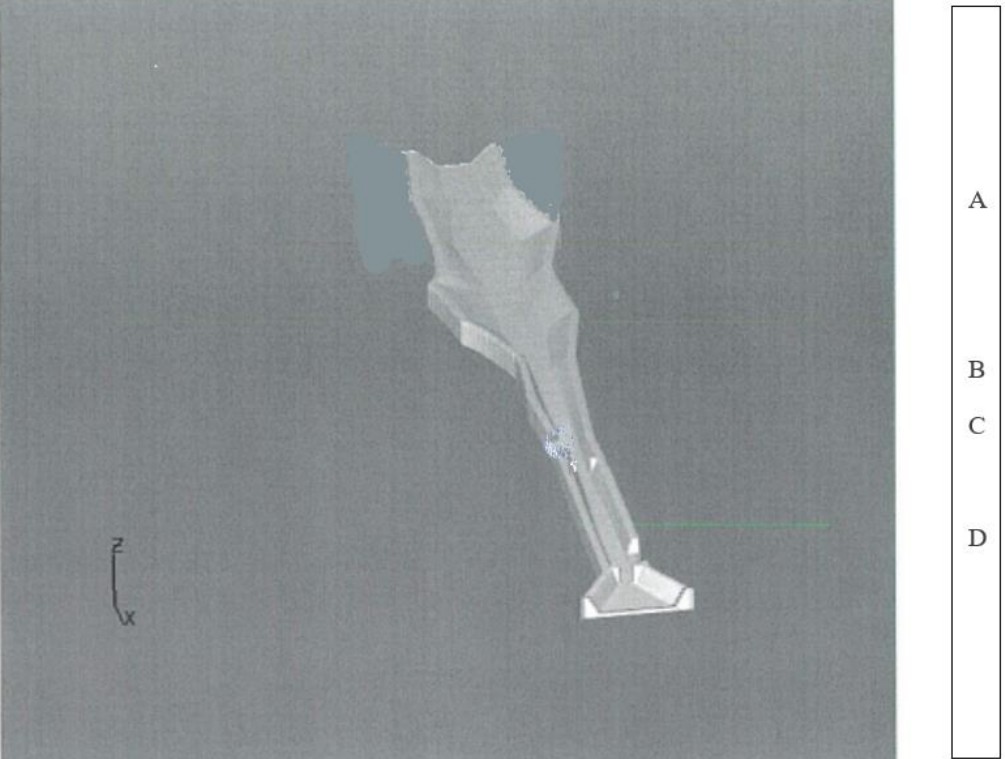

**Figure 10.** Northern embankment canal geometry ahead of the steep descent canal to the lower aqueduct. FLOW-3D model [1], p.118 modified by author.

Figure 10 indicates a further channel constructed from sidewall opening (Figure 10, location D) that carries a fraction of the canal water to intersect at point E, Figure 11, with the main body of water proceeding down the steep incline canal to the lower aqueduct. Given the low slope of the Figure 11 channel, the subtracted flow is at a low subcritical velocity value. This slow flow now intersects the supercritical flow in the steep declination channel at point E, Figure 11, causing a hydraulic jump to occur in the declination channel elevating the post-jump flow height and decreasing its water velocity. This solution lowers the impact velocity of water flowing on to the lower aqueduct from the steep descent channel and limits the erosion of the lower earth-fill aqueduct at the impact location. Water from both sources then continues across the low aqueduct to agricultural field systems.

FLOW-3D CFD analysis [17] reveals the design intent of the Chimú engineering solution. Again, a major design consideration is to limit damage to the low Hoya Hondada earth-fill canal system from El Niño flood events that can destroy the entire system. Subcritical velocity water proceeding down the Figure 10 sloped canal segment increases in velocity and height as the canal contraction zone (A,B) is approached. As the dual side wall (choke) obstacles (C) are encountered, critical flow matching the design canal flow rate is established that establishes the safe design flow rate through the area between side wall obstacles. When the canal flow rate exceeds this design flow rate from El Niño flood water washing into the canal along its length, water height before the choke elevates and proceeds through the elevated sidewall opening (Figure 10, location C) with the excess flow directed to a lower drainage area. To this point, the canal design flow rate (or flow rates lower than the design flow rate) is achieved to limit canal damage from excessive flood conditions. Flow at the design flow rate proceeds to location D (Figure 10), where a fraction of the flow rate is subtracted in the contour channel (Figure 11) to cure the water impact canal erosion problem on the low-height Hoya Hondada aqueduct impact location. The

Chimú water engineers thus solve multiple design problems to diminish the water impact destruction of the lower earth-fill aqueduct at the impact location on the lower aqueduct as well as threats from El Niño flood wash-in events. The aqueduct impact problem is largely mitigated by rerouting a portion of the canal flow, as Figure 11 describes, where a subcritical subtracted flow from the main canal intersects a supercritical flow in the steep declination canal to induce, through a hydraulic jump, a lower velocity combined flow, limiting erosion of the lower Hoya Hondada aqueduct.

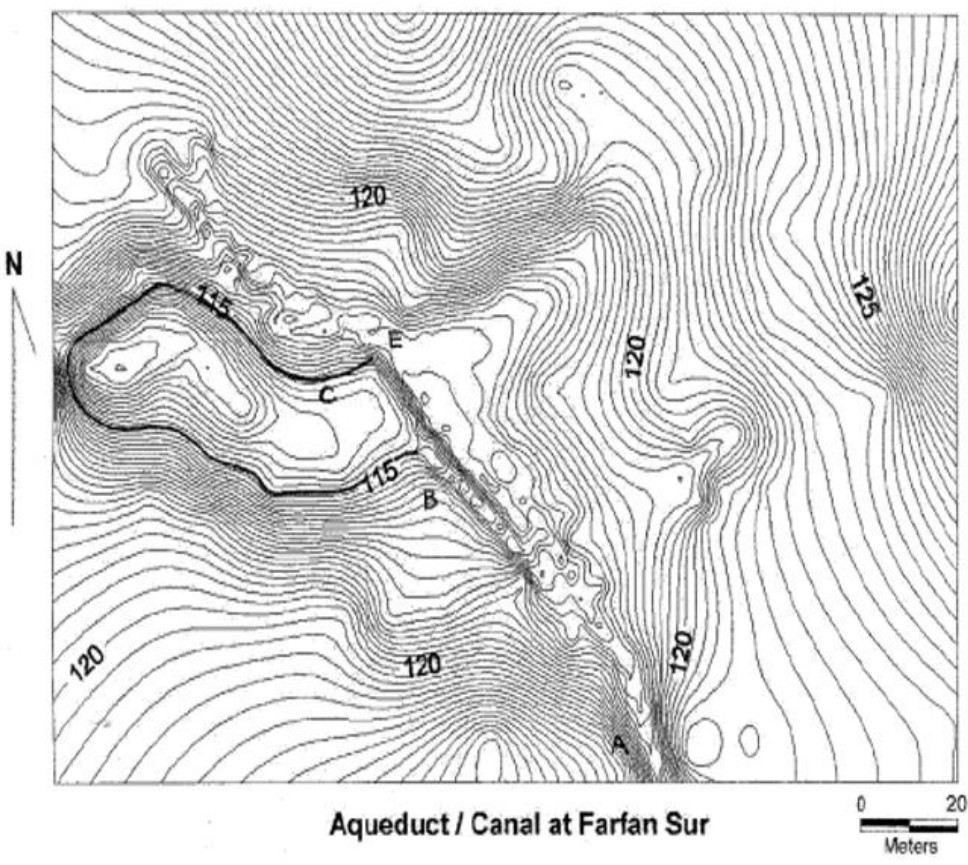

**Figure 11.** Low slope channel path from Figure 10, D channel wall opening to an intersection location midway on the steep decline canal path to the low aqueduct. Figure from [1], pp. 121–122.

Further details of the hydraulic engineering of the Chimú aqueduct are found in ([1], pp. 111–124). As precise surveying is required for the success of this engineering design, previous investigation of surveying accuracies ([2], p. 89) achieved by ancient Andean societies indicates a ~$10^{-3}$ degree accuracy achievement by the Late Intermediate Period Chimú society in Moche Valley canals adjacent to their capital city of Chan Chan. Later Late Horizon Inka canals achieve ~$10^{-3}$ degree accuracies, while an earlier Middle Horizon Tiwanaku canal pair indicates an accuracy of $\pm 10^{-4}$ degrees in a dual canal system characterized by two nearby canal water flows in opposite directions. By comparison, Roman surveying accuracies achieve similar accuracies as measured from onsite measurements at the Pont du Gard aqueduct.

Figure 12 illustrates the flow subtraction from Figure 10, D location at the design flow rate from FLOW-3D CFD analysis of the water system to counter aqueduct damage. The combined set of design adjustments indicated by Figures 10 and 11, B and C flow subtractions serve to preserve the canal from flood wash-in erosion damage and indicate the hydraulic engineering capabilities of Chimú water engineers to anticipate problems and provide innovative solutions to the problems.

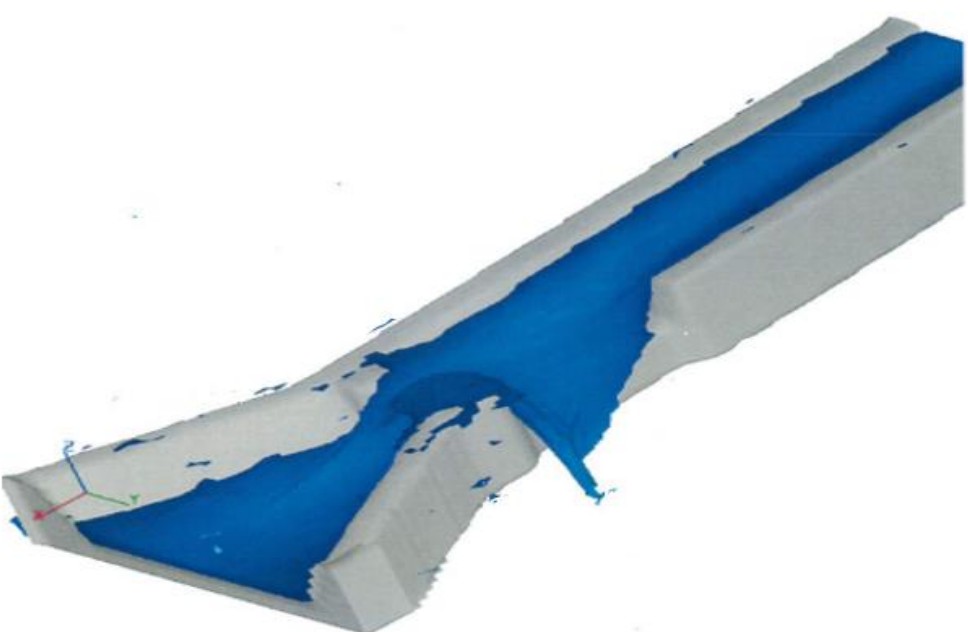

**Figure 12.** FLOW-3D CFD computed exit flow subtraction from the Figure 10, D location at the design flow rate. Figure from [1], p. 120.

The flow innovations represent a degree of hydraulic knowledge that only CFD analysis can bring forward to demonstrate aspects of the state of hydraulic engineering knowledge achieved by South American pre-Columbian societies.

## 4. CFD Analysis of a Self-Cleaning Water Drainage Channel at the Mid-Eastern Greek Site (300 BCE–CE 100) of Priene in Coastal Western Turkey

The site of Priene, located ~15 km from the coastal area of western Turkey, is unique as a Greek colony city without later Roman architectural influence. Given the tolerance of Greek civilization to multiple religions, beyond temples to Greek and Egyptian gods found on site (Figure 13A,B,D,J,M,N), an early Hebrew synagogue was recently found by archaeologists on site. Originally an ally of Athens, Priene was conquered by the Persians in the 6th century BCE. Later, as a member of Greek cities in western Turkey participating in the Ionian revolt against Persian occupation under Darius, the city was relocated in 350 BCE to its present inland site. The city was visited by Alexander the Great in 334 BCE, where his stay featured the planning of the conquest of the Persian Empire; a house reputed to be his residence during his stay in the city was given his name and served as my base of operations during many site visits. Later occupation by Seleucids and the Pergammon administration followed, with the city later incorporated into the Roman Empire. From Constantinople as the seat of the eastern Roman Empire, there followed later century occupation by Byzantine forces. Later in the city's history, a Byzantine church (P, Figure 13) indicates Christian occupation until later Muslim conquest of sites by the Ottoman Empire (and Constantinople) itself in the 15th century CE, when the Priene was finally abandoned.

Details on the history and water systems of the city are given in ([1], pp. 338–339; [15,18]). City water supplies largely originated from karstic spring sources in the city northern highlands that, through an extensive pipeline network, distributed water to many areas of the city, including the marketplace agora, the theater, the stadium, the multiple gymnasiums, and the city's council chambers (Figure 13).

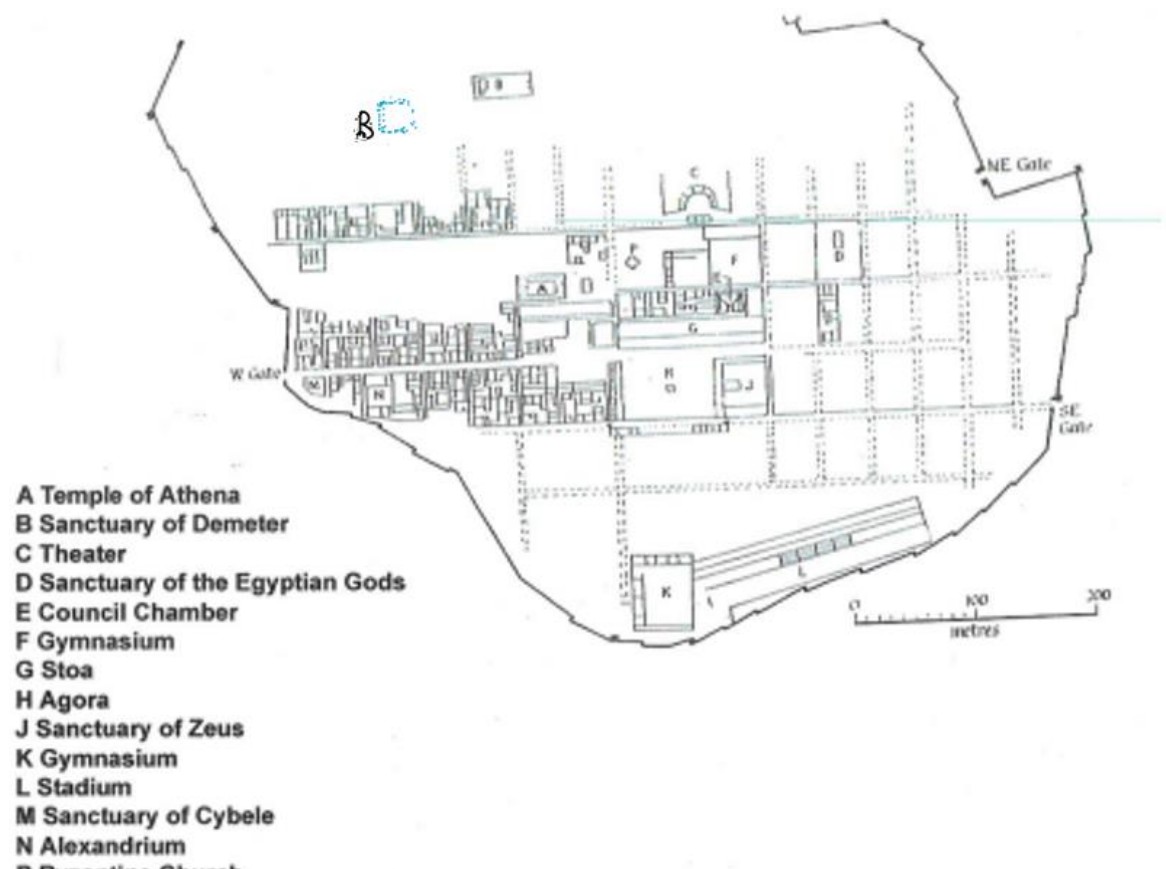

A Temple of Athena
B Sanctuary of Demeter
C Theater
D Sanctuary of the Egyptian Gods
E Council Chamber
F Gymnasium
G Stoa
H Agora
J Sanctuary of Zeus
K Gymnasium
L Stadium
M Sanctuary of Cybele
N Alexandrium
P Byzantine Church

**Figure 13.** Detailed map of walled urban Priene and inner city main structures. Figure from [1], p. 338.

Of interest to the present study is the water drainage system at the West Gate area of the city. The city (Figure 13) is composed of urban living quarters centered around the main city street leading to the West Gate. To supplement the pipeline water distribution system, large underground cisterns were built in the central housing district to collect channeled rainfall as well as "used" water from city residences supplied from the continuously flowing pipeline system. Drainage water from all occupied areas, and particularly from the busy marketplace agora site, was largely delivered to an open channel (Figure 14) on the main thoroughfare for passage through a bottom opening in the city wall. Water for ceremonial temples (listings given in Figure 13) required a high degree of purification for rituals; this was achieved by multiple silt collection basins inserted into pipelines that had access ports for frequent cleaning. Of interest then are water purification systems and water structure designs that have an "automatic" method of self-cleaning built into their design.

At Priene's West Gate (Figure 13), debris-laden drainage water from housing, the agora marketplace (H) and government buildings, temples, and the theater was served by springs north of the main street pass through the street channel (Figure 14) into the West Gate drainage opening (Figure 15) through the city outer wall. The street approaching the drainage structure has a steep channel that accelerates drainage water to supercritical (Fr > 1) conditions with the high-velocity water height approaching normal depth as it proceeds toward the drainage opening [19–21]. While a straight channel exit continuing the geometry of the Figure 14 inlet channel would normally be expected, the complex geometry drainage structure indicated in the Figure 15 CFD model reconstruction is in place from field measurements. Rapidly flowing, debris-laden water enters this structure from the positive y direction (Figure 15) and interacts with the complex internal shaping before exiting into a canyon outside of the city wall. The channel water height approaching the Figure 15 drainage structure is at the lowered height, high-velocity, normal depth and is contained within the lead-in channel without spillage by design. A stone baffle

plate exists to cover the top front part of the Figure 15 drainage structure that contains the input flow to avoid spillage and flooding of housing structures close to the city wall (Figure 13). The Figure 14 rectangular cross-section channel has a width of ~1.5 m, a depth of ~ 0.5 m, and a declination slope of ~20°. FLOW-3D CFD calculations (Figure 16) indicate complex streamline vortex structures accompanied by high levels of turbulence induced by flow passage through the complex Figure 15 structure. The intent of the Figure 15 design is now revealed: severe agitation and entrainment of the debris-laden flow ensures that debris settling does not occur as would be the case if a straight-through, lead-in continuance channel design were emplaced. The debris water load comprises suspended lighter particles and heavier mud and debris particles settled on the channel bottom that slowly move from water shear effects toward the Figure 15 drainage structure. The drainage system is "self-cleaning", as agitation of the water flow from CFD calculation (Figure 16) provides resuspension of heavier and lighter debris particles to clear the channel of suspended and settled waste particles.

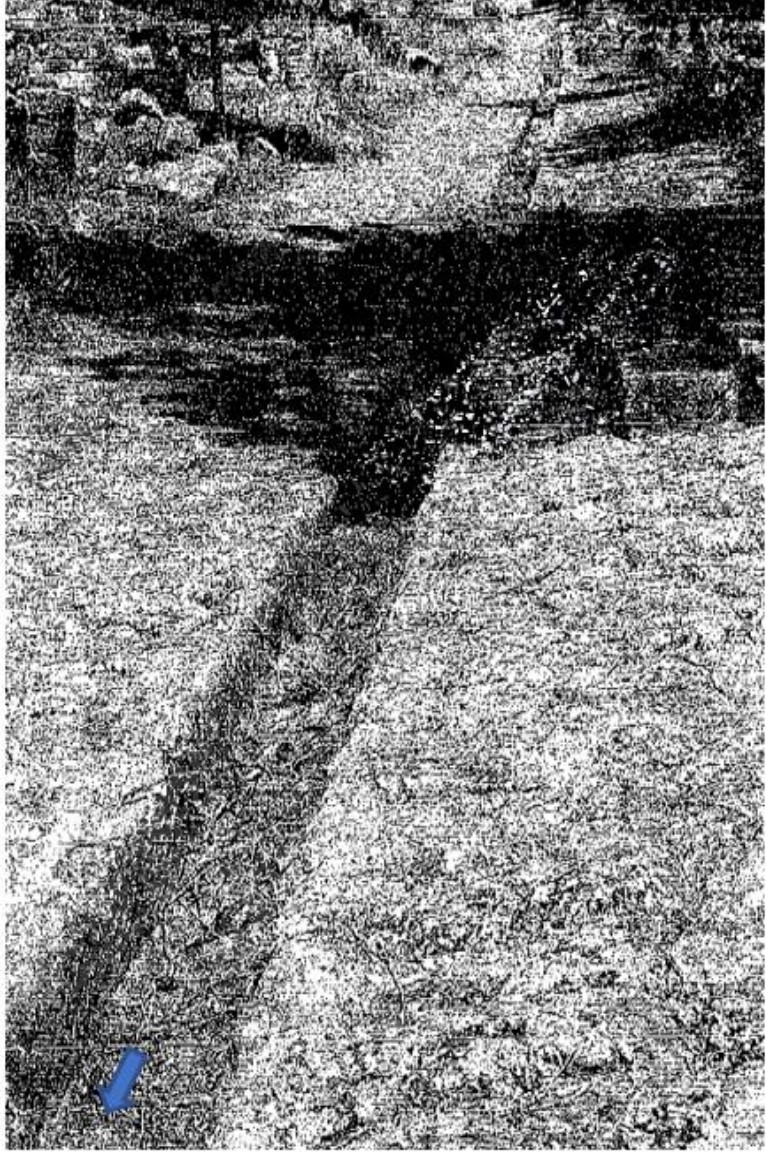

**Figure 14.** Steep channel along the center of the Priene main east–west throughfare, leading water drainage from housing and the agora to the complex West Gate drainage structure shown in Figure 15. Approximate mean channel width ~ 1.2 m; approximate mean channel depth ~ 0.25 m. Arrow denotes flow direction. Photo by author at Priene in Turkey.

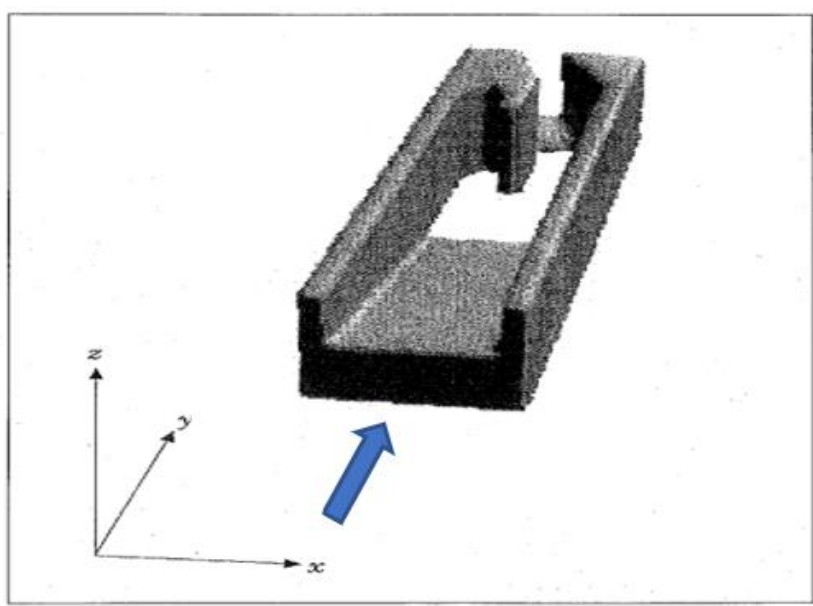

**Figure 15.** Drainage structure passing through the base of the West Wall, accepting drainage water from the Figure 14 channel. The arrow denotes the water flow direction. From [1], p. 341; FLOW-3D CFD model by author.

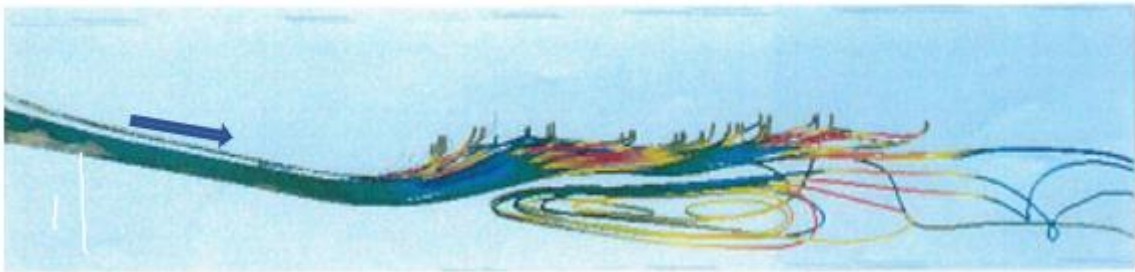

**Figure 16.** X plane CFD computed streamline patterns within the Figure 15 structure, indicating complex vortex patterns and high turbulence levels. Arrow denotes flow direction.

In summary, Greek water engineers provide a sophisticated self-cleaning channel exit design to facilitate the city's high hygiene standards observed by many water purification devices. It is noted that the pressurized pipeline supplying ceremonial water basins in the Sanctuary of Demeter (Figure 13B) proceed underground lower than the basins then rise upward to the basins; this design ensures that debris settles in lower pipelines to provide purified water to the basins.

## 5. Water Supply and Distribution System of 800–1400 CE Cambodian Angkor West Baray

The history and cultural development of Angkor in Cambodia is well known. Of interest are the two major 9th century CE barays (water reservoirs) shown in Figure 17, known as the West Baray and East Baray. For the scale of the site, the dimensions of the West Baray are 8.3 km length, 2.4 km width, and a depth of ~10 m from the top of the built-up containing mound structures bounding the West Baray. The two barays were designed to capture heavy rainfall from two monsoon seasons as well as water from northern rivers. Within the city itself, the north-to-south flowing Siem Reap river provided water to inner city settlements, temples, agricultural zones, internal city reservoirs, and local population settlement areas through many surface channels. As suspended soil particles are carried by rivers from rainfall runoff, silt is continually delivered into the West and East Barays, causing deep layers of sediments to deposit in the barays over centuries since initial construction in the ninth century CE. The original purpose for the barays is

somewhat obscure, as various scholars have attributed their construction to ceremonial purposes or city water needs through seasonal changes in rainfall availability. Various scholars have interpreted the purpose of the West Baray as the irrigation source for rice fields in the dry season; others propose a religious function based upon the discovery of an upper body fragment of a 10 m high bronze statue of Vishnu at the Mebon structure central to the West Baray. Of interest are the sites of Ta Prom, Pre Rup south of the West Baray, the sites of Ta Nei and Ta Keo to the west of the baray, and Angkor Wat to the far southeast of the West Baray. These sites have as their water source a diversion of the Siem Reap river that was designed to replenish the West Baray in the dry season. Over the centuries, the southern reaches of the West Baray have been modified to provide channeled surface water irrigation to downslope rice farmlands.

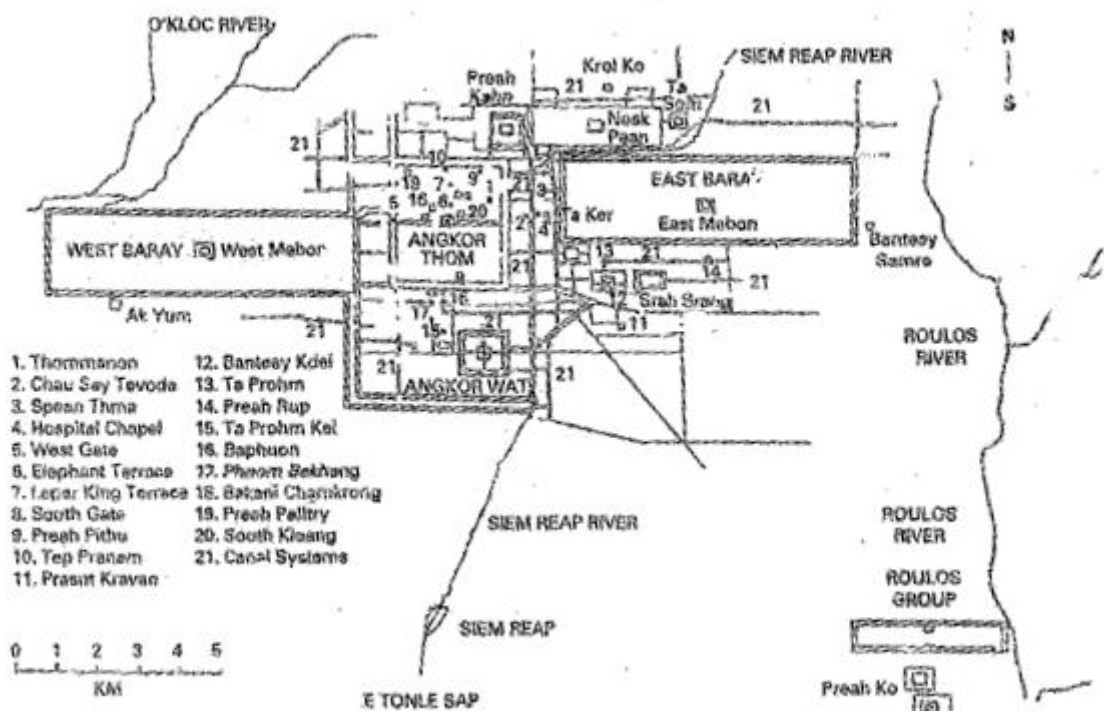

**Figure 17.** Plan view of urban Angkor and surroundings showing East and West Barays. Site names in Appendix A.2. Figure from [1], p. 359.

The present chapter Flow-3D CFD study focuses on the use of the West Baray in off-season water supply conditions to supplement rice farming areas south of the baray. Given the large population of Angkor, multiple-season rice farm cropping continuity was vital to sustain the city during the dry season. Efforts to maintain a high West Baray water level by rainfall collection and channeled river water were vital to provide the southern farming area sufficient groundwater to sustain continuous rice field agriculture.

In the figures to follow, t (time) represents months after the dual monsoon season ends and the dry season initiates; z denotes depth in meters below the original interior baray bottom surface. The ff fraction of fluid scale represents aquifer water concentration: ff = 1 represents water; ff = 0 represents air in porous aquifer soil; intermediate ff values represent air/water mixture states within the porous aquifer. Rainy season water seepage from the West Baray into the southern, downward-sloped agricultural field system aquifer south of the baray is maintained from rainfall and river sources providing water to the baray. The high hydrostatic pressure from West Baray water depth contained by the ~10 m high encircling embankment (when the baray is full) provides sufficient hydrostatic pressure to extend the water penetration distance and depth into the southern aquifer for several kilometers, sufficient to provide water to root systems for several crop types as well as to excavated pool areas for rice cultivation.

During the dry season, with limited input river flow, high evaporation rates, and limited seepage from areas north of the baray, the West Baray's water level drops to a lower level with reduced seepage into the southern field system area to maintain height groundwater height. For rice farming, a stable saturated aquifer underlying water pool bottoms is necessary to continuously maintain the rice farming conditions.

Given different varieties of rice cultivated in the southern area field systems, various field system configurations necessary for their growth can be implemented given different water pit levels dug into the saturated aquifer that sustain different crops moisture levels. The original West Baray bottom surface is approximately at the same level of the farming area just outside the ~10 m high, mountainous earth-fill barrier berm that encompasses the baray interior.

For the CFD study, the dry season West Baray water level is assumed low due to extended drought with a relatively dry southern field system aquifer. Figures 18–21 detail the computed CFD progression of water seepage from the West Baray into the southern reaches of the farm area aquifer during initiation of wet season rainfall. With the end of the extended drought period, monsoon rains return, filling the West Baray water level through rainfall accumulation, water input from the nearby river, and groundwater seepage from the northern, higher-level recharged aquifer. The CFD study then informs of the reactivation of the West Baray's main purpose as the monsoon season initiates and starts water transfer to the southern farm area aquifer to reinstate rice farming. The farm area saturation level maintained under wet season conditions is now of importance as the dry season initiates to determine the ability to continue rice farming as baray water levels decrease. Figures 18 and 19 show the aquifer fluid fraction (ff) concentration 3.75 and 4.08 m below the southern agricultural area surface 2.53 months into the dry season, resulting from a limited wet season rainfall level. The red zone is the low West Baray water level. Results indicate that water transfer into the aquifer initially proceeds slowly due to the low hydrostatic pressure at the baray bottom. Near-surface aquifer penetration is low, with remnant aquifer water fluid fractions remaining at depth from earlier monsoon season water deposits.

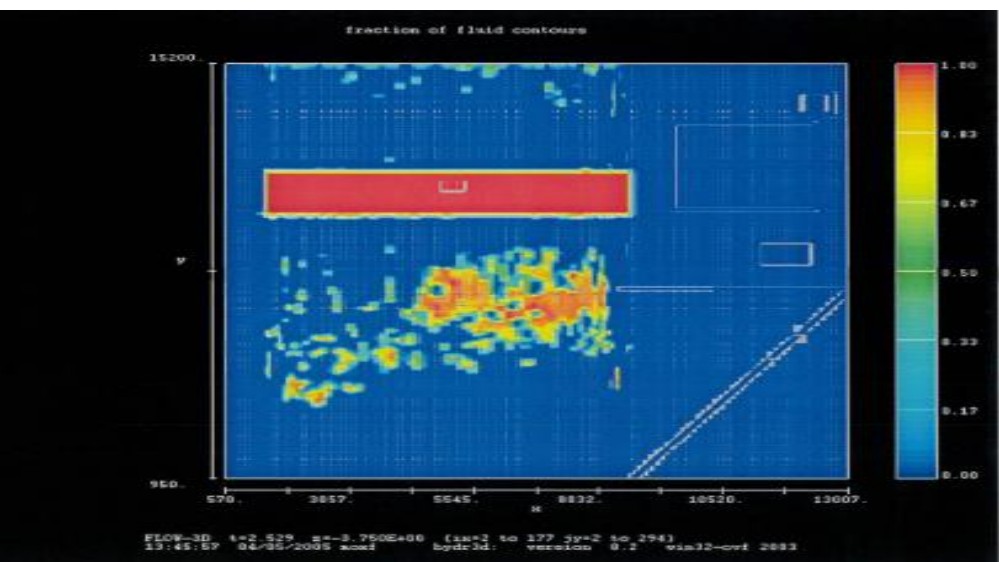

**Figure 18.** Aquifer fluid fraction (ff) concentration at 3.75 m below the southern farm area surface 2.53 months after the extended drought ends. Color version added to [1], p. 367.

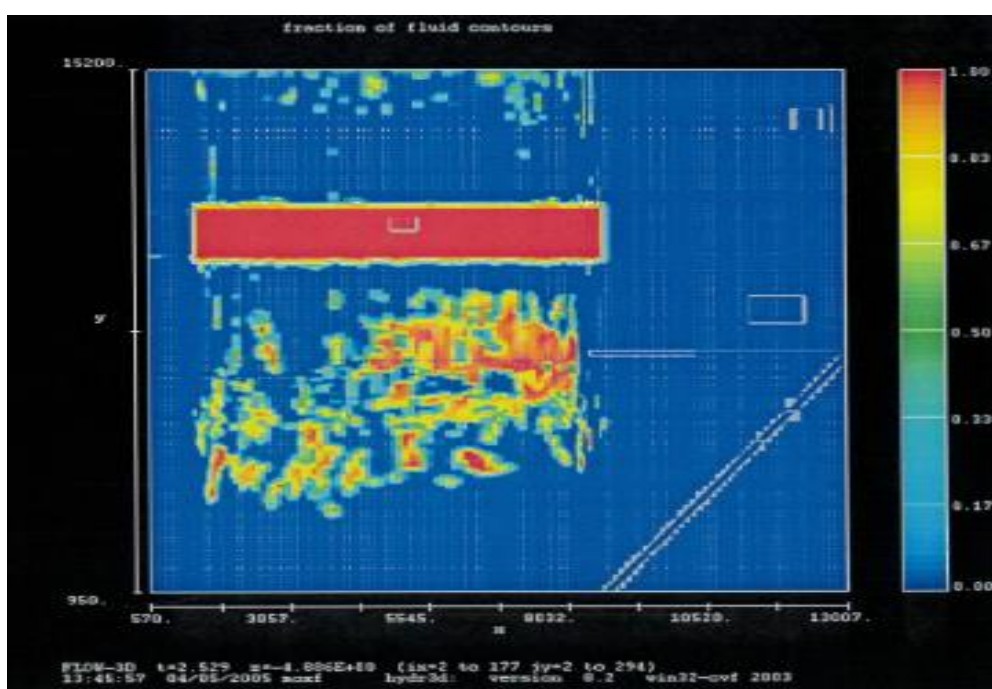

**Figure 19.** Aquifer fraction of fluid ff concentration at 4.08 m below the southern farm area surface 2.53 months after the monsoon season initiates. Aquifer water levels remain high at depth from earlier rainy season deposits. Color version added to [1], p. 367.

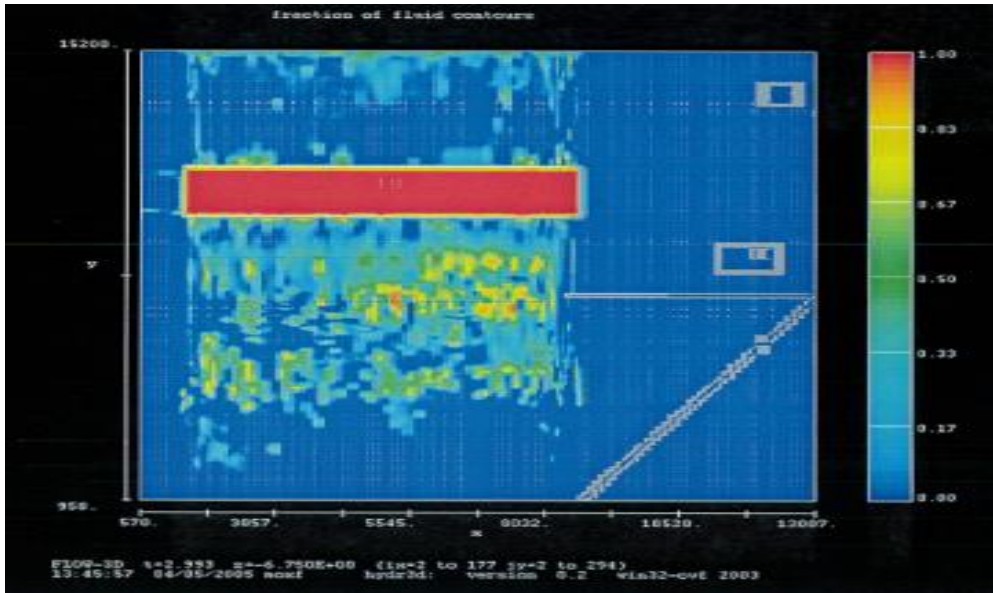

**Figure 20.** Aquifer fraction of fluid ff concentration at 6.75 m below the southern farm area several months into the monsoon season. Color version added to [1], p. 367.

As the rainy season initiates months later with baray infilling, near-surface aquifer infilling initiates (Figure 20)—this process is aided by the saturated aquifer at depth left over in the dry season, as Figures 17 and 18 indicate.

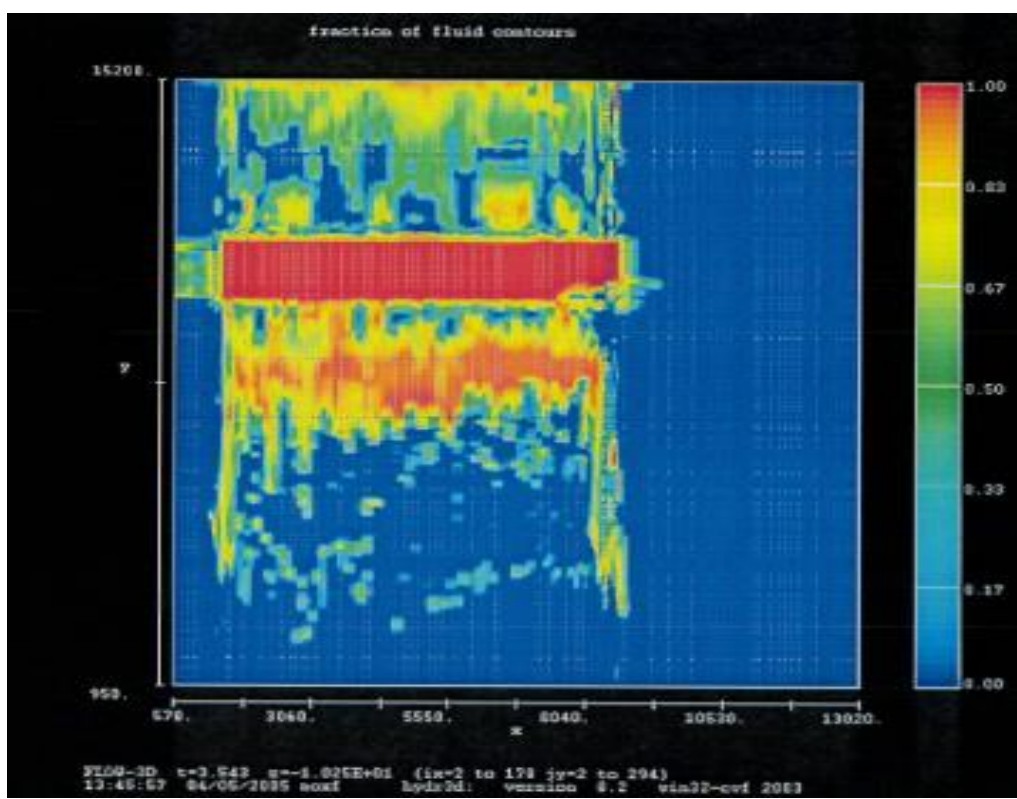

**Figure 21.** Aquifer fluid fraction ff concentration at 1.07 m below the southern farm area surface 3.54 months after the monsoon season ends. Note the later groundwater supply from ingested rainfall from the northern mountain area contributing to the baray water supply. Color version added to [1], p. 367.

Figure 20 indicates that as the West Baray continues to fill to higher water levels as the monsoon season initiates, the aquifer is now experiencing saturation to a large depth relative to the ground surface as the lower part of the aquifer is already saturated and is unable to accept further water input. This observation is confirmed by Figure 21 which indicates that at a later time, with continued rainy season baray filling, at a depth of 1.07 m below the ground surface, the southern agricultural area is saturated to a large depth ~ 1.5 km from the baray. This is the intended purpose of the West Baray, as now-excavated rice farming basins excavated into the deeply saturated ground surface fill with water suitable for rice cultivation. Under normal climate circumstances where infrequent drought periods exist, the West Baray retains a high water level from higher-elevation northern area aquifer seepage and river water input. Figure 21 is then the standard condition for rice farming through both wet and dry seasons, as (1) in the dry season, saturated soil conditions are maintained at depth from the previous wet season deposits and accessible for rice farming by pits dug to lower depths or for other crops with deep root systems; (2) in the wet season, a saturated aquifer at depth is available for rice farming (or other crops). With continued availability for farming during wet and dry seasons, the large population of Angkor can be well supplied with food resources.

At present times, the barays are no longer functional due to a large depth of silt accumulation that occurred after site abandonment at ~1430 CE. Several channels exiting from the central south side and westernmost corner of the West Baray are present, indicating water transfer to surface field systems to the west and south of the West Baray. These high-level through-berm water supply channels may have originated later in the site's life when West Baray high silt accumulation limited previously used aquifer-provided agricultural resources to southern field systems.

CFD results proceed from specified aquifer properties [22] (porosity, hydraulic conductivity, particle size, specific surface) and water properties (density, viscosity). Given

that fertile soil deposition from rain erosion of northern mountain range soils had occurred over millennia, the southern farming areas of the West Baray were well positioned to provide high rice yields. As this area is nearby to urban population centers, great interest in developing this area for rice farming was a priority for Angkor water engineers. CFD results presented indicate that Figures 18–21 represent the basis for sustained agricultural production throughout yearly seasonal changes, given the two monsoon events in June–September and October–November. Of note is that silt washing into and depositing into the barays over the many years of Angkor's existence slowly reduced their main function, as silt deposits presented a barrier for water transfer into the southern aquifer and reduced baray water storage capacity. At present, after centuries of site abandonment, many meters of silt cover the original bottom of the barays, cancelling their former practical use.

## 6. Landscape Change, Beach Ridge Formation, and Agricultural Field System Change from El Niño Flood Events at Preceramic (2500–1600 BCE) Caral on North Coast Peru

The preceramic site of (2500–1600 BCE) Caral is in the Supe Valley on the north coast of Peru, approximately 40 km inland from the Pacific Ocean coastline; Caral is adjacent the south side of the Supe River, as indicated by 13, Figure 22.

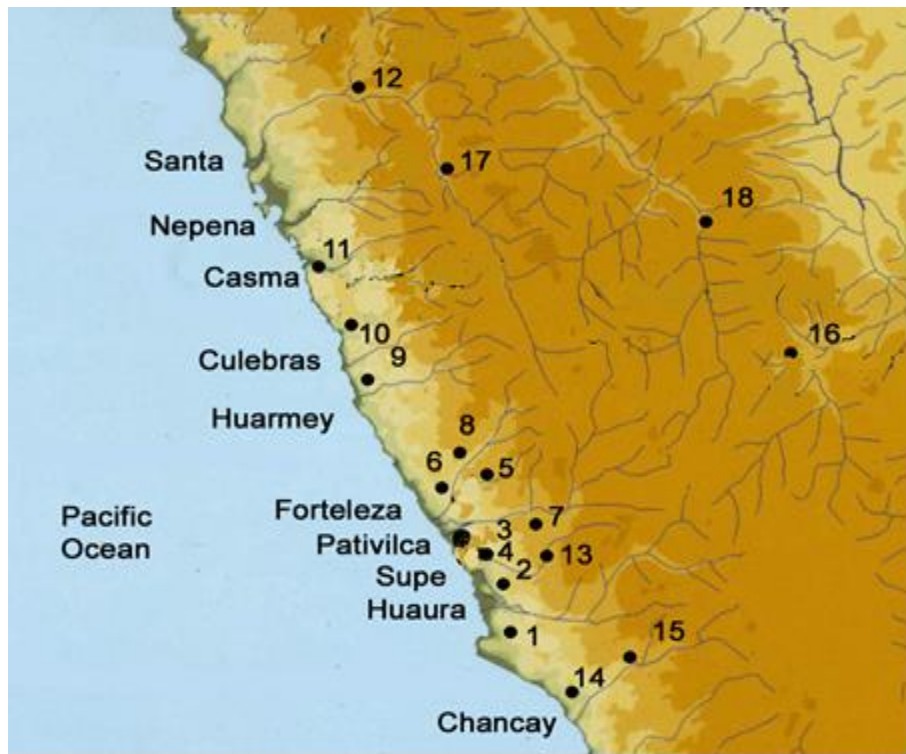

**Figure 22.** Location of Caral (13) in the Supe Valley on the north coast of Peru. Additional numbers indicated relate to other Preceramic, Formative, and Early Horizon sites in adjacent valleys. Color version of figure derived from [2], p. 224.

Caral is known as the "earliest city" in Peru owing to its founding date of ~2500 BCE; the site consists of many stone-faced pyramid structures, residential quarters, and several ceremonial buildings. Only recently discovered through detailed excavation efforts by Dr. Ruth Shady Solis, together with her many journal and book publications detailing site architecture [23–29], the site has received UNESCO World Heritage status. Given the discovery of a ~2500 BCE new world site with multiple elaborate pyramid structures (Figures 23–25) contemporary in time with early-dynasty Egyptian pyramid structures, Peruvian prehistory now emerges with new discoveries that indicate the presence of a previously unknown, very accomplished society that starts and underlies development of later Andean societies.

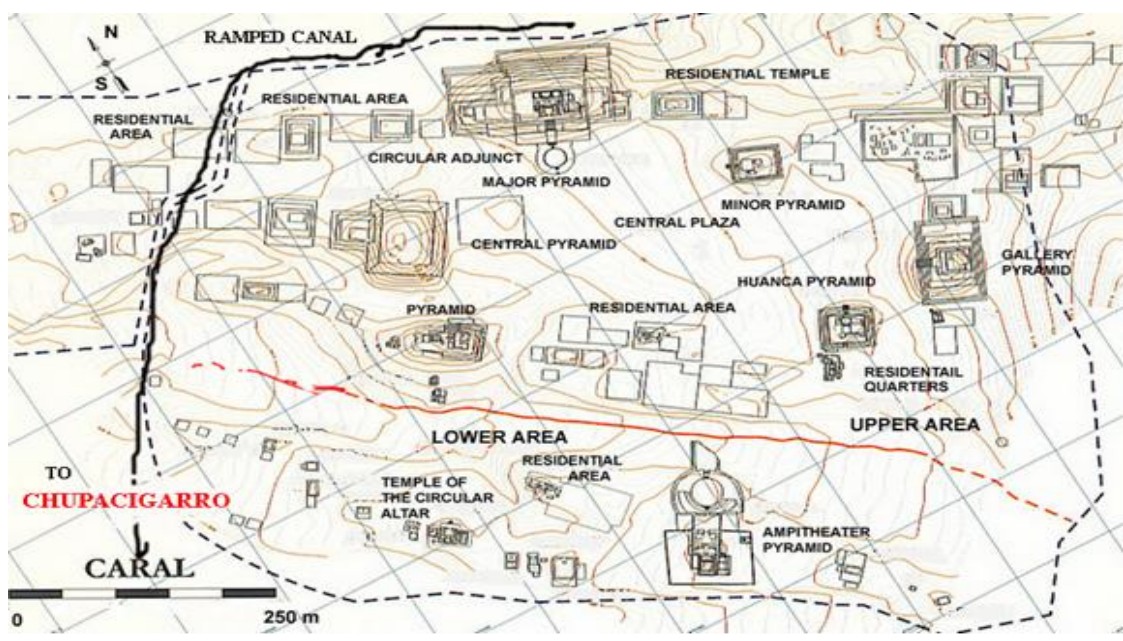

**Figure 23.** Details of the Caral urban center. Color version of figure derived from [2], p. 225.

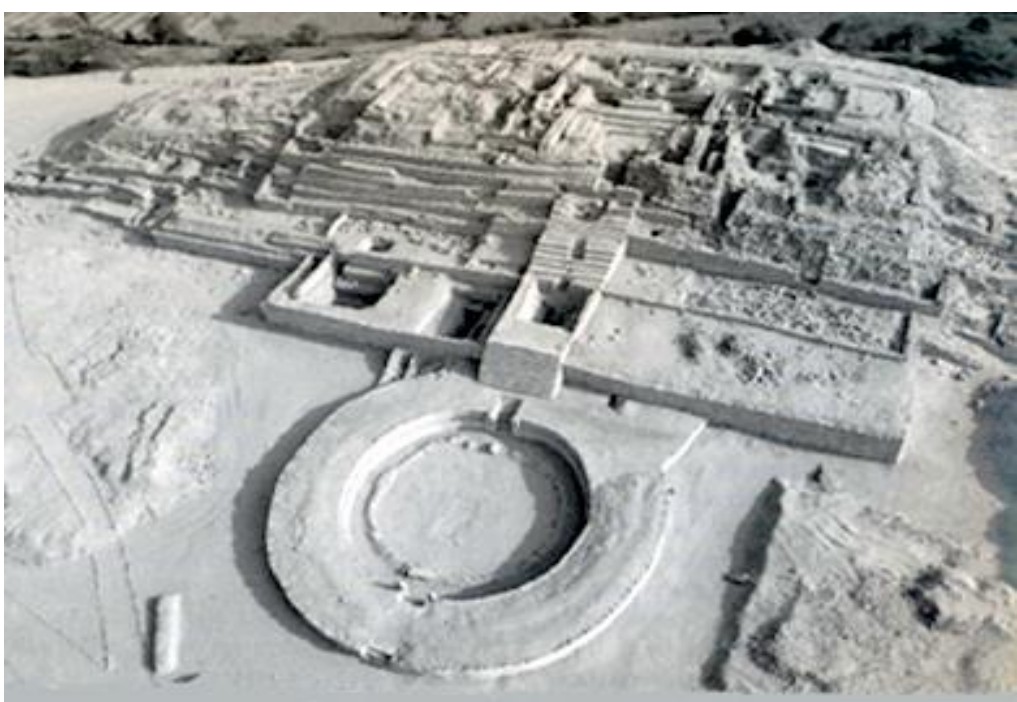

**Figure 24.** Details of the Major Pyramid; location indicated in Figure 23. Figure derived from [2], p. 226.

The present CFD studies are comprehensive in that they examine the sources of environmental landscape changes brought about by ENSO El Niño flood and drought effects that affect Caral's history. As area field observations encounter beach ridges, landscape erosion events, and soil deposition events that alter the agricultural, marine, and living quarter landscape upon which Caral society maintained its existence for over 1000 years, these effects derive from fluid dynamics effects that CFD analysis can bring forward for the first time to match field observations.

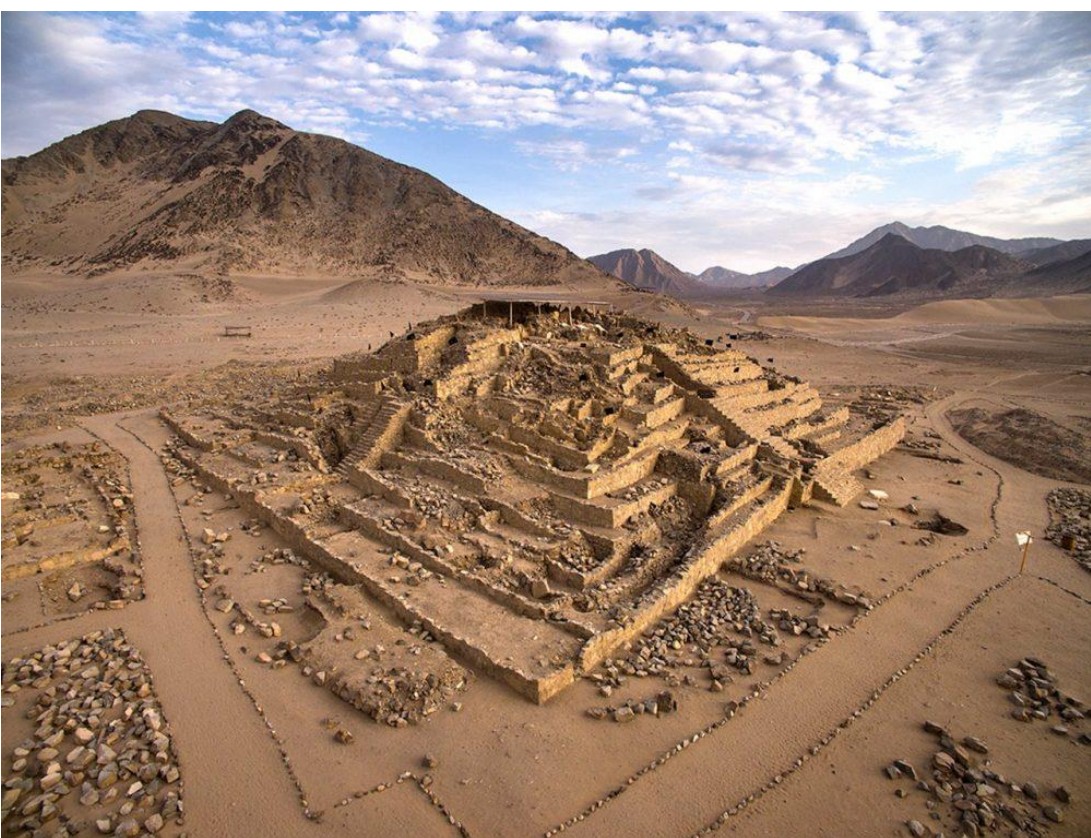

**Figure 25.** The excavated Gallery Pyramid; location indicated in Figure 23. Aerial photo by author.

For ancient (and modern) societies of north coast Peru, contention with natural forces derived from ENSO flood and extended drought disasters, as well as climate change effects that alter environmental conditions that affect societal continuity, play a key role in interpreting their historical development. The present chapter concentrates on environmental changes throughout Caral's thousand-year lifespan that influence both its historical development and its demise in later years.

Contemporary with Caral proper (Figure 23) are 18 other Supe Valley sites that, together with Caral as the capital city, form an early cooperative city state sharing the same cultural and religious values made possible through intrasite trade of agricultural and marine resource products. For example, coastal sites involved in harvesting marine and marsh resources (fish, shellfish) traded their protein-rich products with inland sites involved in agricultural product farming to provide the nutrition balance for all participating Supe Valley site members. Excavated pyramid structures shown in Figures 24 and 25 indicate many levels of occupation quarters of elite members of Caral society in addition to circular arena areas (~25 m in diameter) used for religious ceremony purposes. Apparently, even for the earliest of Peruvian societies (as yet discovered), emergence of a religion and an elite class to conduct rituals comes with the development of class difference structure. Figure 23 details the presence of residential areas of lesser structural detail than those associated with pyramid structures, indicating the presence of a worker class involved in site maintenance, agricultural activities, and intersite trade activity necessary for progress of all site occupants. Further details on Caral's history and archaeology ([3], pp. 31–68; [23–29]) are given in these references. Figure 24 is the excavated Major Pyramid indicated in Figure 23, indicating multiple rooms, platforms, and ceremonial open plazas for the elite class of Caral's population; similarly, Figure 25 indicates similar details for the Gallery Pyramid, whose location is given in Figure 23. Of note are fire pits within the Major Pyramid that have circular air passageways close to the pit's bottom; this convection feature is designed to draw in air to sustain a wood fire's hot, lower-density fumes as they rise upward. As

other pyramids share this similar feature, it may be surmised (or conjectured) that pit fires were used as a signaling feature to communicate information between distant pyramids. As Caral was deeply buried by aeolian sand deposits prior to Ruth Shady's excavations starting in the late 1980s, it is unclear whether the many rooms shown in Figures 24 and 25 originally had reed roofing. Given the adjoining circular ceremonial area for the Major Pyramid with an access stairway to the inner reaches of the pyramid, the site played a significant role in ceremonial activities for Caral's population. A similar, but smaller, circular ceremonial area that is associated with a temple at Chupacigarro (Figure 23) was found prior to Caral excavations.

Of interest to the site's long history is the agricultural system evolution over centuries to feed the growing population of the valley's many sites. As Peru is well known for its ENSO climate changes, particularly major El Niño flood and long-term drought effects, challenges to agricultural productivity arose over centuries of site occupation. Of note is the special water availability that the Supe Valley has compared to all other coastal valleys: from high levels of highland rainfall in the Andean Cordillera Blanca mountain range, freshwater lakes evolve. Water from these lakes proceeds down to valley lowlands through geologic faults to perpetually maintain valley ground water to with ~1 m of the valley bottomland ground surface, making multicropping agriculture possible at high productivity levels. This water transport feature, known as *amunas* in prehistory, was thought to occur from subterranean tunnels built by the ancients as informed by local valley farmers. Together with intermittent Supe River flow, the Supe Valley experiences sufficient water resources for multicropping on a year-round basis. Thus, the Supe Valley was a logical place for a complex society to initially develop given the valley's large coastal and interior valley bottomland areas and plentiful water resources that exist even to the present day. As climate challenges affect the agricultural base of a society, FLOW-3D CFD investigation of landscape changes adversely affecting agricultural production are next considered to tell the story of the collapse of Caral society after ~1000 years of successful occupation of the Supe Valley.

Figure 26 presents results from test probes to find a specific mollusk presence found only in coastal water depths of ~1 m. C14 dating of found mollusk shells then provides the shoreline position changes over long time periods (Tom Dillehay, personal communication). For example, Figure 26 indicates that a far inland coastline at Waypoint 49 accumulated rainfall washed-in soils carried by ocean currents, aeolian sand transfer, and surface soil transfer from rainfall runoff over the next ~1100 years to form a new coastline addition to Waypoint 48. Subsequently, over the next ~1600 years, the coastline eroded back to Waypoint 44 from many large-scale, mass-wasting El Niño events followed by shoreline additions up to present times, as Figure 26 indicates. This figure denotes the importance of landscape change over Caral's lifetime that influenced its agricultural land base as later discussion details. Observed at certain locations on the Peruvian north coast shoreline are inland sequences of linear beach ridges, one of which is shown in Figure 27. The origin and composition of beach ridges and landscape geometry change are next determined by a CFD coastal model (Figure 28) as these formations proceed from fluid dynamics effects.

The full-scale CFD model in Figure 28 includes the sloped coastal landscape incorporating the ~50 km distance between the Santa and Viru rivers. An El Niño flood-derived slurry mixture of soil particles, sand, pebbles, rocks, and boulders collected from adjacent hillside runoff proceeds down the two river valleys to interact with coastal and ocean currents; details of flow rates and slurry physical properties are given in [2] (pp. 48–50).

In Figure 29A,B, the scale denotes 1.94 slugs/ft$^3$ for water and deposition density values representative of the composition of the runoff slurry. Results shown are for a singular El Niño event. Note that northward ocean and offshore currents carry slurry mixtures northward to create lengthy subsea deposits, given the 50 km distance between the Santa and Viru valleys. With ongoing tectonic uplift and further El Niño slurry deposition events occurring over time, sequences of inland beach ridges form. Note that the heavier slurry particles (pebbles, rocks, boulders) appear to centrally deposit within lighter slurry

mixtures and, with later rain and flood events washing away lighter particles, form beach ridges mainly composed of heavier stones, as Figure 27 indicates.

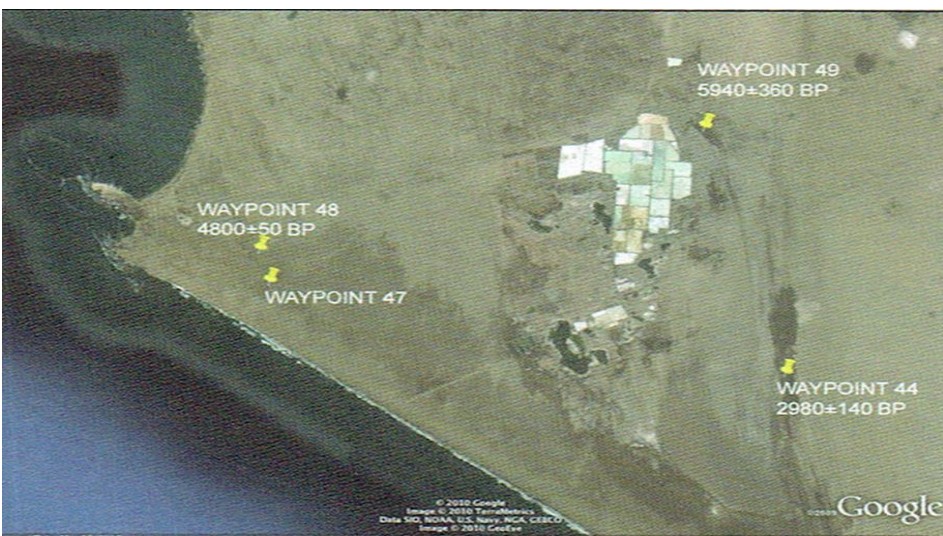

**Figure 26.** Coastal area landscape changes over time. Color version of figure derived from [2], p. 228.

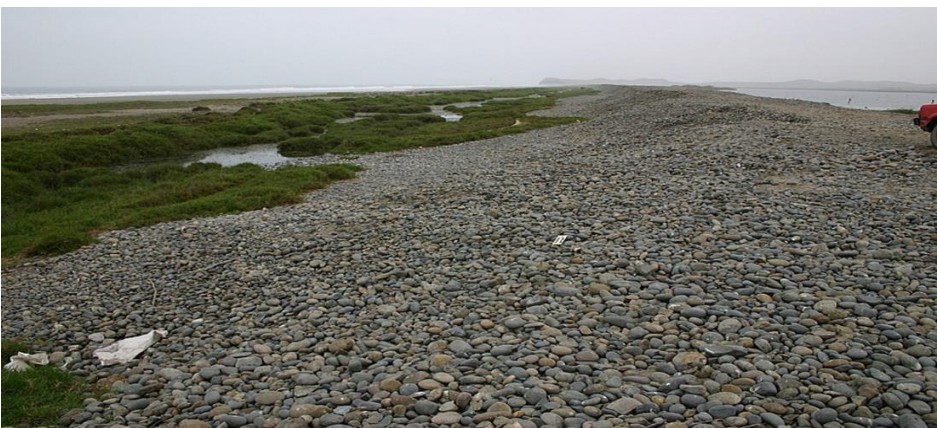

**Figure 27.** Coastal beach ridge formation on north coast of Peru formed from a major El Niño flood event. Photo by author at Supe Valley coast in Peru.

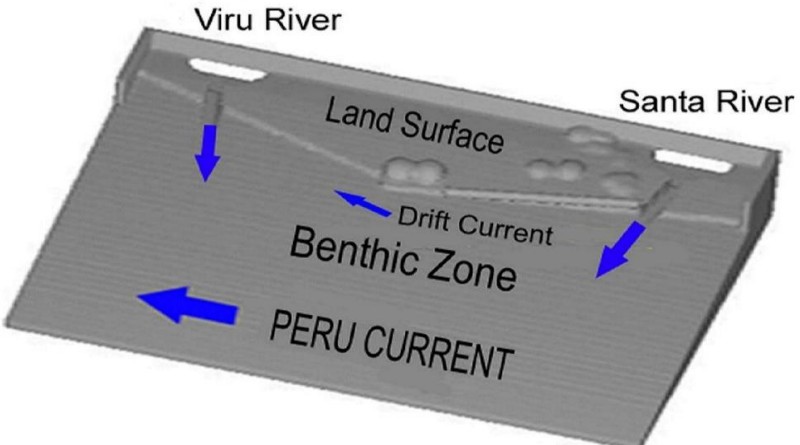

**Figure 28.** FLOW-3D CFD coastline model from the Santa to Viru River Valleys. FLOW-3D CFD model by author [2], p. 243.

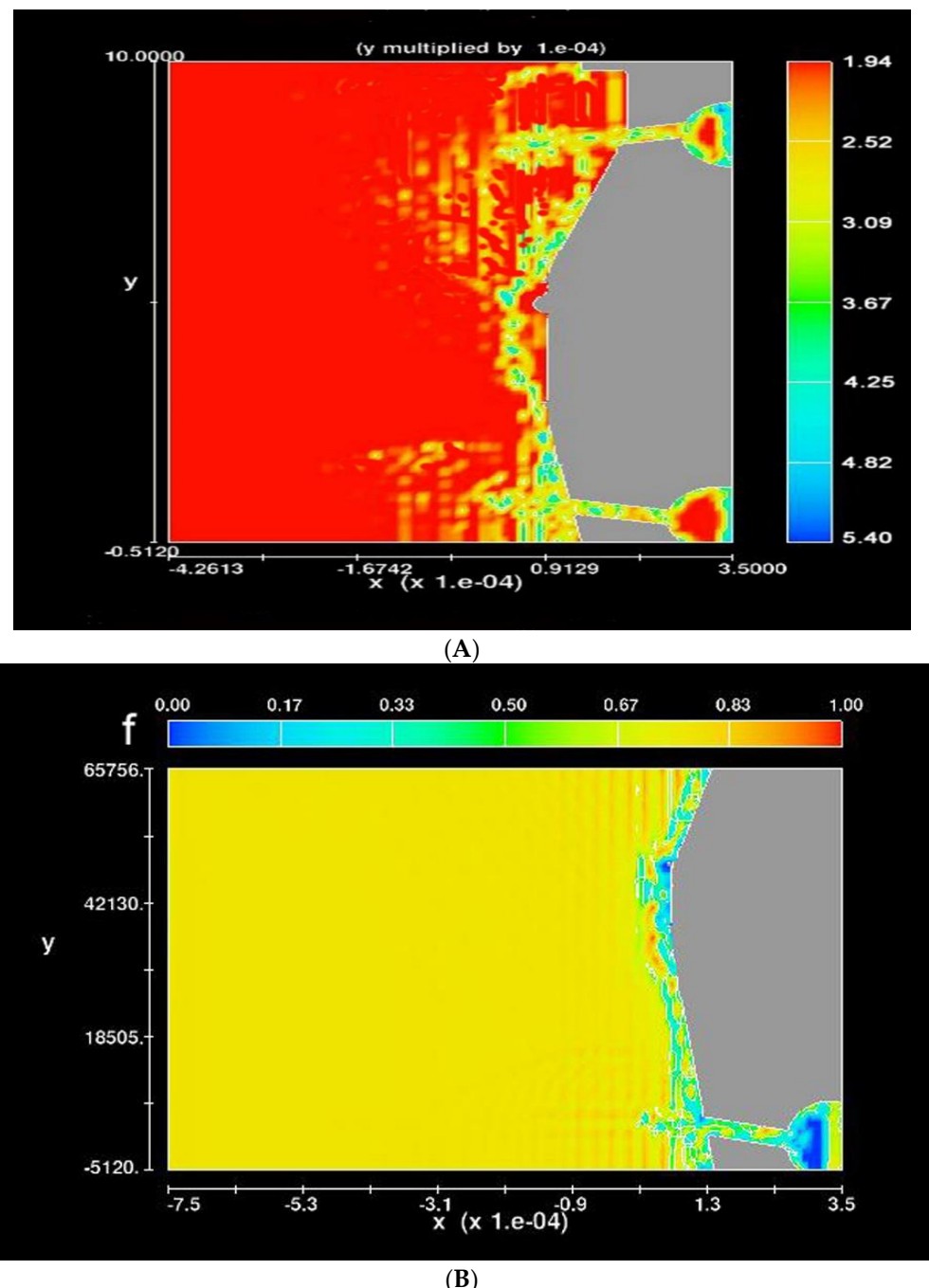

**Figure 29.** (**A**) Offshore slurry deposition density from an El Niño flood event from CFD calculations for a low offshore benthic slope. Scale density in slugs/ft³ (1 slug/ft³ = 515.38 kg/m³) representative of slurry mixtures with high rock content. Result from [2], p. 246. (**B**) Offshore slurry deposition fluid fraction density from an El Niño flood event from CFD calculations for steep benthic slope typical of the Supe Valley offshore region. Result from [2], p. 248 (1 slug/ft³ = 515.38 kg/m³).

Figure 29A,B show calculated slurry deposition close to the shoreline for an offshore shallow benthic slope (Figure 29A) and for a steep benthic slope typical of the Supe Valley coastal area (Figure 29B). The steep offshore benthic slope case ultimately leads to a more concentrated linear beach ridge deposition typical of Figure 27. With coastal tectonic uplift and coastal landscape additions, earlier stranded beach ridges are found sequentially located inland from the coastline. Figure 30 shows a previously settled offshore slurry ridge (gray area) subject to a further El Niño event. From the offshore blockage ridge formed

from the earlier slurry deposition event, a further flood event can create a marsh and/or an infilled area behind the ridge, as Figures 30–32 indicates. Surveys of several well-preserved coastal areas indicate a sequence of beach ridges; these ridges, C14-dated from mollusk shell slurry debris, date major El Niño flood events that influenced the landscape and often the survival history of archaeological sites in north coast areas. With respect to coastline shape changes from geophysical events, as shown in Figure 26, Figure 31 indicates bay infilling from a major El Niño event that deposits slurry in a coastal concave bay area subject to vortical water motion that traps silt-laden ocean currents. Such events, as shown in Figures 30 and 31, lead to the Figure 26 coastline shape changes.

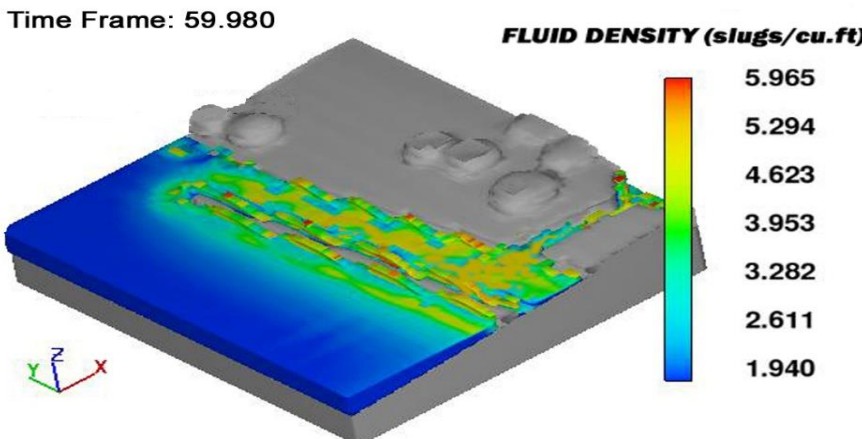

**Figure 30.** FLOW-3D CFD deposition results of slurry contents in Pacific Ocean coastal area between the Santa and Viru valleys. Result from [2], p. 248. (1 slug/ft$^3$ = 515.38 kg/m$^3$).

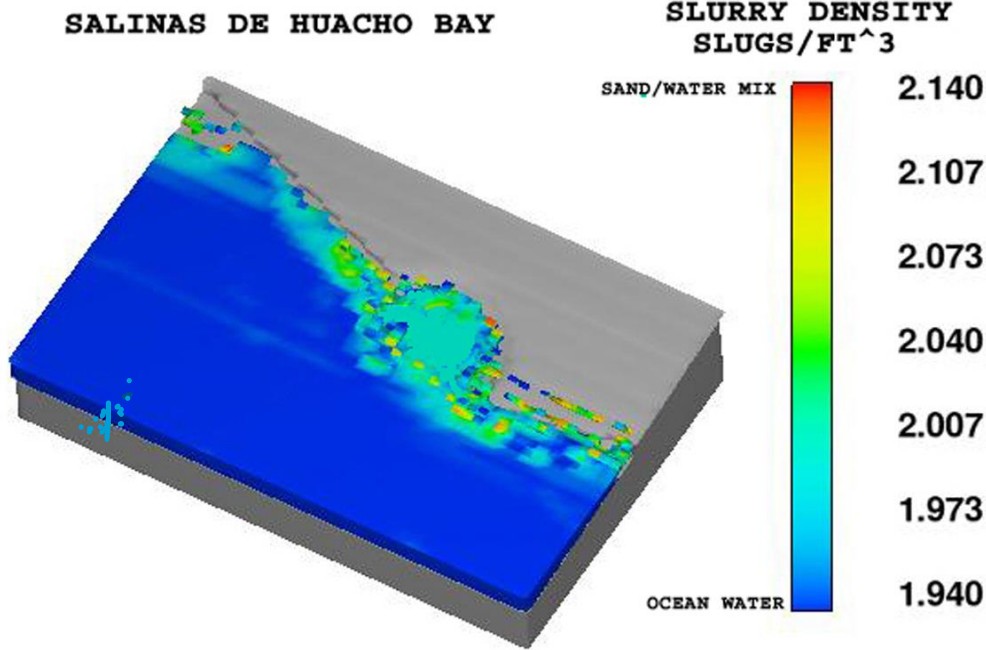

**Figure 31.** FLOW-3D CFD prediction of a coastline change that proceeds from a major El Niño event that traps silt laden ocean water to deposit in bay areas. Result from [2], p. 249. (1 slug/ft$^3$ = 515.38 kg/m$^3$).

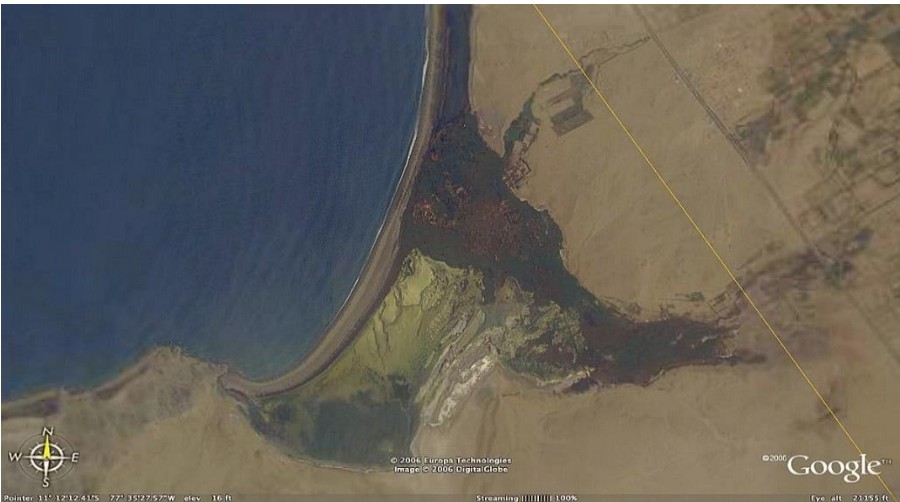

**Figure 32.** Extensive beach ridge formation in the Supe Valley coastline area. Existing segment of the Medio Mundo beach ridge shown. Result from Google Earth.

To this point, geophysical landscape changes originating from major El Niño flood events were demonstrated by CFD calculations to qualitatively duplicate nature's productions. The question of the extinction of Caral in the ~1600 BCE time frame is next addressed by geophysical changes that affected Caral and its satellite settlements' agricultural and marine resource base.

The presence of a major beach ridge along the northern Peruvian coastline created in the late 11th–12th century BCE (the Medio Mundo Ridge, Figure 32) originates from a major El Niño flood event and affects the continuity of Supe Valley sites. One major effect was the diminishment of the offshore fishing and shellfish gathering marine resource base due to bay infilling, as Figure 31 indicates. A further landscape change noted from this major beach ridge presence was blockage of Supe River drainage resulting in sediment deposits behind the beach ridge and marsh creation unsuitable for agriculture (Figure 32). These consequences led to a contraction of the near-coastal farming areas (Figure 33) now replaced by marsh areas unsuitable for agriculture. Given the large extent of the Medio Mundo beach ridge (~90 km), coastal infilling, as shown in Figures 30 and 31, similarly affected the Supe Valley's agricultural and marine resource base supporting Caral's settlements.

A further Supe Valley landscape transformation involved northwesterly aeolian sand transfer from beach-ridge-deposited sands in exposed beach flat areas (Figures 30, 31 and 34) inundating previously established agricultural and settlement areas. The net result over time was contraction of the coastal agricultural base to narrow inland bottomland valley areas (Figure 33) and reduction of easily netted small fish varieties and mollusk marine resources necessary to maintain the large Supe Valley population at previous levels.

With the reduction of farming areas and the marine resource base from the Medio Mundo event, society population levels could not be sustained at earlier levels. Of further note is that valley bottomlands are flat, resulting in Supe River course meander during intermittent rainfall flows. During a major El Niño event, shallow fertile farming soils are washed away, requiring many years before mountain soil runoff redeposits fertile soils suitable for agriculture. A conversation with a local farmer reveals that during the 1989 El Niño major flood event, some 50 acres of his farmland was washed away; such events occurred in Caral's historic times requiring contraction of valley agriculture to limited plateau areas, thus limiting the valley's ability to sustain large valley populations. The totality of environmental changes, as demonstrated by CFD investigation, thus precipitated contraction of Supe Valley settlements as sources of food supply declined to support the large valley population; this course of events in the ~1500 BCE time period ultimately led to the abandonment of all Supe Valley sites.

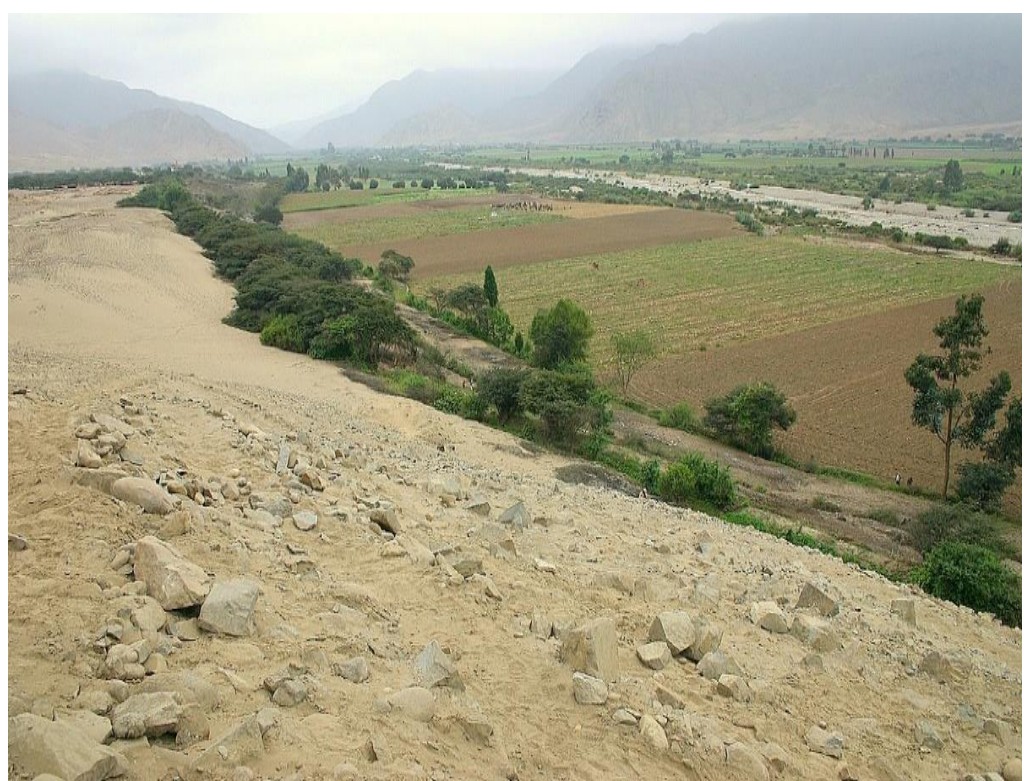

**Figure 33.** Present-day Supe Valley bottomland agricultural area; Caral situated on left elevated plateau. Photo by author.

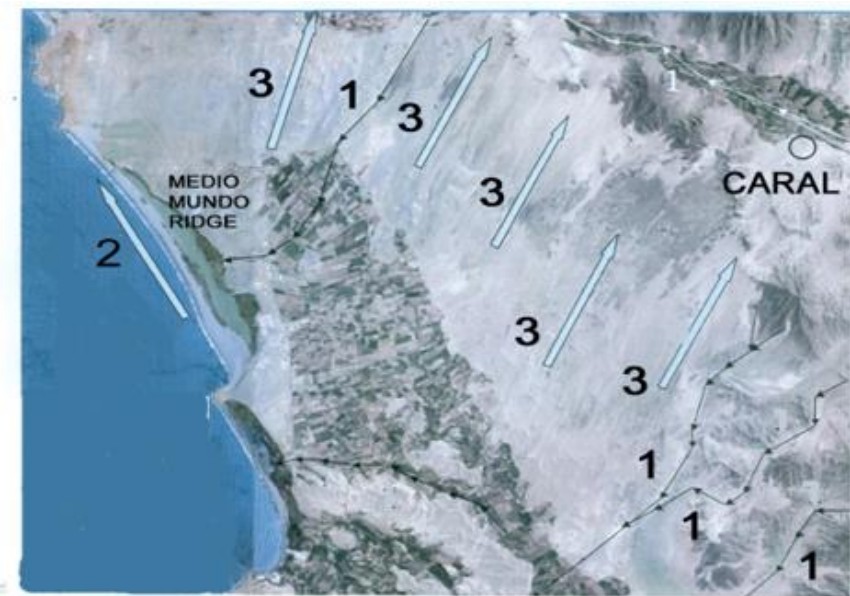

**Figure 34.** Major environmental changes resulting from the Medio Mundo beach ridge. Derived figure from Google Earth.

## 7. FLOW-3D CFD Investigation of the Pont Du Gard Aqueduct Design (Nîmes, France) and Roman Water Engineering Technology

The first century CE Pont du Gard Aqueduct bridge (Figure 35) and the water distribution castellum (Figure 36) located in southern France (Figure 37) has received wide attention by researchers over the years, focusing on its history, construction, and water engineering technology. A detailed summary of this research is given in ([2], pp. 295–318).

As little descriptive material about the water engineering necessary for the construction and operation of the aqueduct system survives from Roman times, use of CFD analysis provides new insights into the depth of Roman water engineering that underlies the totality of the water system's design.

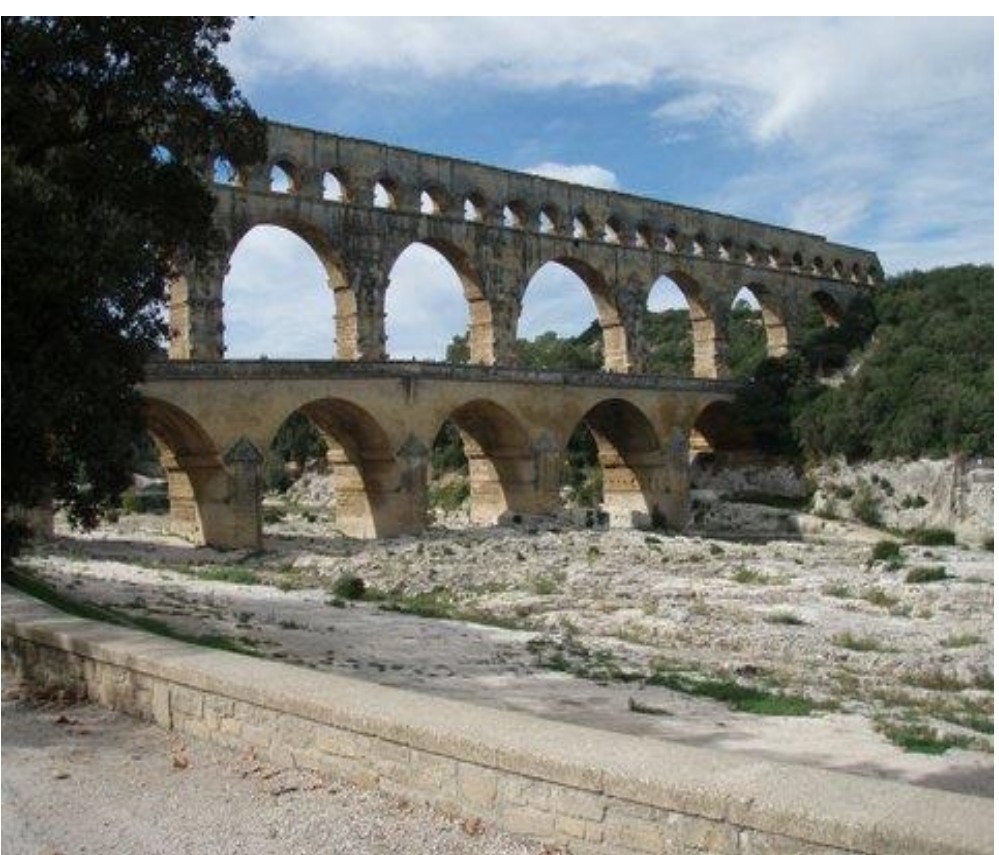

**Figure 35.** The Roman Pont du Gard Aqueduct Bridge. Photo by author.

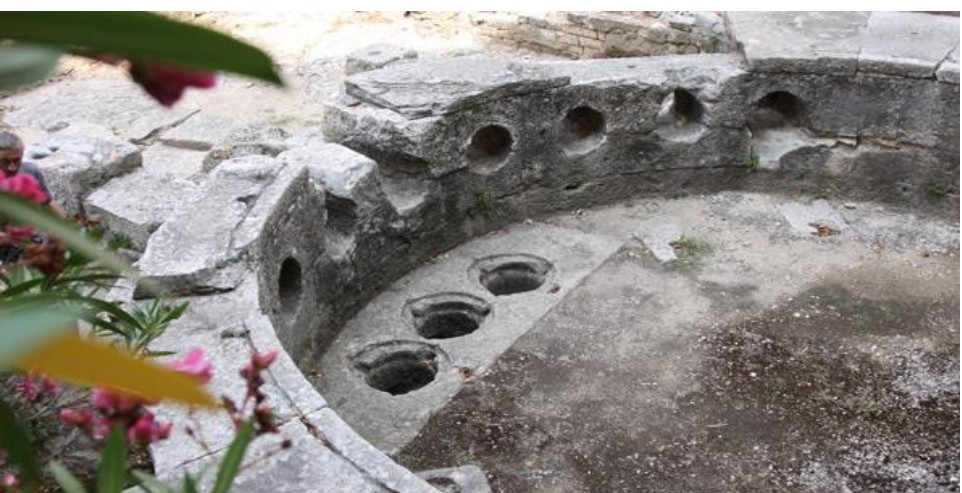

**Figure 36.** Interior view of the Pont du Gard water distribution castellum. Photo by author.

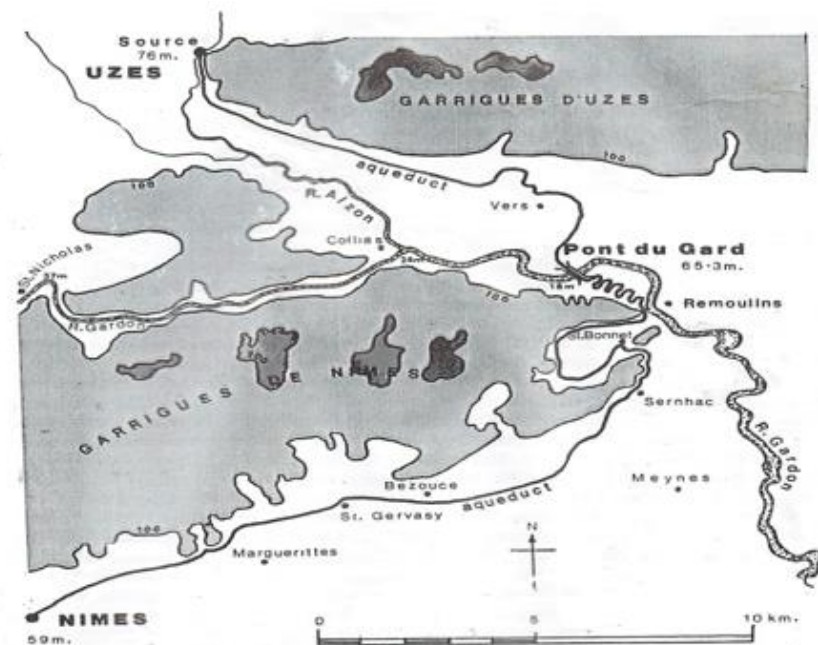

**Figure 37.** The aqueduct path from the spring source at Uzes to the water distribution castellum, Figure 36, within the city of Nîmes, France (Roman Nemausus). Figure derived from [30].

Of interest to determine Roman water engineering capabilities are several main design factors: (1) the 0.002 degree channel slope leading aqueduct water to the castellum (Figure 38); (2) the seven castellum wall exit ports and the three castellum floor ports (Figures 36 and 39); (3) details of the castellum wall's ~1.5 m low height (Figures 36 and 39); and (4) the geometric details of the castellum water entrance from the aqueduct (Figure 38) that had a vertically movable sluice plate (now lost) to control castellum water height.

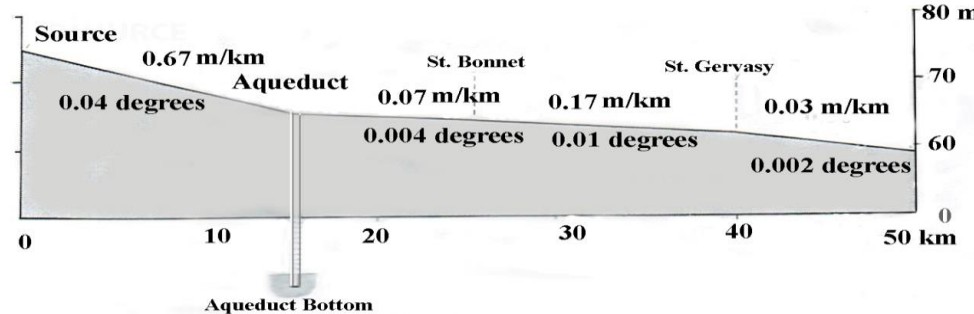

**Figure 38.** Aqueduct slope measurements from the Uzes spring source to the castellum. Figure derived from [30].

The goal of all water engineers, in past and present centuries, is to design a water structure that minimizes cost and construction labor while maximizing water transfer efficiency with aesthetics typical of all things Roman. Technical details of Roman water engineering used in the Pont du Gard channel and castellum not previously reported in the archaeological literature are given in ([2], pp. 295–318) and summarized in the discussion to follow.

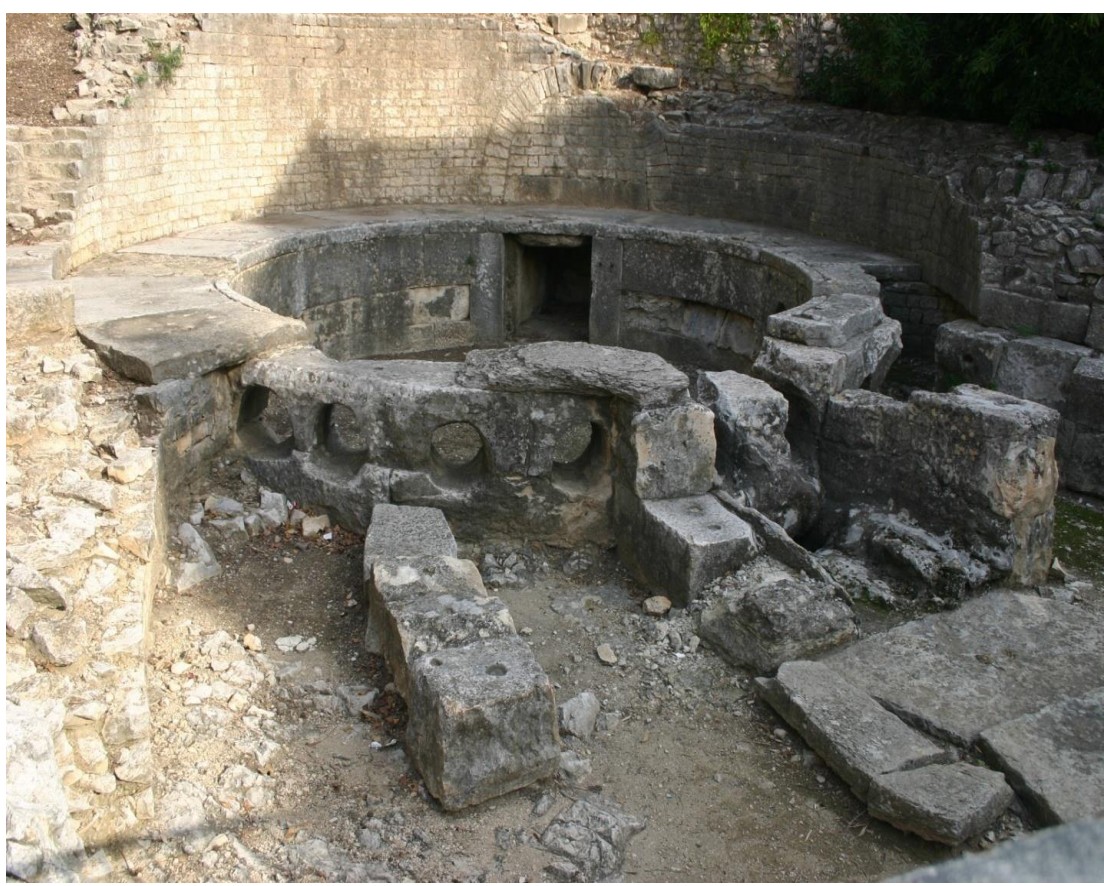

**Figure 39.** The Pont du Gard castellum showing the back wall rectangular water delivery port from the aqueduct and several of the castellum front wall water exit ports to pipelines leading to city destinations. Photo by author.

From Figure 40, initial pipeline sections exiting from the castellum wall ports (Figures 36 and 39) are nearly level to facilitate junction with the castellum wall ports; downstream attached pipelines to city destinations are sloped to match the city street slope. Flow in these sloped downstream pipelines must be designed to support partial flow with air over the water surface to reduce pressurized, full flow leakage at pipeline joints (Figure 40C) such as would occur in a near-level pipeline A design. For Figure 40B, at a steeper prescribed declination angle than the landscape slope, supercritical (Fr > 1) flow exists, and hydraulic jump creation is possible as the pipeline inner wall surface is exposed to pipeline roughness over long lengths. Post the hydraulic jump, full flow exists, inducing pipeline joint leakage. Figure 40C shows the desired flow type within the sloped castellum pipelines joined to short, level castellum wall openings. These pipelines are set at a critical angle $\theta_c$ matching the landscape slope. This condition, called critical flow, induces an airspace over the partial flow, reducing pipeline joint leakage possibilities while producing the maximum flow rate that individual pipelines transport [19–21]. The key to the Roman design is to induce Fr = 1 critical flow in downstream sloped pipelines (Figure 40C) where the $\theta_c$ pipeline slope matches the landscape declination slope. This is carried out by selecting the height of the castellum from the landscape ground base height; this requirement determines the incoming aqueduct low 0.002 degree slope, as noted in Figure 38. From Figures 36 and 39, the pipeline exit ports are extremely close to the top of the castellum-containing wall. This design implies that if entry flow into pipelines is at $y_c$ critical height ([20], p. 51; [2], p. 310), then critical, partial flow enters the sloped pipelines that are designed to continue critical flow at the $\theta_c$ declination angle (Figure 41). This Roman design explains the low wall height of the castellum (Figure 36) as the entry

flow height from the aqueduct only reaches up to half of the exit pipelines' diameter and this is guaranteed by the sluice gate open height.

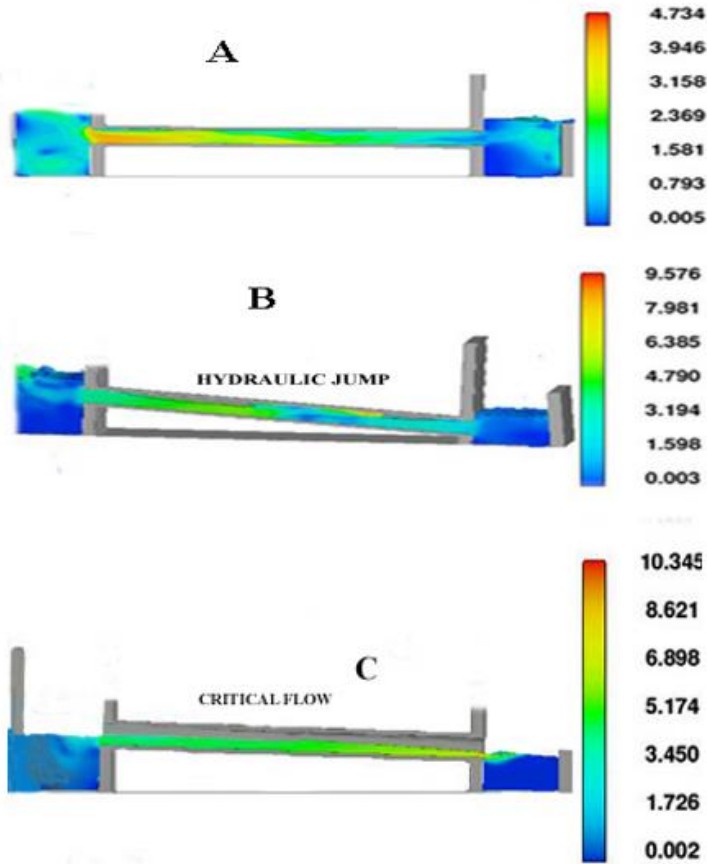

**Figure 40.** CFD pipeline flow characteristics for pipelines at different declination angles. Velocity is in ft/s. FLOW-3D CFD results from [2], p. 310.

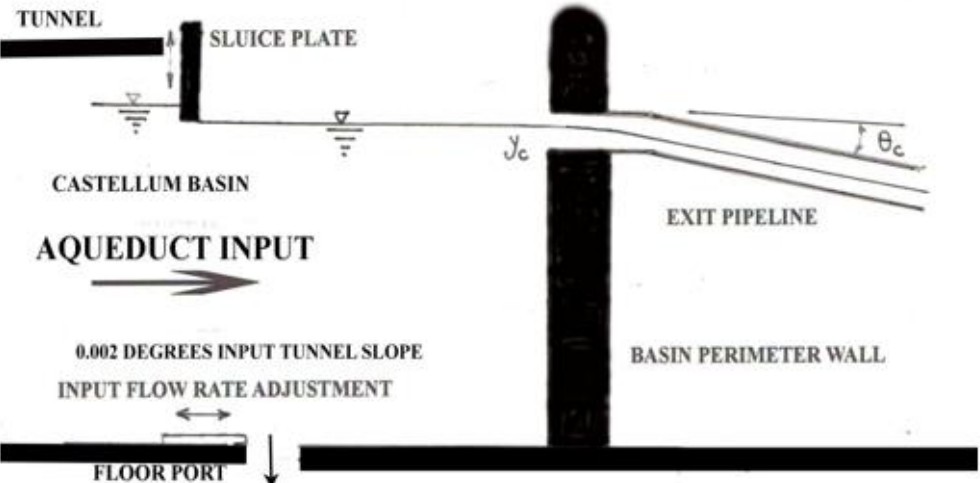

**Figure 41.** The Roman design of the Pont du Gard castellum (not to scale). Results from [2], p. 310.

As the input spring flow in the aqueduct may seasonally vary from the design flow rate, the movable sluice plate (Figure 41) can be set to ensure that entry flow depth into pipelines is at the $y_c$ depth; this design flow rate condition is met by adjustable openings in the castellum floor (Figure 36) that drain away excess water to maintain the design flow rate. From Figure 38, it is noted that the aqueduct channel slope entering the castellum

port (Figure 39) is practically level (0.002°). The reason for this near-level slope section (beyond raising water height, lowering entry velocity to the castellum, and determining the height of the castellum above the ground) is to guarantee that the low height of the castellum walls safely contains the input flow and that exit pipelines can be set at $\theta_c$ which is the slope of the Nîmes street carrying the pipelines to downhill city destinations.

Figure 41 summarizes the castellum design originated by Roman water engineers. Again, the vertical setting of the sluice plate is set to maintain the $y_c$ water depth in the castellum; this is designed to ensure critical (Fr = 1) flow into the $\theta_c$ angled sloped exit pipelines for design flow rate condition. This design limits spillage from the castellum wall top surface by means of excess water removal past the design flow rate by castellum floor ports that have movable baffles to adjust the flow to the design flow rate. A further benefit to this design is to maintain the aesthetics of the castellum basin water surface—a viewer would see a calm and stable water surface worthy of the elaborate structure that once enveloped the castellum ~2000 years ago.

Given that the Roman castellum design can be understood in terms of modern hydraulic terminology, counterparts to this knowledge must exist in Roman times, albeit in terminology or documents yet to be discovered. Given Roman expertise in water science, as evidenced by many aqueducts, fountains, public bath structures, and port structures in Roman cities, water technology that clearly originated from experiments and observations over many years enabled the sophistication of the Pont du Gard castellum.

## 8. FLOW-3D CFD Analysis of the Moat Structure Hydraulic Engineering at 600–1100 CE Tiwanaku in Bolivia

Located in the interior of the ancient city of Tiwanaku is the prominent Gate of the Sun (Figure 42), whose iconography offers an introduction to learning the depth of spiritual beliefs that influenced the citizens of that city, the structure of their society, and the architectural patterns of ceremonial compounds that gave their priests the means to communicate with their deities. Commensurate with religious aspects of Tiwanaku society is the practical side involving urban and agricultural water supply and distribution systems to support the large city population. Here, accomplishments in the water sciences for urban and agricultural use are vital to the development and continuity of the Tiwanaku society. To this end, details of new hydraulic engineering discoveries as applied to Tiwanaku's urban and agricultural systems ([2], pp. 1–30; [31–36]) are summarized in the present chapter from CFD analysis and modern hydraulic engineering studies of their water supply and distribution systems.

The UNESCO World Heritage site of 600–1100 CE Tiwanaku located in altiplano Bolivia was the subject of early archaeological investigations [37,38] to determine its role in Andean history. Of note to these scholars was the investigation of an encompassing "moat" that surrounded the elite quarters of the city which harbored the Acapana multistoried pyramid, the Calasasyaya royalty compound, the subterranean temple noted for its central stele of a prominent deity as well as its many wall-mounted deity heads, and other administrative and royal compounds indicated in Figure 43. Later excavations within the moat area revealed an intricate drainage system consisting of dual subterranean pipelines connected to surface structures, indicating the presence of advanced water engineering not previously noted or analyzed. Later investigations [32,33,39,40] brought forward the argument that the moat had a previously unanticipated use beyond being a separation boundary between religious and administrative elite quarters and the secular, working class housing of the city population.

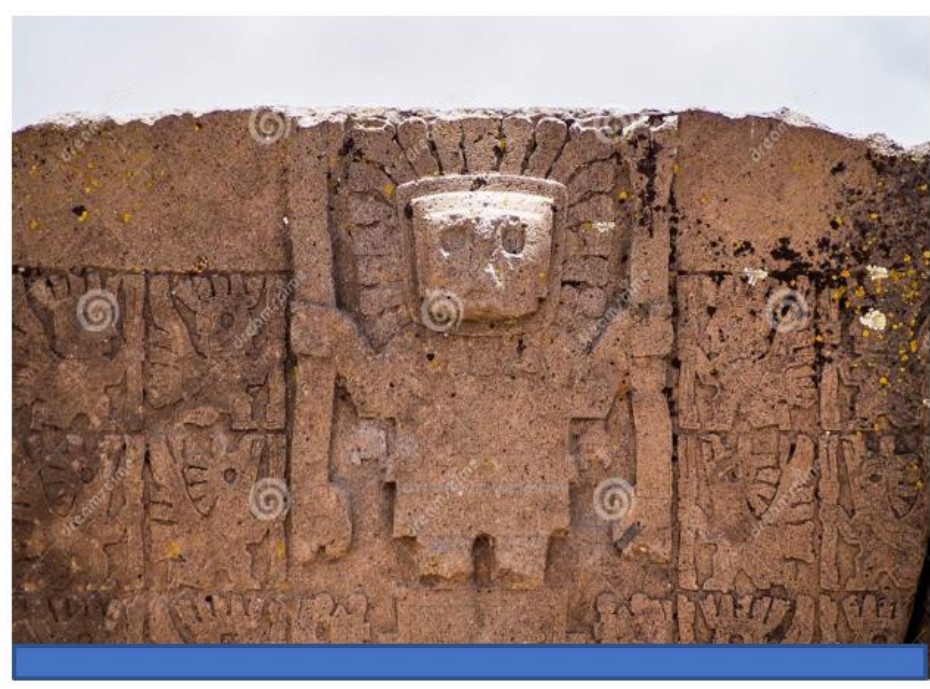

**Figure 42.** Gateway of the Sun deity iconography; site located in central urban Tiwanaku. Photo by author.

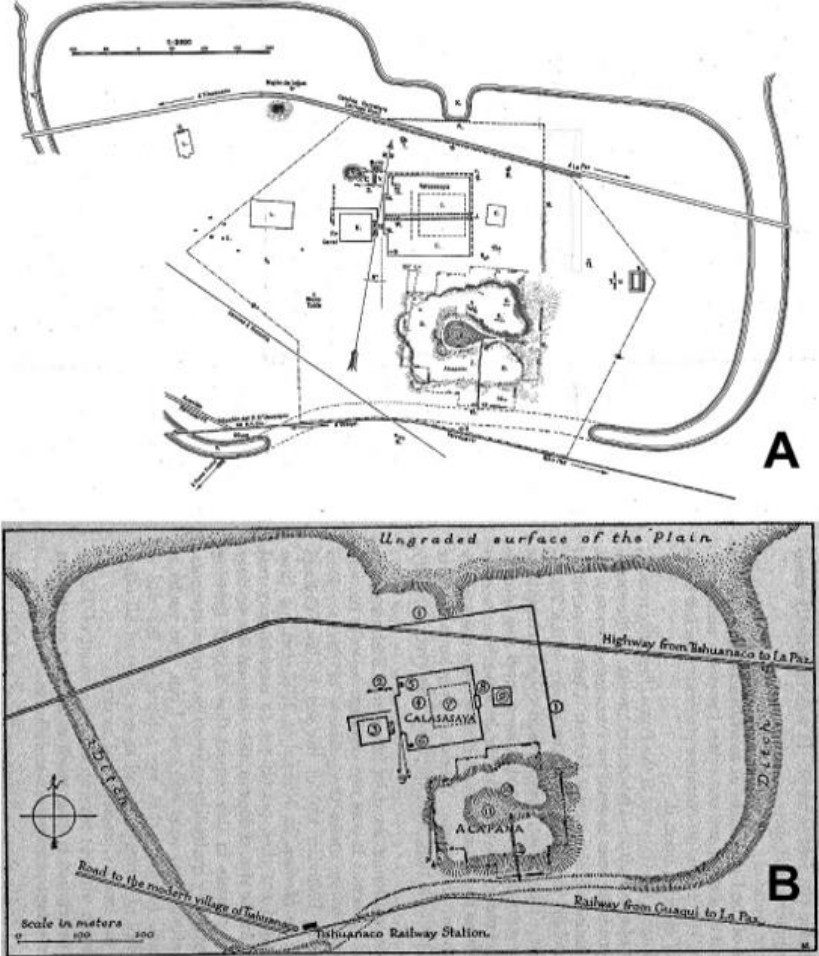

**Figure 43.** Early ground drawings of the moat structure by Bandelier 1911 (**A**) and Bennett 1934 (**B**) References [37,38].

Recent discovery of site aerial site photographs revealed further details related to the moat and its connection to numerous spring-supplied channels originating from nearby mountain areas. Many of these lengthy channels are connected to the moat. For the moat, estimated surface east–west length dimensions (Figure 43) range from 700 to 1000 m, and the surface north–south width dimensions are ~400 m. The moat depth is sufficient to penetrate the groundwater level in the dry season, estimated to be on the order of ~5 m [39].

The hydrodynamic engineering role played by the moat can be investigated through use of a CFD model that includes all of the discovered geometric features of the moat and their relation to the ceremonial structures (Figure 44) within the moat boundary.

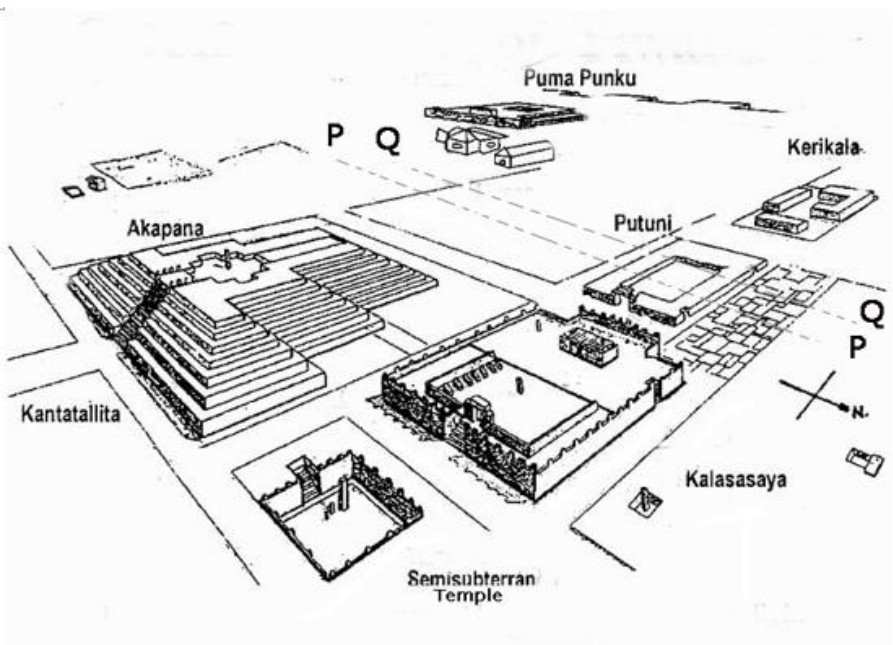

**Figure 44.** Details of elite and administrative structures within the moat and subterranean P, Q pipelines (from [2]).

One recently discovered feature of the moat is the Mollo Contu channel intersecting the southern portion of the moat (Figures 45 and 46) that plays a central role in interpreting the purpose of the moat. The Mollo Kontu canal carries water from a reservoir at the base of the southeastern mountain range that remains at full capacity from rainy season runoff from mountain gullies as well as water transport from nearby springs. Ground traces of this channel are apparent from the aerial photographs together with other features that include many additional channels, inner-city field systems (*cochas*), flood plains, and major (then unexcavated) architectural features. Of note are subtle moat geometry differences between earlier investigator's versions (Figure 43) and the later aerial photograph versions (Figures 45 and 46) likely due to soil transfers from yearly heavy rainfalls as well as human activity on the site. For the CFD model of this area, the preference is for use of the earlier Figure 43 versions as they represent an earlier undisturbed view of the site.

The CFD (Figure 47) model incorporates an aquifer below the ground surface with specified hydraulic conductivity, porosity, specific surface, and particle size variables included in the model aquifer description; water viscosity and density, as well as aquifer properties, are specified [22]. Shown are two subterranean stone-lined channels (P and Q, Figure 47) originating from moat arm W to arm V then continuing underground to empty in the Tiwanaku river A′–B′. The Mollo Kontu channel U empties into the W moat upper arm. The CFD model is tilted 1 to 2 degrees downward in the south to north direction as well as the east–west direction to facilitate water drainage into the Tiwanaku River that empties into Lake Titicaca. The P and Q channels are composed of ~0.75 m stone plates on all sides with a top plate to seal channel (Figure 48). Although only two subterranean channels, P

and Q, are shown, indications of other subterranean channels exist to the west of those shown whose presence remains to be verified by excavation. A series of perforated stone disc plates lead vertically upward from circular openings on the subterranean channel's top plates to connect surface structure drainage to the P, Q channels to allow wastewater drainage from high status buildings. Subsidiary water input from Corocoru spring-fed channels N and O serve wetland agriculture in the K region, with excess water emptying into moat channel D at point a (Figure 47).

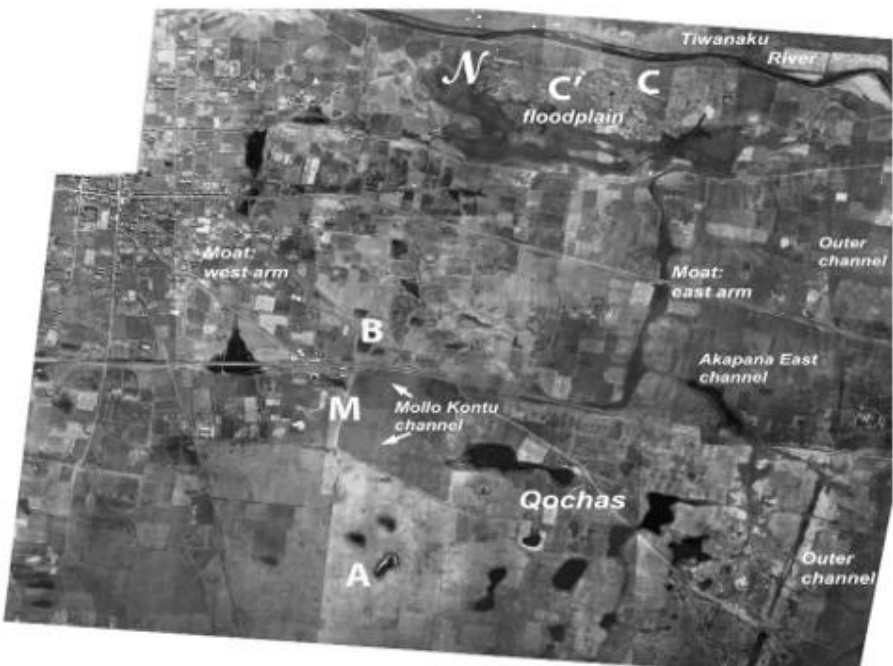

**Figure 45.** The recently discovered Mollo Contu canal. Figures originally derived from 1940s aerial photos from Alan Sawyer at Tiwanaku and later annotated 2010 by John Janusek.

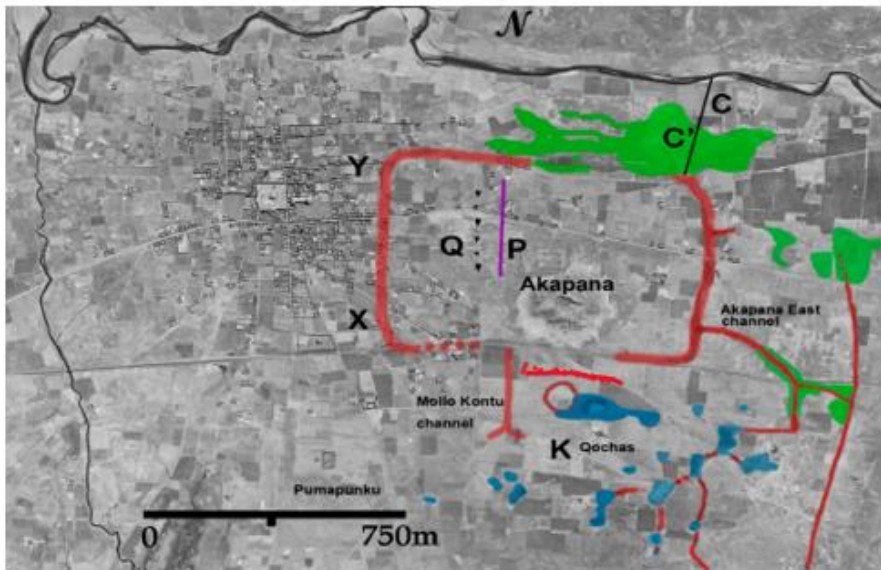

**Figure 46.** The recently discovered Mollo Kontu canal. Figures originally derived from 1940s aerial photos from Alan Sawyer at Tiwanaku and later annotated 2010 by John Janusek, Vanderbilt University.

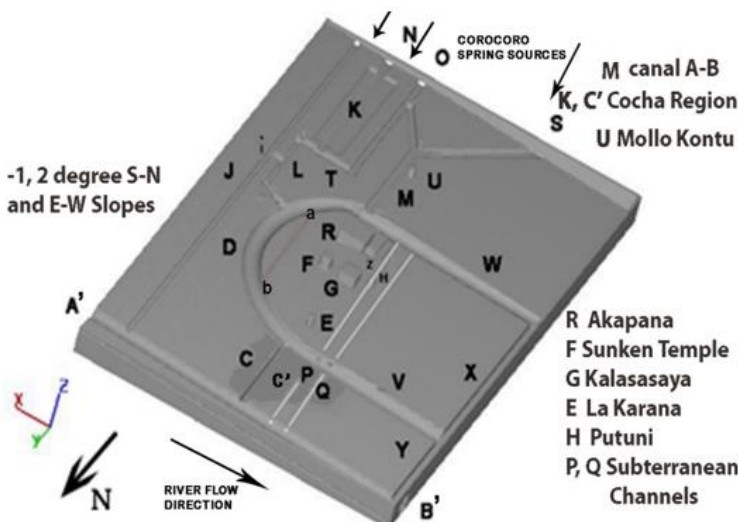

**Figure 47.** FLOW-3D CFD model of features in the moat location area, from [2], p. 173.

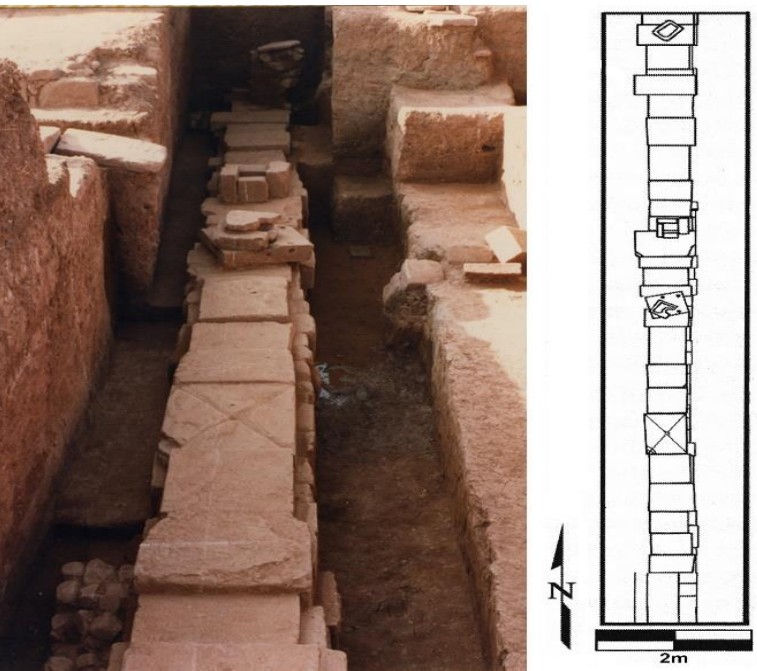

**Figure 48.** Excavated subterranean channel P located ~3 m below the ground surface. Photo by author.

To examine the function of the moat in wet and dry seasons, FLOW-3D CFD [15] calculations were performed to indicate the moat's purpose. Figure 49 results concern the post rainy season at a time well into the dry season. The fluid fraction (ff) detail from FLOW-3D calculations of a section of the moat east arm (D) shows the dry (blue) ground surface within the largely paved and roofed elite structure area inside the moat boundary that promotes rainfall runoff to keep the ground surface dry. The red ground area has absorbed heavy rainfall leading to a saturated ground surface now showing signs of evaporation and aquifer drainage water loss. Figure 49 shows the moat's inner surface walls conducting seepage water to the saturated bottom of the moat (dug below the water table) and the start of surface drying from the rainy season saturated (ff = 1) surface. The aquifer seepage plus surface evaporation thus helps to dry housing areas interior and exterior to the moat, inducing hygienic benefits by control of mold in housing structures. As the moat depth penetrates the saturated groundwater level, water from aquifer seepage cannot penetrate the moat bottom and thus transfers water to moat arms D and V for

drainage into the Tiwanaku River. Note that ff = 1 denotes aquifer and ground surface saturation, ff = 0 denotes dry surfaces, and intermediate ff values denote moisture levels in the aquifer. Drainage water then flows downslope to marsh area C' used for specialty crops to continue seepage drainage into the Tiwanaku River A', B'.

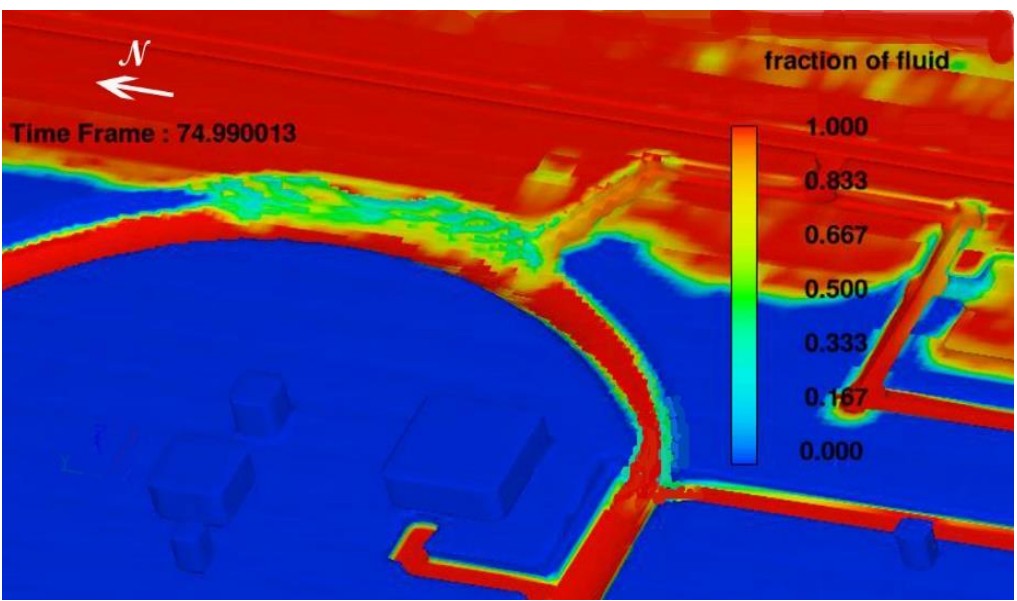

**Figure 49.** End of rainy season and well into the dry season fluid fraction distribution indicating aquifer drainage into the moat; from [2], p. 179.

An elevated channel C conducts excessive volumes of seepage water and rainy season surface runoff directly into the Tiwanaku River. Seepage maximums occur from saturated land areas adjacent to the moat at D. Continuous surface channel flow into the moat from the Mollo Kontu canal (M, Figure 47) and canal (a) draining excess irrigation water from (*cocha*) pasturage area K contribute further water to the moat for drainage to the Tiwanaku River. The following set of figures details the hydraulic engineering function of the moat through dry and wet seasons.

The Mollo Kontu M channel continues its flow from spring-fed mountainside reservoirs during both dry and wet seasons. The subsidiary channel L (Figure 47) drains water from pasturage area K which is supplied from channels N and O to add additional drainage water into moat arm W. Note also that straight channel J supplied from mountainside springs has a canal branch diversion channel (i) into channel L (Figure 47) to add further drainage water into the moat's D and V arms. Seepage water into the westernmost moat X channel appears to drain directly into the Tiwanaku River by a direct channel.

Figure 50 indicates fluid fraction results into the dry season with surface drying achieved by aquifer seepage and surface evaporation. Note that the Mollo Konto channel M (Figure 47) continues to provide water to subterranean P and Q channels as they originate from the northern moat wall surface W arm (Figure 51) continually supplied by the Mollo Kontu canal. Aquifer leakage continues providing water to the moat bottom to enhance drying together with surface evaporation drying. The continuous ground surface drying, as the dry season progresses, is amplified from high seepage levels continuing into later dry season times in the D region of the moat, as indicated in Figures 49 and 50. Figure 51 results are given on a depth plane of the P and Q subterranean channels and indicate the saturated (red) aquifer in regions east of moat branch D while indicating dry regions west of D. Figure 51 shows water flowing through P and C channels from the Mollo Konto source M channel. Further water input into the moat occurs from surface runoff from the elite structures within the moat (Figure 44) and occurs as areas within the moat boundary were largely paved with roofed structures, causing rainfall to runoff directly into moat arms

W–D–V–Z for drainage into A', B' to Lake Titicaca. Note the (red) saturated groundwater aquifer at depth outside the moat and the dry (blue) area at depth within the moat boundary due to extensive runoff from paved areas within the moat boundary.

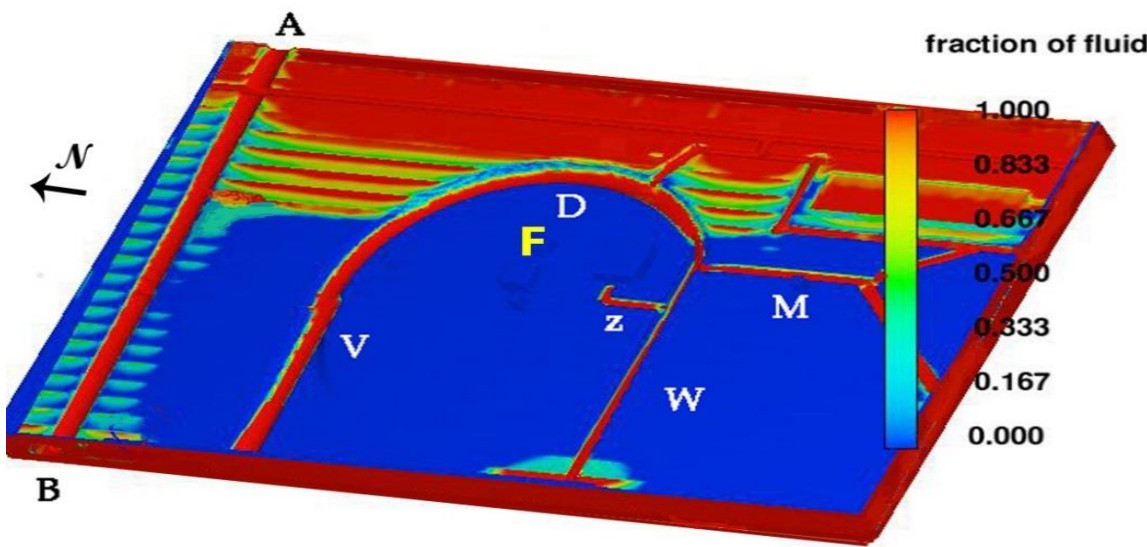

**Figure 50.** Dry season continuance with additional ground surface evaporation drying and aquifer drainage; from [2], p. 179. A,B represents the Tiwanaku River.

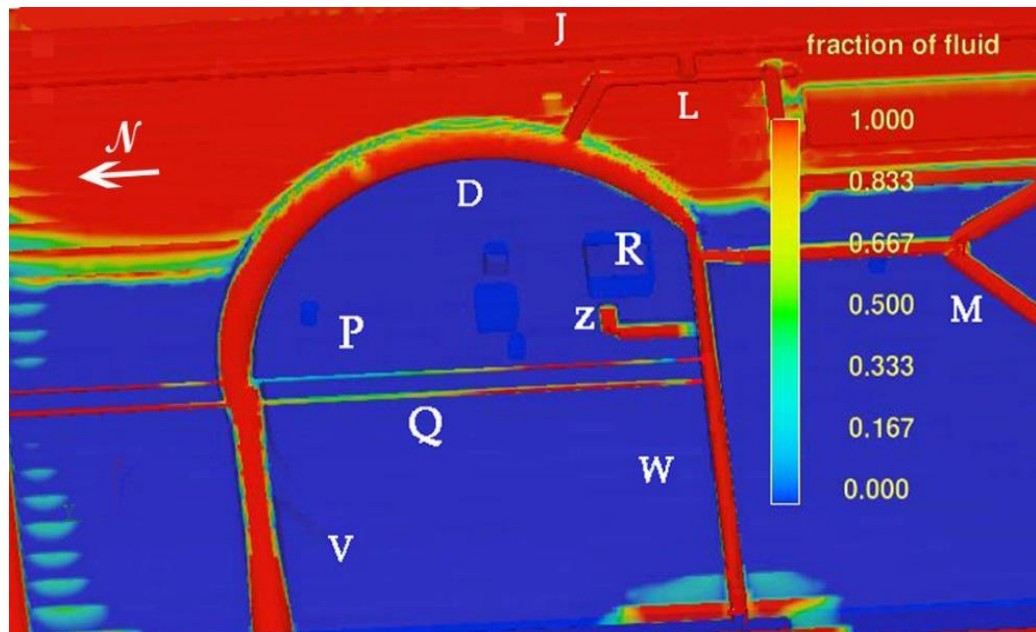

**Figure 51.** Dry season fluid fraction results shown at the depth of subterranean channels P and Q; from [2], p. 180.

Water input from the Mollo Konto M channel input into moat arm W provides water to subterranean channels P and Q to flush elite structure wastewater downhill from the Putini Palace (Figures 44 and 47) complex to drainage channel arm V (Figure 47) then on to the Tiwanaku River A', B' and Lake Titicaca. The Mollo Kontu channel (M, Figure 51) provides water through both wet and dry seasons to the structures within the elite compounds and on to other elite compounds through subsurface channels close to the ground surface (Figure 52). Z indicates a water channel connection from the Kalasasaya Platform (Figure 47, G) to the moat's W arm.

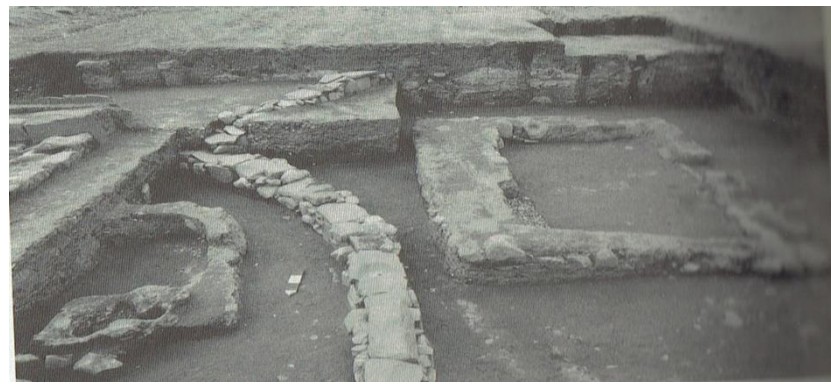

**Figure 52.** Excavated channel within the moat elite quarter diverting water from the P and Q source to other compound structures. Photo by author.

Dry season fluid fraction results on a subterranean plane incorporating the P and Q subterranean channels are shown in Figure 53. Moisture levels in *cocha* region K and depressed marsh area C' indicate sustainable pasturage and agriculture due to contact with the deep water table and the discharge from the V moat arm. Water input from the Mollo Kontu M canal continually delivers water to the moat arm W that enters the subterranean channels P and Q. Aquifer drainage continues into all moat arms that, together with water transfer from P and Q subterranean channels and seepage from the C' marsh area, deliverer drainage water to the Tiwanaku River A', B'. Note that below the P and Q channel depth, the moat bottom penetrates the groundwater-saturated soil level (red area).

**Figure 53.** CFD fluid fraction results during the height of the dry season shown at the depth of subterranean channels P and Q; from [2], p. 180.

Figure 54 shows the dry season end fluid fraction results for the east arm D that indicate decreased seepage water into the moat causing accelerated ground surface drying together with evaporation drying. Mollo Kontu water continues to flow into the moat's W arm and continues to D and V moat arms on the moat's saturated bottom surface to drain into the C' marsh area and channel C enroute to the A', B' Tiwanaku River. The ground surface (blue) now indicates surface dryness to depth for the region's interior and exterior to the moat boundary. Limited water in the moat bottom results from the Mollo Kontu channel input as aquifer seepage is now limited during the terminal phase of the dry season.

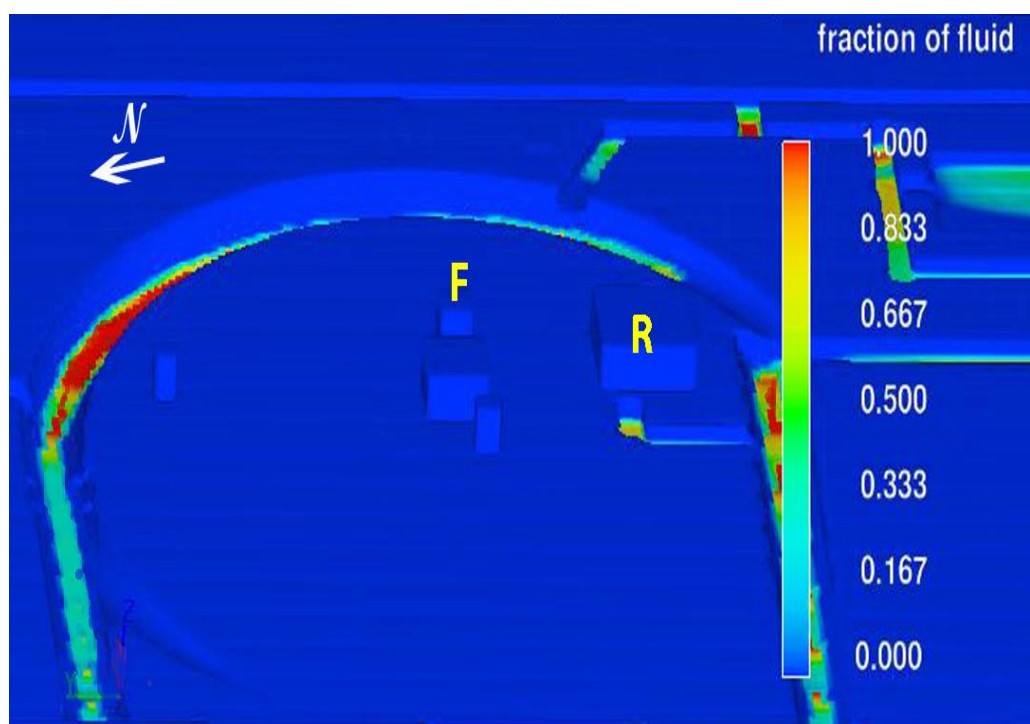

**Figure 54.** Late dry season fluid fraction results; from [2], p. 181.

Figure 55 details wet season conditions typically experienced by a typical moat section and the design intent of the moat as envisioned by Tiwanaku water engineers. Rainy season surface runoff enters the moat from ground surface declination angles of one to two degrees in south–north and west–east directions together with aquifer seepage and contributions from the Mollo Kontu M channel and L, Z channels (Figure 46). As the bottom of the moat is essentially saturated soil dug into the water table, no further water subtractions occur so that water collected in the south-to-north sloped moat can flow by gravity to C and C' connections into the Tiwanaku River and on to Lake Titicaca, as Figure 55 summarizes.

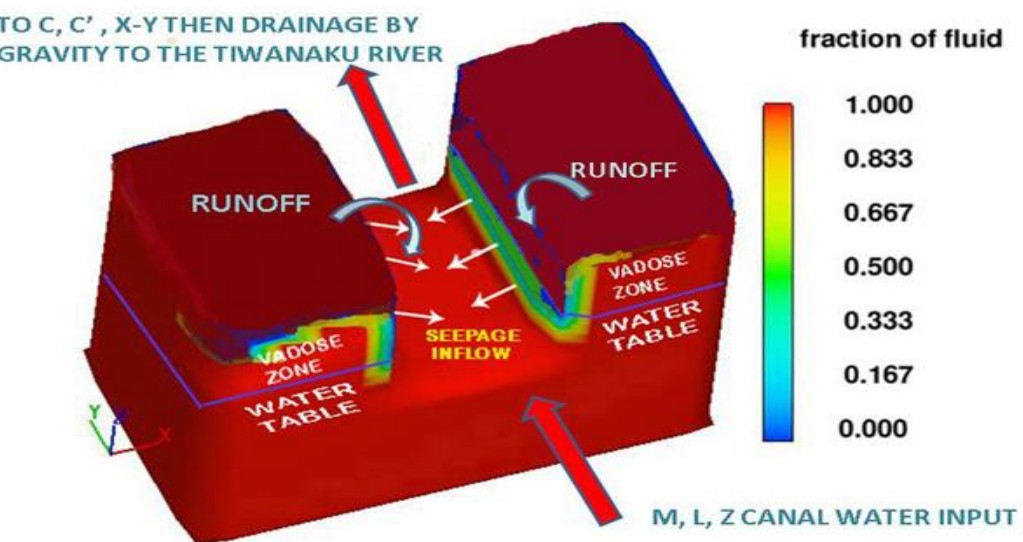

**Figure 55.** Summary of wet season effects on a typical moat section; from [2], p. 178.

During the dry season, Figure 56 indicates a drying ground surface aided by evaporation and ongoing aquifer seepage with continuous water flow from the Mollo Kontu channel and L, Z channels that may have partially reduced flow rates under dry season

conditions. As a result, some decline in the water table may exist with a an extended near-saturated vadose intermediate region.

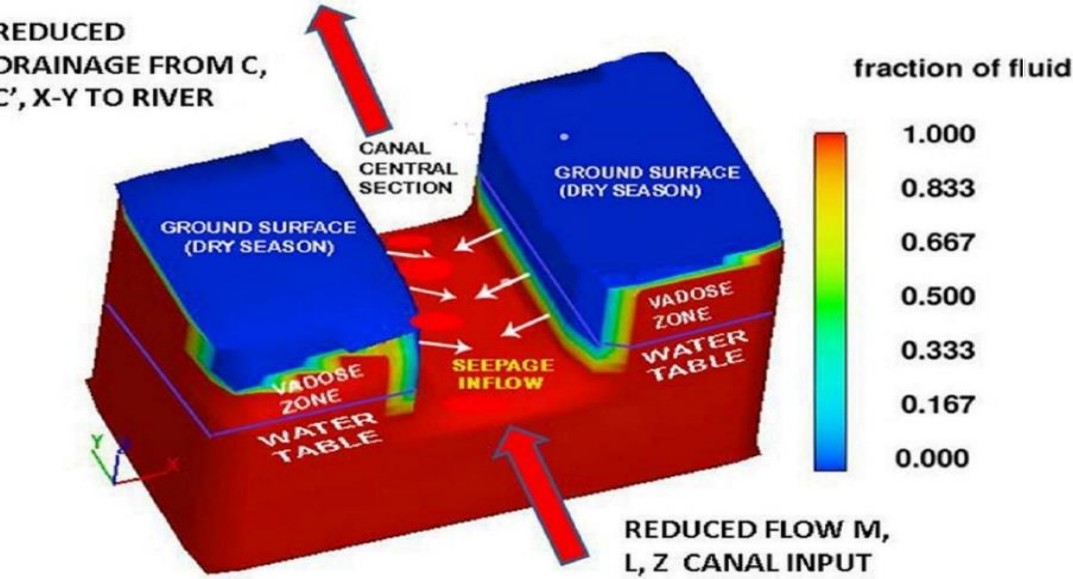

**Figure 56.** Summary of dry season effects on a typical moat section; from [2], p. 181.

In summary, the presence of the moat helps to accelerate seasonal ground-level dryness, limiting mold buildup in site housing structures, which is a definite health benefit. A further benefit is the near-constant high groundwater level maintained through seasonal rainfall and dry periods provided by water input from the Mollo Kontu and L, Z channels. This feature helps maintain the structural stability of the immense Akapana pyramid within the moat boundary (Figure 44) as the underlying saturated aquifer is not subject to compression volume change by weight compression of the temple. With the high groundwater height seasonally maintained, wells can be constructed in individual houses for personal use together with reservoir basins within the city to facilitate personal collection water. Further technical details on the urban and agricultural water system of Tiwanaku are found in ([3], pp. 1–30).

A contribution to the demise of Tiwanaku is attributed to long-term drought [41–44] which slowly over the course of years sank the groundwater level necessary to support raised field agriculture [41,42] supporting the economic base of urban Tiwanaku. Further authors emphasize social unrest and external and internal threats in the termination phase of Tiwanaku, but without a viable agricultural base, no society can continue its existence.

## 9. The Internal Reservoir at the Theater of Roman Ephesus, Turkey—Recovery of Lost Greek Water Technology

Ephesus history dates to long before the Roman occupation period. The city area was first inhabited by the local Carians, then captured by Ionian tribes in ~1050 BC. After 500 years of Ionian rule, the city was invaded by Lydian King Croesus in 560 BC but came to an end after the Persian invasion in 546 BC. Alexander the Great, defeating Persian forces at the Battle of Granicus, entered the city in 334 BC; Ephesus later passed after the death of Alexander to his subordinate commander Lysimachus in 295 BC. Under Greek rule and later Roman rule in later centuries, the city became a trade center with fleets of ships in its Mediterranean harbor serving cities under the Roman sphere of influence. With a population of ~250,000 in Roman occupation times [44], water supplies by new multiple aqueducts, together with numerous architectural changes reflecting the Roman vision of city composition occurred in the form of multiple outdoor gymnasiums, temples, commercial agoras, a coliseum, multiple elaborate baths, public lavatories, multiple public fountains, the Library of Celsus (Figure 57), a massive theater (Figure 58), an administrative center for

Roman officialdom (Hanghaus), numerous marble statues on paved streets, multiple city fountains and castellums, aqueducts, and new water system modifications, were emplaced. Newly built aqueducts led water to several city castellum water distribution structures from which multiple pipelines led to different city districts, serving public fountains, government buildings, baths, public lavatories, the theater, temples, the coliseum, gymnasiums, and private housing districts, as well as *nymphaea* with elaborate statuary incorporating aesthetic water displays. An elaborate subterranean drainage channel system collected waterborne waste material and served the Hanghaus Roman administrative center as well as almost all public use buildings. The water supplied on a continuous basis to public and private structures, when drained into the subterranean channel network, provided continuous flushing of waste material into the harbor area (Figure 59) fronting the Mediterranean sea to maintain high hygienic standards for the city's population.

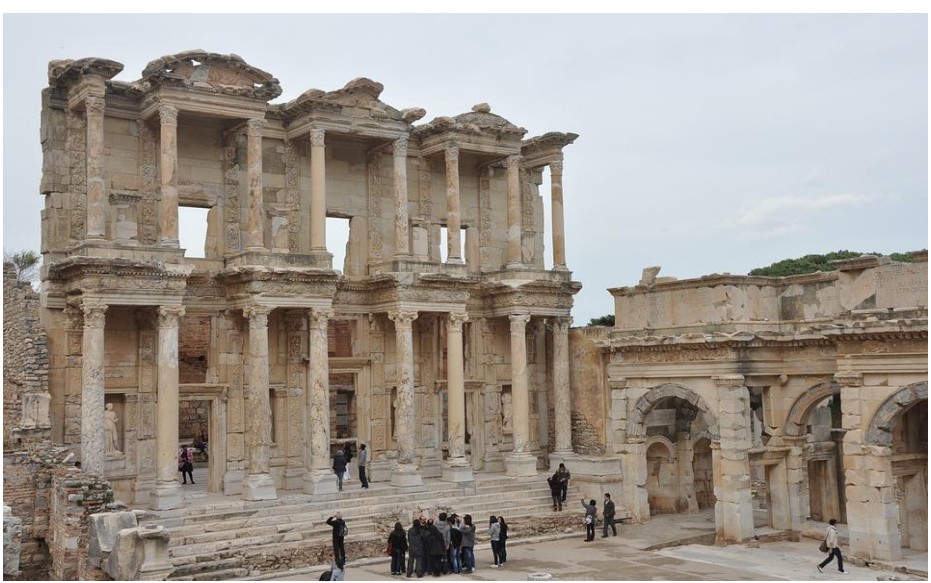

**Figure 57.** The Roman Library of Celsus. Photo by author.

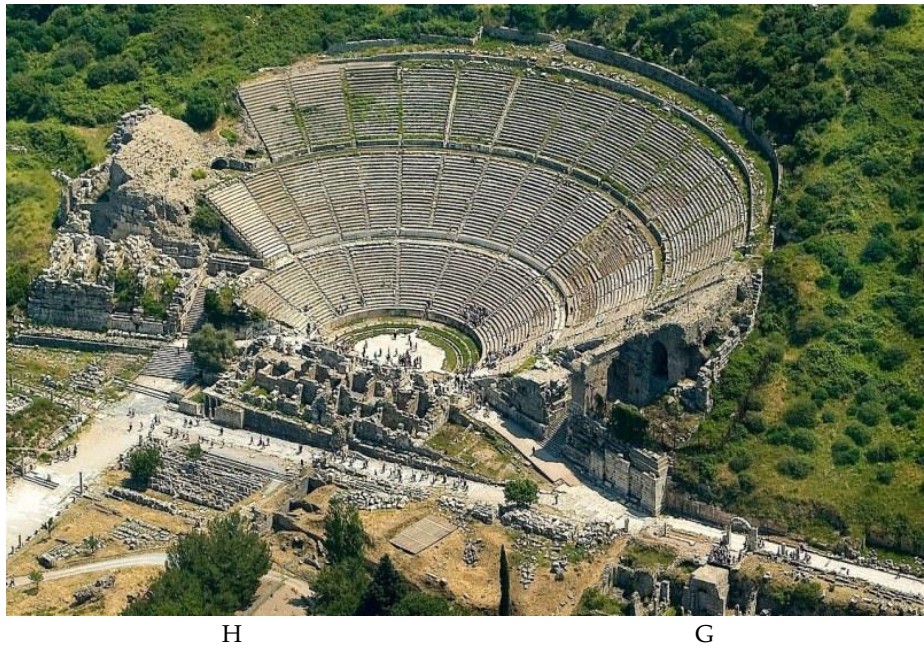

**Figure 58.** The Roman Theater. Aerial photo by author. H represents an earlier Greek structure adjacent to the Theater; G represents a Greek fountain house still in use in Roman times.

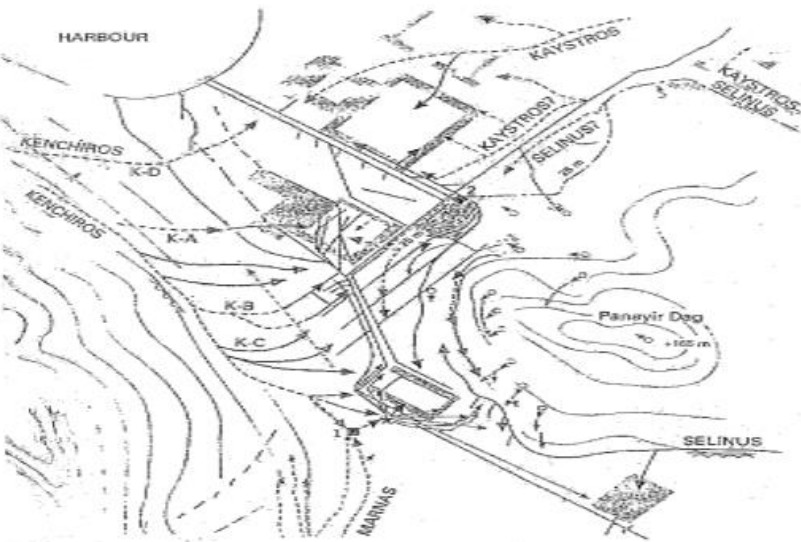

**Figure 59.** Multiple Roman branched aqueducts [1], p. 298.

Further details of Ephesus site history together with FLOW-3D CFD and hydraulic engineering analysis of aqueduct and water drainage systems and current research findings are found in ([1], pp. 295–337, [45]).

In the process of rebuilding the city to Roman standards, remnants of earlier Greek subterranean water channels and pipeline systems no longer of use were buried under new Roman constructions. New water increase requirements were necessary to serve elite Roman government administrative living quarters (Hanghaus), *bouleuterion* meeting quarters for legislative member assemblies, housing for the ~250,000 citizens, marketplace agoras with domestic and imported food and trade items, public baths, public lavatories, recreational gymnasium facilities, and multiple fountains with water collection basins for daily domestic use.

A series of aqueducts (Figure 59), the Kenchiros, Kaystros, Selinus, and Marnias, were the main water supply providers to the city to serve the comforts and needs of the large city population. The Kenchiros aqueduct's many branches (K–D, K–A, K–B, K–C, and others) served specific sites; remaining aqueducts had multiple branches, several of which led to castellums and settling basins to further channel water by pipelines to specific buildings and multiple fountains throughout the city.

Of specific interest is the water supply to the Theater in Roman times from the Selinus aqueduct source. Before later Roman construction, a Greek fountain house with multiple water streams from elevated lion head fixtures existed at the base of the Roman Theater. From five years' participation in excavations in the 1990s with Dr. Dora Crouch and members of the Austrian Archaeological Institute (ÖAI) to track the Roman water supply, distribution, and drainage systems, additional excavations related to the earlier Greek fountain house (located above G on the northside of the road on Figure 58) were made. An unusual structure (Figure 60) buried about ~5 m below the Roman surface was infilled with large stone blocks from Roman construction of the new Theater, and through excavation, the structure was revealed, showing a ceramic pipeline of ~12 cm outer diameter with a wall thickness of ~2 cm of earlier Greek origin positioned about ~1 m back from the back wall of the deeply buried Figure 60 structure. It is surmised that this pipeline segment originally joined with the pipeline serving the upstream Greek fountain house and was later intersected by the Figure 60 structure past which a pipeline (or channel) continued to further Greek structures. The Figure 60 structure had a stone block backwall height of ~4.45 m, a width of ~5 m, and an excavated stone lined path of ~10.3 m from excavations performed. The location of the excavation pit in Figure 60 structure is directly above H on Figure 58 and directly to the left of the lowest row of seats in the Theater at location 2. The

question to be answered is: "What is the function of the Figure 60 structure?"—a question that FLOW-3D CFD analysis can answer.

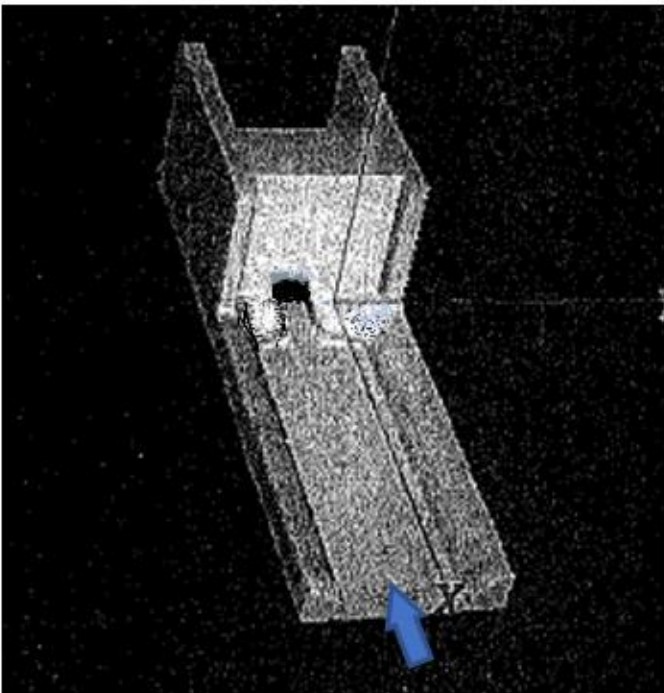

**Figure 60.** FLOW-3D CFD model of the deeply buried water transfer structure. Arrow denotes flow direction. Central opening is ~0.5 m by 0.5 m. Portions of back wall height were destroyed by later constructions. Note the dual ellipsoidal structures at base of wall, the leftmost one shown in Figure 61. Figure derived from [1], p. 332.

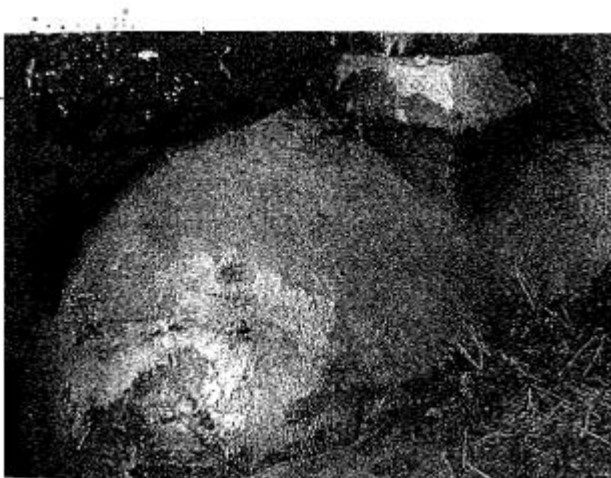

**Figure 61.** Photograph of the leftmost fractional ellipsoidal structure at the base of deep excavation pit. Photo by author.

The Figures 60 and 61 FLOW-3D CFD models show a rectangular opening on the backwall, ~1.3 × 1 m in size, that had ellipsoidal hardened clay sections on each side of the wall opening, as shown in Figure 61, presumably to facilitate smooth entry flow. It may be surmised that the revealed pipeline section shown past the backwall opening (Figure 62) was once part of a continuous longer pipeline that provided water to the Greek fountain house and further on by pipeline or channel to Greek structures. This pipeline was intersected by the elaborate structure represented by Figure 60. Water flow from the

fountain house pipeline then flowed into the Figure 60 basin and continued through the lower wall opening to an open channel or pipeline serving Greek structures east of it, as shown in Figure 58.

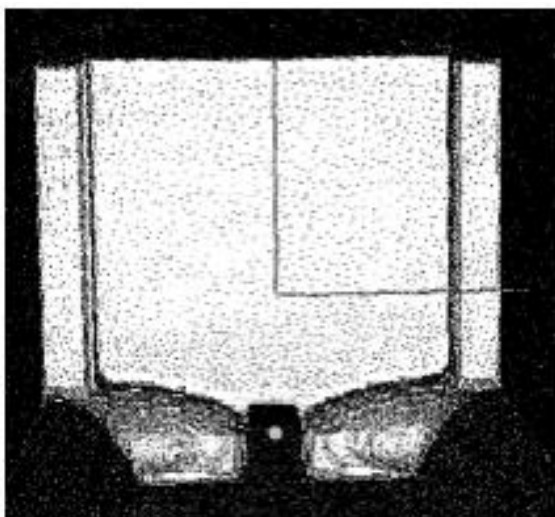

**Figure 62.** FLOW-3D CFD model of front view of the discontinued internal pipeline located ~ 2 m back from the wall opening.

Given that hard water through long ceramic pipelines precipitates calcium chloride sinter over long usage times that ultimately limits water transport through the progressively narrowing internal pipeline opening, it may be surmised that the Figure 60 structure was proposed by Greek water engineers as a "fix" to the anticipated water blockage problem to continue water flow to other city structures. Additionally, silt particles originating from the spring source flowing in a low declination angle pipeline can settle in low-speed water flows, further increasing flow blockage. Intuitively, the dual ellipsoidal structures adjacent to the lower back wall opening would appear to create smooth contracted streamlines from entry flow into the rectangular back wall opening to facilitate flow passage, but determining the hydraulic engineering function that Greek water engineers had in mind with the Figure 60 structure can only reliably come from CFD analysis.

Excavations revealed that this transition structure existed between the Greek fountain house and the entrance to the Figure 60 structure. To determine the intended design effect of the Figure 60 structure, CFD analysis was performed, originating the velocity vector plots given in Figure 63A,B. The upper (Figure 63A) view is at the midpoint of the entry channel and indicates circulatory flow patterns upstream of the rectangular back wall opening. Results from Figure 63A indicate that the dual vortex flow is nonsymmetrical and turbulent (the k–ε turbulence model is incorporated in the FLOW-3D CFD calculation) before entry into the rectangular opening. A further view from Figure 63B at a plane just above the bottom surface reveals a circulatory flow pattern induced by the Figure 60 structure. Figure 63A,B shows further velocity vector plots indicating chaotic flow patterns. As opposed to an intuitive estimate of the effect of the dual ellipsoidal structures to smoothly transition incoming flow through the rectangular back wall opening, an opposite effect is achieved from CFD analysis as flow appears highly chaotic before its entry through the back wall rectangular opening.

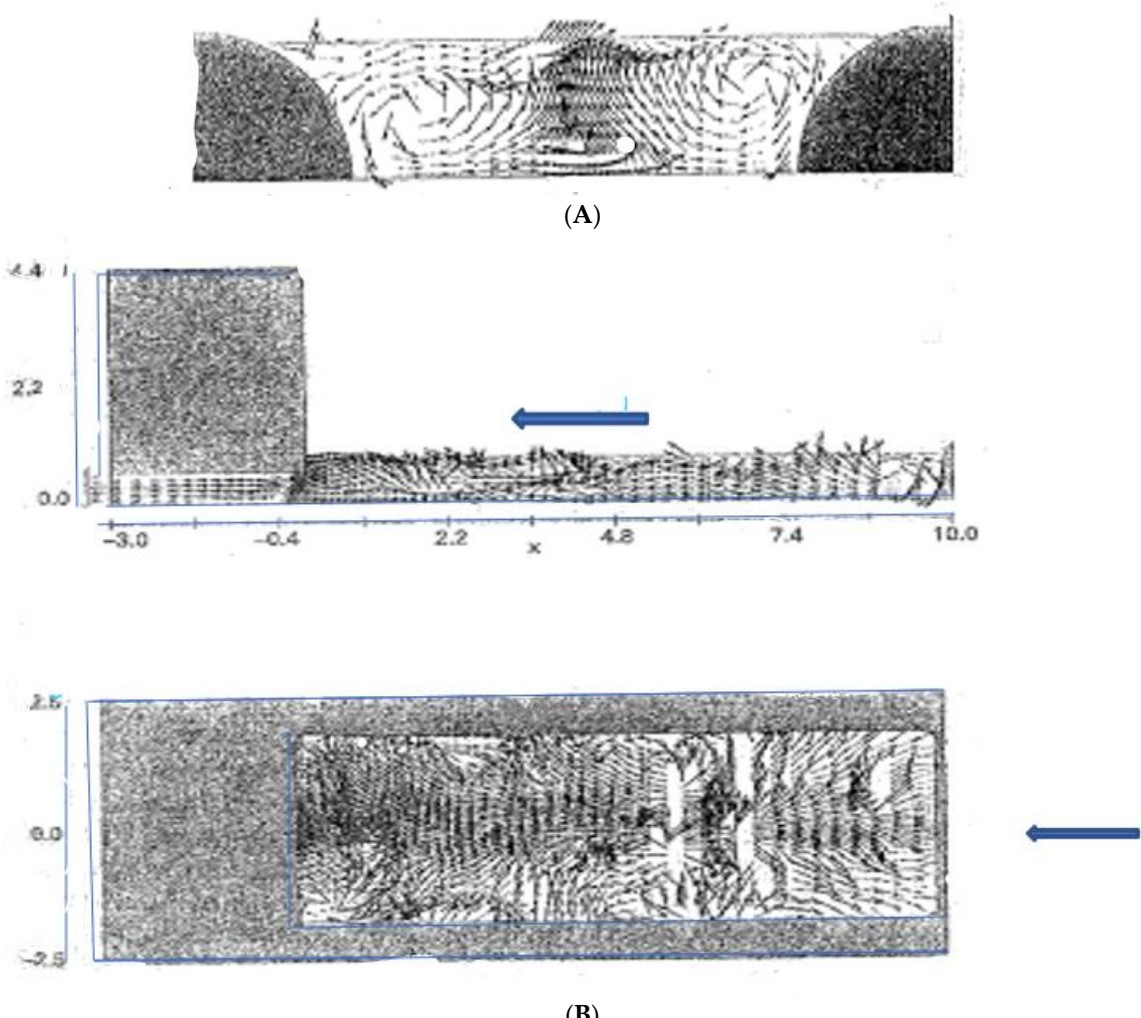

**Figure 63.** (**A**) Front view of flow velocity vectors prior to the back wall opening; Ref. [1], p. 336. (**B**) CFD midpoint side view (**top figure**) and top view (**bottom figure**) velocity vectors; Ref. [1], p. 335. Figures indicate a high creation level of water vortical structures and turbulence before the wall opening. Arrow represents flow direction. Scale in Av meters.

An interpretation of the intended effect of the Figure 60 shaping by Greek hydraulic engineers would be to induce chaotic flow that would entrain any settled silt and debris particles that arrived from the incoming pipeline and permit their transfer further on to a downstream settling basin where they could be removed to bring water quality to potable level for downstream occupied administrative structures.

The development of this water system is as follows: (1) the original lengthy pipeline that passes the Greek fountain house slowly transports deposited silt from fluid shear effects together with entrained silt sinter and debris particles to ultimately clog the pipeline intended to transport water to distant Greek housing structures; (2) Greek water engineers install the Figure 60 structure whose chaotic flow (Figure 63A,B) entrains arriving silt sludge, sinter, and debris particles back into circulation through a channel (or pipeline) continuing from the back wall rectangular opening to a downstream settling basin to clarify water to potable level for downstream occupied Greek structures; (3) as the Figure 60 structure is located at a low level within the earlier Greek theater, there may have been lavatory human waste also channeled into the water stream preceding the Figure 60 structure that provides a further reason for keeping particles in suspension prior to entry to a downstream settling tank; (4) with later Roman aqueduct constructions, new water supply and drainage systems (Figure 59) supplying the new Roman Theater were built over prior Greek constructions.

Excavations of the Figure 60 structure revealed large stone block infilling in Roman times. Some aspects of the earlier Greek theater design continued into Roman times with stage construction modifications to accommodate citizens that enjoyed the benefits of plays from both Greek and Roman authors.

Of note is Greek hydraulic technology involving sediment deposit removal from water structures. Figure 15 indicates a hydraulic structure at the Greek site of Priene designed to clear the major drainage system from agora waste debris and washed-in debris from city housing. Figure 60 indicates a Greek hydraulic structure at Ephesus designed to return settled pipeline transported silt and sinter debris back into highly chaotic flow (Figure 63A,B) so that it can be removed in a downstream settling tank to provide potable water to downstream structures. Similarly, hydraulic structures at Ephesus [45] supplying clearing water to public lavatories has similar turbulence-inducing structures to keep debris and human waste products in suspension before delivery by pipeline into the bay area. The sacred water basins of Demeter, located in Figure 13B at Priene's northern hills, were primarily used for fertility rituals and have a subterranean pipeline water supply downhill from the basins to ensure that sediments settle before entry to uphill ceremonial-use basins. From observed excavations within urban Priene, a pipeline supplied spring water to a first settling basin with overflow to a second settling basin, then overflow from the second basin to a third settling basin whose purified flow continued to public fountains and basins for domestic use. Sediments accumulated at basin bottoms clearly purified water for domestic use. Greek water purification technology from several example cases presented indicates the presence of advanced hydraulic engineering knowledge not previously reported in the open literature.

## 10. Inka Water Technology at the Royal Site of Tipon

The present chapter utilizes modern hydraulic engineering technology to uncover Inka water technology at the site of Tipon.

The site of Tipon, located in Peru, ~13 km east of Cuzco along the Huatanay River, at south latitude 13° 34′ and longitude 71° 47′, at 3700–4000 masl is known for its many unique hydraulic features coordinated in a practical and aesthetic manner to demonstrate Inka knowledge of water control principles. Details of the water supply canals and architectural features of the Tipon site are shown in Figure 64. The site has an early Middle Horizon (600–1000 AD) presence, evidenced by an encircling ~ 6.4 km long outer wall (Inka Canal Principal, Figure 64) attributed to earlier Wari control of the enclosed area.

The ~2 km$^2$ interior site area was under Inka control after ~1200 AD and was later converted into the royal estate of Inka Wiracocha in the early 15th century, as evidenced by the royal residence and ceremonial compounds of Sinkunakancha and Patallaqta. The site is composed of thirteen major agricultural platforms (Figure 65), with the lowermost platforms associated with nearby ceremonial centers. The agricultural platforms were mainly irrigated from water supplied from a branch of the Main Aqueduct sourced from the Rio Pukara (Figure 64); branches of the Main Aqueduct provided water to the lowermost ceremonial areas.

Each agricultural platform had a drainage channel at its base that collected post-saturation aquifer groundwater seepage as well as rainfall runoff; excess water beyond that required to irrigate the next lower agricultural platform was led to side drainage channels directed to lower site occupation and ceremonial areas. As only a fraction of each platform's water supply was used to provide the necessary moisture level for specialty crops, excess water was passed on to the next lower platform with excess water directed to an easternmost collection channel. A further portion of water from agricultural platform seepage collection channels passed through a series of interconnected surface and subsurface channels to provide water to the Principal Fountain (Figure 66) supplying the Waterfall structure that supplied water to domestic and elite residential and ceremonial areas at lower site areas. High-level springs were an additional water source to Tipon's intricate water supply and

control system to guarantee that Tipon's water systems functioned on a year-round basis appropriate for a royal residence site.

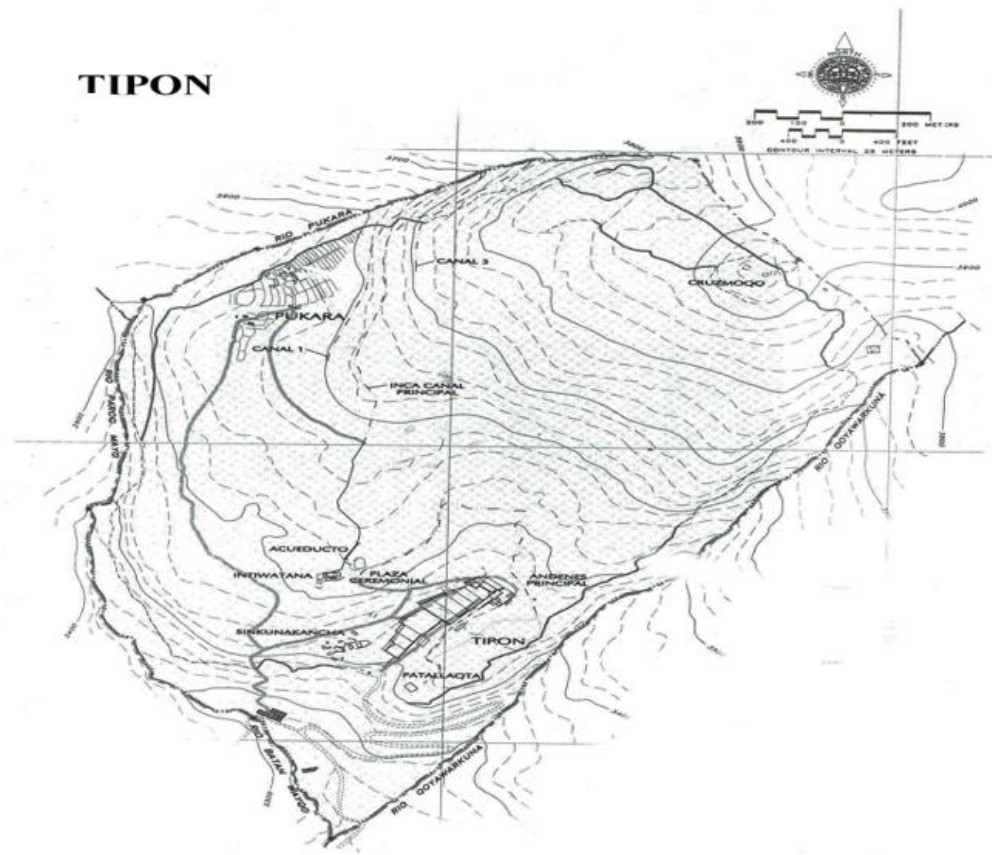

**Figure 64.** Site feature map of Tipon from Wright et al. 2006 [46].

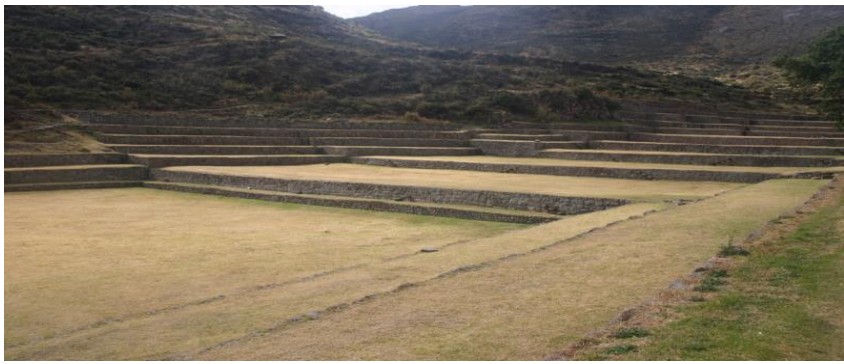

**Figure 65.** Thirteen agricultural terraces at Tipon. Photo by author.

Of special interest is the geometry of the upstream water supply channel (Figure 66) from the Principal Fountain to the Waterfall structure shown in Figure 67. A channel contraction occurs from a 0.9 m wide, ~2.5 m long entry channel to a ~0.4 m contracted wide, ~10.5 m long channel upstream of the Principal Fountain. Both channel sections have a rectangular cross-section at the same mild low slope. The water source to the upstream wider channel section derives from eight separate water supply conduits together with a major spring, indicative of the totality of water control systems used to maintain a constant flow rate to the Waterfall during seasonal changes in water supply. Measured flow rates into the (reactivated) wide channel from two different tests [46] yielded 0.68 ft$^3$/s and 0.58 ft$^3$/s, leading to an average 0.63 ft$^3$/s (0.02 m$^3$/s) flow rate. The question arises as

to the water engineering design intent of the abrupt width change of the channel section shown in Figure 66 and its effect on the aesthetics of the downstream Waterfall.

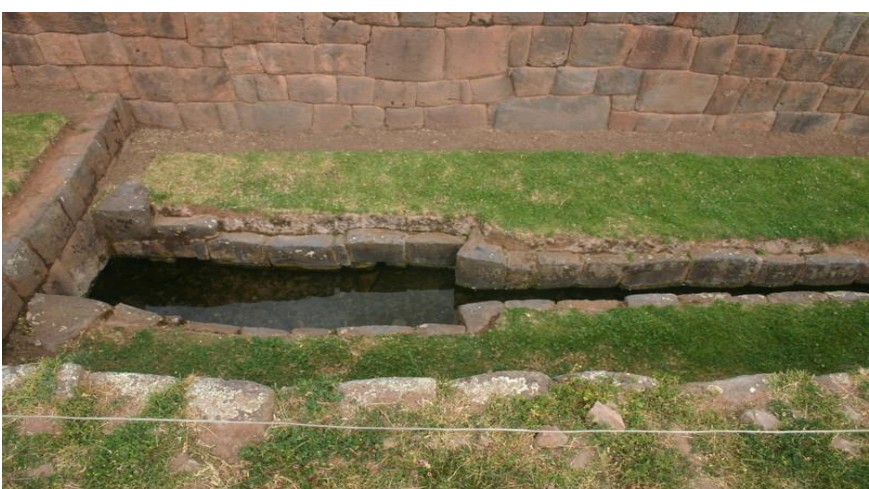

**Figure 66.** Channel cross-section geometry change made to induce critical flow in the contracted channel section to downstream channels supplying the Figure 67 Waterfall. Photo by author.

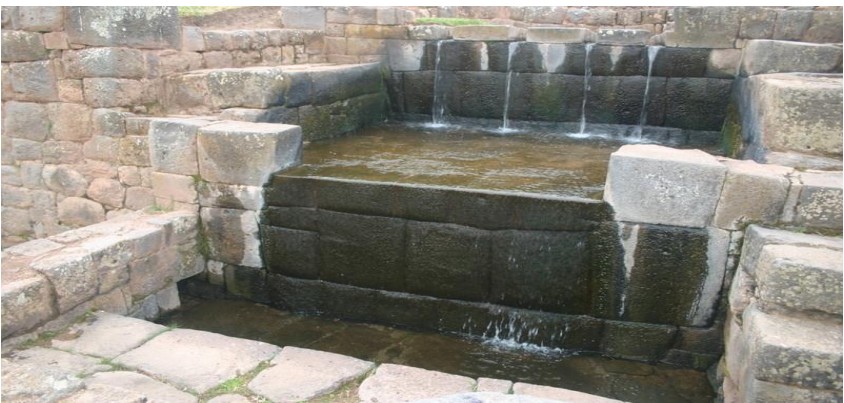

**Figure 67.** Waterfall structure supplied from the Figure 66 channel. Photo by author.

It is noted that the site of Tipon was reactivated after years of abandonment by restoring the water sources and channel systems as in Inka times. While reconstruction of certain time-degraded elements of the site have been made, the stonework associated with Figures 65–67 is original and therefore of use to determine the hydraulic engineering function intended by Inka hydraulic engineers. In the discussion to follow, use of modern hydraulic engineering practice is employed to reveal Inka hydraulic engineering practice. Although the reason behind the Inka design can be made apparent through this method of analysis, the Inka source of hydraulic engineering knowledge paralleling the modern analysis remains unknown as the Inka had no written language nor any known hydraulic test facilities that have yet been identified. The source of Inka hydraulic knowledge may therefore derive from observed nature observations of flow phenomena then codified into practice in their urban and agricultural water systems, or possibly from importation of Chimú water technicians from Inka-conquered north coast territories in the late 14th century CE.

To understand Inka hydraulic engineering in terms of modern hydraulics technology, use of the Froude number (Fr) is convenient to explain water behavior. For shallow depth D flows, the Froude number is $Fr = V/(g\,D)^{1/2}$, where V is the water velocity and g is the gravitational constant. Physically, Fr is the ratio of water velocity V to the gravitational wave velocity $(g\,D)^{1/2}$—when Fr > 1, water velocity exceeds the signaling gravitational

wave velocity so that water has no advance warning of a downstream obstacle; this leads to the creation of a sudden hydraulic jump before the obstacle. In physical terms, for Fr > 1, there is no upstream awareness of the presence of an obstacle until the obstacle is encountered by the water flow, as the gravitational wave signaling velocity that informs the flow that obstacle exists $(g\,D)^{1/2}$ is much less than the V flow velocity. For Fr < 1, the gravitational wave signaling velocity travels upstream of the obstacle faster than the water velocity V to inform the incoming water flow that an obstacle lies ahead. This causes the water flow to adjust in height and velocity far upstream of an obstacle to produce a smooth flow over the obstacle. The "obstacle" for the present application is the large contraction in channel width shown in Figures 66 and 68. Fr > 1 flows are denoted as supercritical, Fr < 1 flows are denoted as subcritical, and Fr = 1 flows are denoted as critical. While the presence and characteristics of subcritical, critical, and supercritical flows can be calculated from modern hydraulic theory, a simpler method exists to determine flow types. Insertion of a thin rod into a water flow that produces a downstream surface V-wave pattern behind the rod is indicative of supercritical flow. If the water surface pattern shows upstream influence from the rod, then subcritical flow in a channel is indicated. If only a local surface disturbance around the rod is noted, then critical flow is indicated. This simple test can determine different flow regime types to promote various usages associated with the different Fr flow regimes. As a first observation of water flow patterns giving insight to the hydraulic technology that went into the Figure 66 channel design, Figure 68 gives the first clue. The parallel water surface ripple wave structure pattern normal to the flow direction is consistent with $\sin\theta = 1/Fr$, where $\theta$ is the half angle of the surface wave, so that when Fr ≈ 1, $\theta \approx 90°$, verifying the transverse surface ripple wave structure shown in Figure 68 [19–21]. From Figure 68, incoming water flow in the wider channel section is (Fr < 1) subcritical, and in the narrower channel section, critical flow (Fr > 1) exists. Why then, was this flow type change important to Inka water engineers?

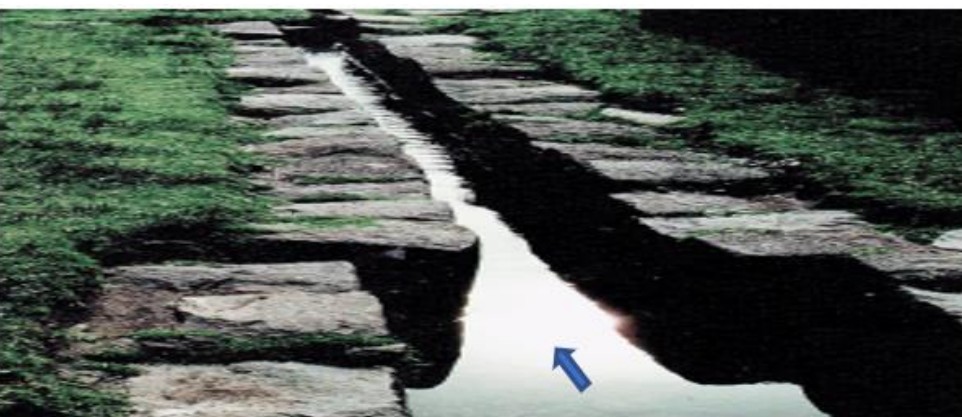

**Figure 68.** Alternate view of the contracted water supply channel, showing surface ripples normal to the flow direction. Photo by author.

Figure 69 is derived from the Euler fluid mechanics continuity and momentum equations that govern fluid flow behavior and is useful to describe the flow transition from sub- to supercritical flow in the supply channel, as shown in Figures 66 and 68, where water viscosity effects have minor effects on flow patterns [47,48]. As all water motion is governed by the mass and momentum conservation equations, Figure 69 indicates flow transitions based on Froude number change due to channel geometry changes. The *x*-axis represents $(1/2)\,Fr_1{}^2$ incoming flow (IN) conditions into the wide channel section; the $(1/2)\,Fr_2{}^2$ conditions represent flow (OUT) conditions in the continuing narrow channel section. The W2/W1 curves represent width ratios of the incoming subcritical flow section (W1) to the downstream channel (W2) width change. W2/W1 > 1 represents channel expansion and W2/W1 < 1 represents channel contraction, such as shown in Figures 66 and 68 which illustrate different views of the same channel contraction section.

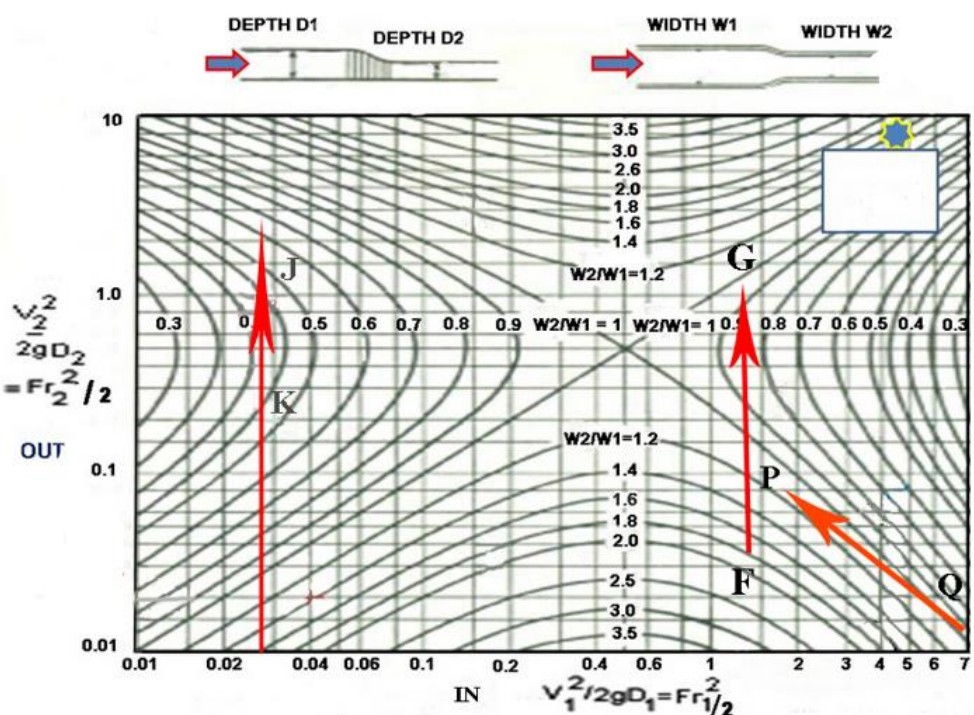

**Figure 69.** Froude number relationship between flow entering the channel wide section (IN) and passing through the contracted channel section (OUT) from Bakhmeteff, 1932 [47].

Using the subscript notation (1) for flow conditions in the wide channel section and (2) for contracted width conditions, the Figure 69 width contraction ratio is W2/W1 = 0.44. The (1) flow entry value using the average flow rate is based on a ~0.3 ft water depth for which $V_1 = 0.72$ ft/s and Froude number is $Fr_1 = V/(g\ D)^{1/2} = 0.23$ ($Fr_1^2/2 = 0.03$ in Figure 69). The contracted channel (2) Froude number is $Fr_2 \approx 1.14$ ($Fr_2^2/2 = 0.65$ in Figure 69) based on ~0.3 ft depth. The channel contraction shown in Figures 66 and 68 takes water flow from subcritical flow (Fr < 1) in the wide channel to a near-critical (Fr ≈ 1) flow in the narrowed channel.

The flow rate per unit width in the contracted rectangular cross-section channel is $q_2 = 0.68/W2 = 0.52$ ft$^3$/s ft and the critical water depth (Henderson 1966) is $y_c = (q_2^2/g)^{1/3} = 0.2$ ft, which is in agreement with the previous critical water depth value calculated for near-critical Fr ≈ 1 flow in the contracted width (2) channel.

Figure 69 illustrates the flow Froude number IN to OUT transition arrow (K to J) resulting from the wide (1) to narrow channel (2) shape change that incorporates transition from sub- to near-critical flow. This indicates that Inka engineers designed the (2) contracted channel section to support near-critical Fr ≈ 1 flow, but not exact critical Fr = 1 flow in the expanded part of the channel. In modern channel design practice, ~0.8 < Fr < ~1.2 is the prescribed Fr range to obtain the highest flow rates that accompany Fr ≈ 1 flows to avoid surface wave instabilities associated with translating large-scale vortex motion below the water surface associated with exact Fr = 1 critical flow. The $Fr_2$ Froude number range in the contracted (2) channel lies between ~0.8 < Fr < ~1.2, and the computed $Fr_2$ ~ 1.14 value is consistent with stable flow according to modern hydraulic engineering. The width reduction construction yields the narrowest supply channel (2) at the maximum flow rate per unit channel width without significant internal surface wave structures causing transient flow instabilities to be passed further downstream to the Waterfall area. In other words, the channel critical flow section does not permit any transient disturbance irregularities from the downstream Waterfall and its water supply channel from propagating upstream to influence the subcritical flow in the wider channel section to cause flow instabilities. The Figures 66 and 68 channel leads to a stilling reservoir ahead of the waterfall to limit disturbances and promote equal, stable flows into the four Waterfall channels. Any dif-

ference in equal amounts of water flowing through each of the four Figure 67 Waterfall channels would produce transient vortex structures in the stilling reservoir that constitute instabilities that the Fr = 1 contracted channel section does not allow to influence [22,48,49] the incoming Fr < 1 flow in the wider channel part of Figure 66.

The contracted Fr > 1 channel (2) therefore serves to limit any disturbances propagating upstream to disturb flow aesthetics in the Waterfall; this the design intent of Inka water engineers. Waterfall aesthetics were a prime concern at the Inka royal site, requiring advanced water engineering knowledge to accomplish. Other Figure 69 RHS arrow markings refer to flow properties of the Inka Canal Principal which involves complex analysis to determine its flow rate; results of this analysis are available ([3], pp. 195–221).

Near-critical flow is associated with the maximum flow rate per unit channel width that a channel can support [19–21]; this is closely achieved for $Fr_2 = 1.14$ conditions. Flow stability is achieved by the current Inka design; knowledge and use of this hydraulic engineering practice is evident in the Principal Fountain/Waterfall design and is vital to produce a constant, stable water delivery flow to the Waterfall area.

In summary, this contracted channel flow design is important for the following reasons: (1) the flow from the spring and channel sources channeled into the wide channel then to a downstream narrow width channel segment is associated with the maximum flow rate per unit channel width; (2) the Fr > 1 flow value in the contracted channel eliminates flow disturbances propagating upstream to the wide Fr < 1 channel; (3) the settling reservoir immediately upstream of the waterfall is intended to give symmetrical flow delivery to the waterfall: this is difficult to achieve but the Fr > 1 narrow channel prevents any disturbances propagating from this effect influencing the wider channel section input Fr < 1 flow.

Why is this size choice of channel sections important? If Inka engineers chose channel sections with wider widths throughout than those shown in Figures 67 and 68, then Fr < 1 would exist in all channel sections, making any downstream disturbances propagate in both up- and downstream directions, affecting waterfall aesthetics. Any water supply decrease (due to drought) or increase (due to excessive rainfall) disturbances would alter the original design intent, causing erratic transient Waterfall patterns that produce a nonaesthetic display. An overview of the Inka technology used for the Principal Fountain/Waterfall display reflects modern hydraulic design principles to preserve fountain aesthetics. Inka knowledge of channel width change effects on subcritical and supercritical flow regime creation to achieve stable flow at the maximum flow rate to the Waterfall is apparent in the channel design; this analysis adds a new understanding of Inka hydraulic engineering capabilities.

## 11. Concluding Remarks

Pythagoras (582–500 BC) stated that "Everything in the universe is governed by mathematical rules and reasons—if we understand the arithmetic and mathematical relations, then we will understand the structure of the universe and mathematics as the basic model for philosophical thought". The present nine chapters bring forward the relevance of his comments into modern times to add a further dimension to what is currently known of archaeological sites and the knowledge base of ancient water engineers not previously reported in the open archaeological and engineering literature. Of the new world sites discussed in Sections 2, 5, 7 and 9, their history, political, religious, and economic science development is well represented in the open literature—to this knowledge base, new revelations relevant to the water engineering knowledge base of pre-Columbian societies are brought forward for the first time to add a further dimension to the accomplishments of these societies. For old world sites discussed in Sections 1, 3, 6 and 8, their archaeological and cultural histories are well known through long periods of excavation and research by many investigators. New revelations at major World Heritage sites regarding hydraulic engineering discoveries at their major urban and agricultural centers developed by ancient water engineers are now brought forward by CFD analysis to add a further dimension of their engineering accomplishments to the archaeological record.

The present manuscript is designed as an introduction to the new field of paleohydrology in a readable manner for scholars familiar with hydraulic engineering practice, scholars in the field of archaeology, and a general audience interested in historical and scientific fields of study. Readers of this manuscript interested in further technical and historical detail of individual sites analyzed in this manuscript are provided references relevant to each chapter where sociopolitical, socioeconomic, and sociocultural matters are discussed in detail to give a more comprehensive accounting of site research performed by others over many years of research. References quoted under my authorship are based upon my fieldwork at each of the chapters' sites conducted over many years based upon hydraulic engineering studies. While paleohydraulics studies utilizing modern CFD and hydraulic analysis tools are new to archaeology to discover technologies utilized by ancient water engineers, work presented in this manuscript thus far presented is but a first step, as many old and new world sites remain to have their secrets revealed.

With respect to obstacles and uncertainties in the use of modern hydraulic engineering methods to uncover ancient water technologies, it came as a surprise that many World Heritage sites did not have journal publications involving modern hydraulic engineering analysis used to reconstruct ancient versions of water engineering. This is an opportunity for present-day water engineers to contribute something new to world history. As many sponsored and funded field projects involve excavation and mapping of archaeological sites and discussion of site historical relevance, those interested in paleohydrology have to consider travel to world archaeological sites with individual walkabout "tourist" field surveys and data collection of surface water systems. The option of joining sponsored projects requires background reading in depth of archaeological sites of interest then contacting archaeology departments of academic institutions to ask if they are interested in paleohydrology inputs that can add a further dimension to their research. Sites investigated in this manuscript evolve from a combination of these paths. As archaeology requires an element of uncertainty and conjecture about the significance of a finding, this, as the present paper signifies, only deepens interest in pursuing paleohydrology as way to bring forward what one's water engineering brothers of past times had accomplished and to bring their discoveries to the world's attention.

**Funding:** This research received no external funding.

**Institutional Review Board Statement:** Not applicable.

**Informed Consent Statement:** Not applicable.

**Data Availability Statement:** Additional research findings commentary are given in references.

**Conflicts of Interest:** The author declares no conflict of interest.

## Appendix A

*Appendix A.1. Petra Figure 1 Captions*

1. Thommanon
2. Chau Say Tevoda
3. Speam Thma
4. Hospital Chapel
5. West Gate
6. Elephant Terrace
7. Leper King Terrace
8. South Gate
9. Preah Pithu
10. Tep Pranam
11. Praasnt Kravan
12. Banteay Kdei
13. Ta Prohm
14. Preah Rup
15. Ta Prohm Kei
16. Baphuon
17. Phnom Bakhang
18. Bakani Chamkrong
19. Preah Palitry
20. South Klrang
21. Canal Systems

*Appendix A.2. Figure 17 Angor Site Captions*

1. Zurraba, M reservoirs
2. Petra Rest House
3. Park entrance
5. Dijn Monument
6. Obelisk Tomb and Triclinium
7. Siq Entrance elevated arch remnants
8. Flood bypass tunnel and dam
9. Eagle Monument
10. Siq passageway
11. Treasury (El Kazneh)
12. High Place Sacrifice Center
13. Dual Obelisks
14. Lion Fountain Monument
15. Garden Tomb
16. Roman Soldier Tomb
17. Renaissance Tomb
18. Broken Pediment Tomb
19. Roman Theater
20. Uneishu Tomb
21. Royal Tombs (62, 63, 64, 65)
22. Sextus Florentinus Tomb
23. Carmine Façade
24. House of Dorotheus
25. Colonnade Street
26. Winged Lions Temple
27. Pharaoh's Column
28. Great Temple
29. Q'asar al Bint
30. Museum
31. Quarry
32. Lion Triclinium
33. El Dier (Monastery)
34. 468 Monument

35. North City Wall
36. Turkamaniya Tomb
37. Armor Tomb
38. Outer Siq Wadi Drainage
39. Aqueduct
40. Al Wu'aira Crusader Castle
41. Byzantine Tower
42. North Nymphaeum
43. Paradeisos, Temenos Gate area
44. Wadi Mataha Major Dam
45. Bridge Abutment
46. Wadi Thughra Tombs
47. Royal Tombs
48. Jebel el Khubtha High Place—pipeline
49. El Hubtar Necropolis—buried channel
50. Block Tombs **B** dual pipeline supplied basin area
51. Royal Tombs d catchment dam
52. Obelisk Tomb s spring
53. Columbarium c cistern
54. Conway Tower D multilevel dam
55. Tomb Complex A Kubtha aqueduct
56. Convent Tomb T Water distribution tank
57. Tomb Complex -d major dam
58. Pilgrim's Spring
59. Jebel Ma'Aiserat High Place
60. Snake Monument
61. Zhanthur Mansion
62. Palace Tomb
63. Corinthian Tomb
64. Silk Tomb
65. Urn Tomb
66. South Nymphaeum
67. Rectangular Platform/Temple

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
