# Peer review of "CFD Investigations of Water Supply and Distribution Systems of Ancient Old and New World Archaeological Sites to Recover Ancient Water Engineering Technologies"

_water, doi:10.3390/w15071363_

Round 1
Reviewer 1 Report (Previous Reviewer 1)
Dear author,
Please address the following two easy but essential issues:
1- Please add a proper reference to each figure, map, diagram, sketch etc., including who first published the figures, who took the photos and when?
2- To avoid repetition, please make the current article different from your previous article ((Ortloff, C.R. Hydraulic Engineering at 100 BC-AD 300 Nabataean Petra (Jordan). Water 2020, 12, 3498. https://doi.org/10.3390/w12123498)). You can do that by going along with the proper references and quotes.
Best regards
Author Response
Reviewer 1: References added to each figure in the revised manuscript. Added notes to each figure indicates the source of each figure, e.g., The "Photo by author on site in ***."

Reviewer 2 Report (Previous Reviewer 2)
The manuscript concenrns nine chapters to follow present brief summaries of ancient sites’ water systems and the use of CFD and modern hydraulic engineering methods to discover the water engineering knowledge base used by ancient water engineers- results are new revelations unknown in the current literature of ancient sites. . Paleohydrology studies presented serve to add a further dimension to the history of ancient New and Old World archaeological sites by bringing forward new details of water engineering projects accomplished by ancient engineers.Remarks: Better quality of some figures should be provided, eg. Line 176: Figure 4. FLOW-3D CFD model of the pipeline connections constructed from Figure 3. Figure 63. C. CFD midpoint side view (top figure) and top view (bottom figure) velocity vectors. The manuscript is too long, some materials from the text should be included in the appendix. The proposed approach should be discussed in terms of its applicability in the conslusion.
Author Response
Reviewer 2: Poor quality figures have been replaced by improved figures in the revised manuscript. For example, Figure 9 has been replaced by a new figure; Figure 14 has been modified to improve its presentation; Figure 16B has been eliminated; Figures 13 and 17 have been replaced by a higher quality figures.
Note also that in the revised manuscript, all figures from previous publications have been given the reference source.

Reviewer 3 Report (New Reviewer)
The authors investigated the use of computational fluid dynamics (CFD) for paleohydrology. The reviewer suggests major revisions before the publication to the journal.
1) The characteristics of the paper is not clear. The novelty is lacking as an original article, because the content seems to have been published in other publications [reference 1-3]. The comprehensiveness is lacking as a review paper, because the content is based on authors' works. It is necessary to clarify the characteristics of the paper and it has to be shown in the introduction clearly.
2) In figure 1, it is preferable to use a map with contour lines to show the elevation of the land. The marks A-E in figure 1 has to be corresponding to those (A-E) in Figure 5. to avoid misunderstandings. In the same chapter in the paper, use consistent marks. The reviewer cannot identify the location in Figure 1, when authors say "lower lefthand B-2 corner" in the title of figure 3. More consistent way of writing is needed.
3) Figure 8 has to be more complete. A part of the figure is lacking due to low quality scanning. In addition, legends for landuse (land features?) are lacking.
4) Figure 11, Low resolution.
5) Figure 13, the positions for A - P in the map is not clear. Draw again the figure.
6) Figure 16. A. and Figure 16. B. should be Figure 16a and Figure 16b. In addition, it is necessary to improve the resolution of the figures.
7) In figure 40, use m/sec instead of ft/sec. Authors are advised to only change the scale of the color bar, if they don’t wish to calculate again.
8) In figures30 and 31, use mg/L instead of SLUGS/ft^3. Authors are advised to only change the scale of the color bar, if they don’t wish to calculate again.
9) Improve the quality of figures 60-63.
10) Appendix 1 and 2 should be in the appropriate positions in the corresponding figures. In the case of figure 17, the numbers indicating the positions in the map are explained in the figure. The manuscript has to be consistent, although the manuscript deals different target sites.
11) Most of the references are difficult to access. The reviewer thinks that some of the references are not listed appropriately. For example, reference no.3 is taken from "Water" (This journal). In that case, you need to supply the article number and DOI number, instead of the location of the publisher. Follow the instruction of the journal. In the case of other references, please supply volume number, page numbers and DOI in the case of journals.
Author Response
Reviewer 3: Comment 1- Some revision of the intent of the manuscript is given in the revised MS as it is intended to be an easily read introduction to the new field of Paleohydrology that may be unfamiliar to the Water readership base. This is done by including many figures show how CFD results can illuminate new findings to classical archaeological research mainly concerned with socio-economic, socio-pollical and socio-cultural studies. The mS is not intended to be a comprehensive technical paper; those interested after reading this MS can go to the references in figure captions and text for further information of details of th CFD studies for each chapter.
Comment 2- If Figure 1 had contour elevation lines, it would be incomprehensible due to the many numbered sites already present. The A, B C,... and 1, 2, 3,.. axis notations are the best I can do to locate the many sites mentioned and their approximate location in a cited square. Interested readers can take the open access MS and ZOOM the Figure 1 size for clarification.
Comments- Comments 3 to 7- To address the quality of figures comments, Figures 9 and 14 in the revised manuscript have been replaced by better higher definition figures; Figure 16B has been removed and the text adjusted accordingly, Figure 9 has been replaced by a larger figure to show the Priene channel better; Figures 13 and 17 have been replaced by higher definition figures.
Comment 8- the use of English density units (slugs/ft^3) is perhaps not too familiar to the Water readership base more familiar with metric units. To facilitate conversion, those CFD figures using English units for density have figure captions containing the conversion to metric units.
Comment 9- An effort to improve the quality of figures 60-63 has been made.
Comment 10- The lengthy Appendices for given figures are placed at the final end of the text as their inclusion after the figures seems to me to make the chapters in which they occur longer than necessary and a distraction to the intent of the chapter. Readers interested certain chapters can certainly go the references provide in each figure in a chapter to find further information about the subject matter. With respect to your comment on references, many are included in the figure captions now to direct interested readers to further details available in these references. Many of the references are given in my three books and thus are easy to find as these books are listed now on Amazon.
Thank you (and the other two reviewers) for their comments that are now incorporated in the revised manuscript. One goal of this paper is to introduce the field of Paleohydrology to Asian, Middle Eastern and South American Water readers with the hope that they can find inspiration to use their considerable technical talents to bring forward to the world the water engineering accomplishments of their ancient water engineering brothers.

Round 2
Reviewer 1 Report (Previous Reviewer 1)
Dear Author,
Thank you for addressing my recommendation regarding adding references to every figure in the paper. Now, I can see that no new figure has been produced in the current paper.
All the best
Author Response
thanks
Reviewer 2 Report (Previous Reviewer 2)
Author put a lot of work for this manuscript.
It is very long like for a single manuscript.
My decision is to accept te ma uscript, but if possible, please shorten it.
Kind regards,
Reviewer
Author Response
thanks
This manuscript is a resubmission of an earlier submission. The following is a list of the peer review reports and author responses from that submission.
Round 1
Reviewer 1 Report
Dear author,
I enjoyed reading your paper, and I think it is an excellent piece of work that answer several questions regarding the history and development of water management system in the past. However, I have two significant points about the work:
1- There is a degree of mixing between the current and previous papers about the same topic. For example, most maps, diagrams, and photos have already been used in previous articles and have been used now without mentioning the original resources.
2- A topic like that deserves a proper conclusion, so please write several paragraphs in the conclusion section.
Best regards
Author Response
An expanded, revised CONCLUSION section has replaced the original text. The intent of this manuscript is to introduce Paleohydrology to the wide audience of Water readers not familiar with archaeological matters by using a readable, open access text with only basic descriptive information accompanied by many figures - technical details related to CFD usage and properties are provided by access to the specific references for each manuscript section.
Reviewer 2 Report
The research deals with the important issue of the Introduction to paleohydrology: CFD investigations of water supply and distribution systems of ancient old and new world archaeological suites to recover ancient water engineering technology. Remarks:
The manuscript consists of 88 pages. The author should consider shortening it or present some parts in the appendix, as to present the most important issues and follow present brief summaries of ancient sites’ water systems and the use of CFD and modern hydraulic engineering methods to discover the water engineering knowledge base used by ancient water engineers. The manuscript is not formatted according to the journals guidelines.
Author Response
With the introduction of a new means to discover the hydraulic engineering underlying major ancient UNESCO World Heritage sites’ water systems, use of CFD and modern hydraulic engineering analysis was introduced in archaeological studies to Water’s Asian, Middle Eastern and European audience perhaps not too familiar with South American sites as well as sites in their home countries. To deliver on this intent, many example cases from Old and New World sites are presented in readable, open access form to introduce Paleohydrology as the means to discover what ancient water engineers utilized in the design and use of their water systems albeit in formats not preserved or as yet undiscovered in their writings. With this lengthy manuscript available to Water readers, my hope is that it will encourage Water readers to look at ancient sites in their home countries and utilize their engineering knowledge to bring forward to the world what their ancient engineering brothers of centuries past had accomplished.
Reviewer 3 Report
The proposed paper presents an interesting subject, which can be interesting to the scientific community, however, the presentation of conducted research needs considerable editing, therefore the current recommendation is reject.
First of all, the article is too long. The authors should think about reducing the number of examples and maybe separating them into different articles (e.g. South America examples in one paper and the Middle East in another), or reducing the amount of detailed explanation of site specifics, and moving them to the supplementary material.
Second, the figure quality needs to be improved.
Figures 9 and 14 quality is too low, nothing can be observed from the figures.
The quality of figures 16A and 16B is also low, and it looks like a scan from some other older paper/report.
The text in figure 17 is not readable.
The scale should be provided in figures 4, 10, 12...etc. where geometries are being presented.
It is not clear why geometries look like they are from different software, currently, it looks like a compliation of previous research.
The software and CFD simulation methodology are not described. Simulation parameters, mesh parameters, and validation of the conducted analysis are all essential parts of the CFD based papers, which is not present here.
In the introduction previous literature needs to be provided to put in the context conducted research. E.g. provide previous work in paleohydrology with their achievements which supports conducted research.
Disucussion regarding the applicability of the proposed approach should be provided. Obstacles, uncertainties, etc. Something that can be beneficial to future applications of CFD for paleohydrology.
Author Response
For the first comment of the manuscript being “too long and should be broken up into different papers”, first let me first say that Water has no constraint on manuscript size and number of figures and thus serves as an introduction to what Paleohydrology is all about in one introductory paper. My thinking is that if Paleohydrology studies are given in one paper to introduce this new field of discovery related to ancient water systems, then interested readers do not have to look up individual papers in separate journal references published at different dates. Additionally, Water publications are ‘open access’ making research manuscripts instantly available to interested readers. These efficiency factors thus serve to best introduce and describe what Paleohydrology can add to archaeological and water science studies available both to Water’s vast readership base and the general public.
Figures 14 and 17 have been replaced with more better figures as requested. Figures 16 A, B are the best available to show CFD vortex and turbulence effects; flow direction arrows are added in all CFD results showing waterflow direction. A new Figure 9 site photograph replaces the prior Figure 9 to show site detail more accurately and clearly. Scale values are added in quoted figures. All results proceed from use of FLOW-3D software- this has been indicated in all Chapters with references that go into further detail on FLOW-3D use and structure.
Details of the CFD analysis (mesh size, physical parameters used, results analysis) are provided in Chapter references for those interested in use and benefits of the FLOW-3D code for free surface flow problems. Obstacles and uncertainty discussion is offered in the Conclusions section. The CONCLUSIONS section of the manuscript has been augmented by additional paragraphs related to obstacles and uncertainties in Paleohydrology studies as well as advice for those with an engineering background wanting to work in Paleohydrology. The author thanks Reviewer 3 for his comments that have now, with text and figure changes, made Paleohydrology more understandable to readers interested in archaeological studies.